# Nociceptor neurons affect cancer immunosurveillance

Mohammad Balood[1,2,13], Maryam Ahmadi[1,13], Tuany Eichwald[1,3], Ali Ahmadi[1], Abdelilah Majdoubi[4], Karine Roversi[1], Katiane Roversi[1], Christopher T. Lucido[5], Anthony C. Restaino[5], Siyi Huang[6], Lexiang Ji[6], Kai-Chih Huang[6], Elise Semerena[1], Sini C. Thomas[1], Alexandro E. Trevino[7,8], Hannah Merrison[7,8], Alexandre Parrin[7,8], Benjamin Doyle[7,8], Daniel W. Vermeer[5], William C. Spanos[5], Caitlin S. Williamson[5], Corey R. Seehus[7,8], Simmie L. Foster[9], Hongyue Dai[6], Chengyi J. Shu[6], Manu Rangachari[2], Jacques Thibodeau[4], Sonia V. Del Rincon[10], Ronny Drapkin[11], Moutih Rafei[1], Nader Ghasemlou[12], Paola D. Vermeer[5], Clifford J. Woolf[7,8] & Sebastien Talbot[1,12 ✉]

Solid tumours are innervated by nerve fibres that arise from the autonomic and sensory peripheral nervous systems[1–5]. Whether the neo-innervation of tumours by pain-initiating sensory neurons affects cancer immunosurveillance remains unclear. Here we show that melanoma cells interact with nociceptor neurons, leading to increases in their neurite outgrowth, responsiveness to noxious ligands and neuropeptide release. Calcitonin gene-related peptide (CGRP)—one such nociceptor-produced neuropeptide—directly increases the exhaustion of cytotoxic CD8⁺ T cells, which limits their capacity to eliminate melanoma. Genetic ablation of the TRPV1 lineage, local pharmacological silencing of nociceptors and antagonism of the CGRP receptor RAMP1 all reduced the exhaustion of tumour-infiltrating leukocytes and decreased the growth of tumours, nearly tripling the survival rate of mice that were inoculated with B16F10 melanoma cells. Conversely, CD8⁺ T cell exhaustion was rescued in sensory-neuron-depleted mice that were treated with local recombinant CGRP. As compared with wild-type CD8⁺ T cells, *Ramp1⁻ᐟ⁻* CD8⁺ T cells were protected against exhaustion when co-transplanted into tumour-bearing *Rag1*-deficient mice. Single-cell RNA sequencing of biopsies from patients with melanoma revealed that intratumoral *RAMP1*-expressing CD8⁺ T cells were more exhausted than their *RAMP1*-negative counterparts, whereas overexpression of *RAMP1* correlated with a poorer clinical prognosis. Overall, our results suggest that reducing the release of CGRP from tumour-innervating nociceptors could be a strategy to improve anti-tumour immunity by eliminating the immunomodulatory effects of CGRP on cytotoxic CD8⁺ T cells.

Cytotoxic T cells express a variety of receptors, including PD-1 (programmed cell death protein 1), LAG3 (lymphocyte activation gene-3 protein) and TIM3 (T cell immunoglobulin and mucin domain-containing protein 3)[6–8], which inhibit the function of T cells after being activated by their cognate ligands. These checkpoint receptors ensure that immune responses to damage or infection are kept in check, thus preventing overly intense responses that might damage healthy cells[9]. Tumour cells express ligands for these immune checkpoints, which, when activated, block the cytolytic functions of T cells, thereby favouring the survival of cancer cells[9,10].

In prostate cancer, doublecortin-expressing neural progenitors initiate autonomic adrenergic neurogenesis[3], which facilitates the development and dissemination of tumours[2]. In head and neck tumours, a loss of TP53 drives the reprogramming of tumour-innervating sensory nerves into adrenergic neurons that promote tumour growth[1]. The presence of such neo-innervation in cancer, together with the diverse actions of neuropeptides on immune cells[11–18], led us to examine whether the local release of neuropeptides from activated nociceptors could favour cancer growth by suppressing immune surveillance.

[1]Département de Pharmacologie et Physiologie, Université de Montréal, Montréal, Quebec, Canada. [2]Département de Médecine Moléculaire, Faculté de Médecine, Université Laval, Québec, Quebec, Canada. [3]Departamento de Bioquímica, Universidade Federal de Santa Catarina, Florianópolis, Brazil. [4]Département de Microbiologie, Infectiologie et Immunologie, Université de Montréal, Montréal, Quebec, Canada. [5]Cancer Biology and Immunotherapies, Sanford Research, Sioux Falls, SD, USA. [6]Cygnal Therapeutics, Cambridge, MA, USA. [7]F.M. Kirby Neurobiology Center, Boston Children's Hospital, Boston, MA, USA. [8]Department of Neurobiology, Harvard Medical School, Boston, MA, USA. [9]Depression Clinical Research Program, Massachusetts General Hospital, Boston, MA, USA. [10]Department of Oncology, McGill University, Montréal, Quebec, Canada. [11]Penn Ovarian Cancer Research Center, Perelman School of Medicine, University of Pennsylvania, Philadelphia, PA, USA. [12]Department of Biomedical and Molecular Sciences, Queen's University, Kingston, Ontario, Canada. [13]These authors contributed equally: Mohammad Balood, Maryam Ahmadi. ✉e-mail: sebastien.talbot@queensu.ca

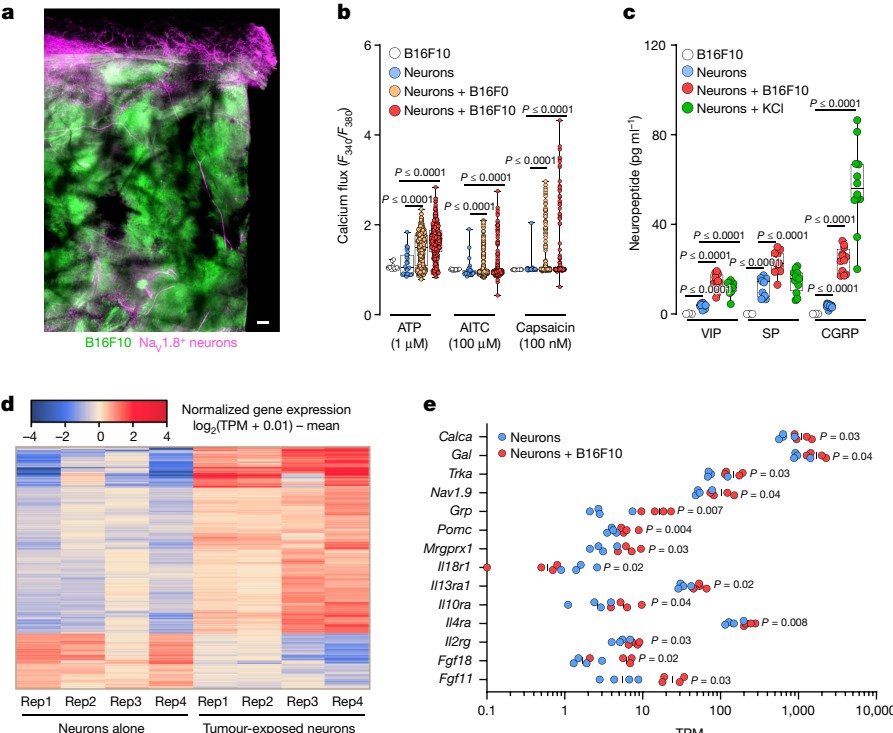

**Fig. 1 | Melanoma cells sensitize nociceptors. a**, Nociceptor (*Nav1.8^cre^::tdTomato^fl/WT^*; magenta) reporter mice were inoculated in the hindpaw with B16F10-eGFP cancer cells (i.d., $2 \times 10^5$ cells; green). Representative image of $Na_V1.8^+$ nerve fibres (magenta) innervating B16F10-eGFP-inoculated mouse skin after 22 days. Scale bar, 200 μm. **b**, In co-culture, B16F0 or B16F10 cells sensitize the response of nociceptors to capsaicin (100 nM), allyl isothiocyanate (AITC, 100 μM) and ATP (1 μM), as measured by calcium flux. A low concentration of the ligands induces a minimal response in control neurons, whereas B16F10 cells show marginal sensitivity to ATP. **c**, Dorsal root ganglion (DRG) neurons co-cultured (96 h) with B16F10 cells release substance P (SP), vasoactive intestinal peptide (VIP) and CGRP. B16F10 cells alone do not release neuropeptides. Stimulation with KCl (40 mM; 30 min) induced a significant release of neuropeptides from cultured neurons. **d,e**, Naive DRG neurons (*Trpv1^cre^::-CheRiff-eGFP^fl/WT^*) were cultured alone or in combination with B16F10-mCherry-OVA cells. After 48 h, the cells were collected, FACS purified and RNA sequenced. Hierarchical clustering of DEGs from the sorted neurons shows distinct groups of transcripts enriched in cancer-exposed TRPV1^+ neurons (**d**), including *Calca* (the gene encoding CGRP; **e**). Data are shown as a representative image (**a**), as box-and-whisker plots (running from minimal to maximal values; the box extends from 25th to 75th percentile and the middle line indicates the median), for which individual data points are given (**b,c**), as a heat map showing normalized gene expression ($\log_2(0.01 + $ transcripts per million reads (TPM))) − mean (**d**) or as a scatter dot plot with medians (**e**). Experiments were independently repeated two (**a**) or three (**b,c**) times with similar results. The sequencing experiment was not repeated (**d,e**). *n* as follows: **a**: $n = 4$; **b**: neurons (29 neurons from 10 mice), B16F10 (16 cells from 10 dishes), neurons + B16F0 (387 neurons from 12 mice), neurons + B16F10 (409 neurons from 12 mice); **c**: neurons ($n = 12$), neurons + B16F10 ($n = 12$), neurons + KCl ($n = 12$), B16F10 ($n = 3$); **d,e**: $n = 4$ per group. *P* values were determined by one-way ANOVA with post-hoc Bonferroni (**b,c**) or two-sided unpaired Student's *t*-test (**e**).

## Melanomas are innervated

Although the expression of genes of neuronal origin is not detected by RNA-sequencing approaches in human malignant cells or immune cells (Extended Data Fig. 1a–c), we observed a significant increase in their expression in biopsies from patients with melanoma[19–22] (Extended Data Fig. 1d). As these clinical data suggested increased innervation of melanomas, we tested for the presence of nociceptor neurons by assessing TRPV1^+ neurons in biopsies from patients with melanoma. TRPV1 immunolabelling was increased by around twofold in the tumour compared to adjacent healthy tissue in each of the ten biopsies examined. The numbers of tumour-infiltrating lymphocytes (TILs) correlated ($R^2 = 0.63$) with increased TRPV1 immunolabelling (Extended Data Fig. 2). These data indicate that melanomas are innervated by sensory neurons and that these neurons may affect the intratumoral numbers of immune cells.

To investigate this in more detail, we inoculated a GFP-expressing melanoma (B16F10-eGFP) cell line into *Nav1.8^cre^::tdTomato^fl/WT^* mice (*Nav1.8* is also known as *Scn10a*). Twenty-two days after implantation, we found abundant $Na_V1.8^+$ nociceptor neurons around and within the tumour (Fig. 1a). RNA sequencing of samples from B16F10-bearing mice revealed that malignant and melanoma-infiltrating immune cells had

no detectable levels of neuronal markers (*Nav1.8* or *Trpv1*), indicating that the $Na_V1.8$ signal could be ascribed to tumour-infiltrating nerves (Extended Data Fig. 3). We next used an in vitro co-culture approach to assess whether malignant cells modulate the function of nociceptor neurons. When co-cultured, TRPV1^+ nociceptors directly extended neurites towards the B16F10-eGFP melanoma cells, and the average length of neurites increased, whereas the overall neuronal arborization or branching decreased (Extended Data Fig. 4a–c). Together, these data indicate that nociceptor outgrowth is enhanced when in proximity to melanoma cells and that skin sensory neuron collaterals sprout directly into the tumour bed. Such tumour neo-innervation may be akin to cancer's neoangiogenesis.

## Melanoma cells sensitize nociceptors

Given that melanoma promotes axonogenesis, leading to tumour innervation (Fig. 1a and Extended Data Fig. 2), we examined whether this physical proximity allows melanomas to modulate the sensitivity of the nociceptor. As nociceptor neurons detect signals from the local environment, we measured changes in calcium flux in response to sub-threshold concentrations of various noxious ligands. When nociceptors were cultured without melanoma cells, few responded

to the ligands at the concentrations selected. However, the number of responsive neurons increased when they were co-cultured with B16F10 cells (Fig. 1b). Similarly, the amplitude of calcium flux responses to the ligands was greater in lumbar DRG neurons (L3–L5) that were collected ipsilateral to a 14-day tumour inoculation in mice, as compared to those collected from mice that were injected with non-tumorigenic keratinocytes (Extended Data Fig. 4d). Signals released from melanoma, therefore, heighten nociceptor sensitivity.

We next tested whether this neuronal hypersensitivity would lead to an increased release of immunomodulatory neuropeptides. In contrast to B16F10 cells alone, DRG neurons co-cultured with B16F10 cells ($5 \times 10^4$ cells, 96 h) actively release CGRP in the medium (Fig. 1c). These data prompted us to test whether exposure to melanoma alters the transcriptome of nociceptor neurons. To do so, we cultured naive DRG neurons (*Trpv1cre::-CheRiff-eGFPfl/WT*) alone or in combination with B16F10-mCherry-OVA cells. After 48 h, TRPV1+ nociceptors were purified by fluorescence-activated cell sorting (FACS) and RNA sequenced. Differentially expressed genes (DEGs) were calculated, and *Calca*–the gene that encodes CGRP–and the NGF receptor *Trka* (also known as *Ntrk1*)were found to be overexpressed in nociceptors that were exposed to cancer (Fig. 1d–e and Extended Data Fig. 4e). Overexpression of *Trka* may help to drive melanoma-induced hypersensitivity to pain, whereas CGRP, when released from activated nociceptors, may immunomodulate TILs.

To identify the mechanism through which melanoma sensitizes nociceptor neurons, we used a co-culture system designed to mimic the interactions that take place in the melanoma microenvironment. Type 1 ($T_{c1}$)-stimulated (ex-vivo-activated by CD3 and CD28, IL-12 and anti-IL-4 for 48 h) OVA-specific cytotoxic CD8+ T cells (OT-I mice), naive DRG neurons (*Trpv1cre::CheRiff-eGFPfl/WT*) and B16F10-mCherry-OVA melanoma cancer cells were cultured alone or in combination. After 48 h, the cells were collected, purified by FACS and RNA sequenced, and DEGs were calculated. Among others, we found that *Slpi* (secretory leukocyte protease inhibitor) was overexpressed in the melanoma cancer cells when co-cultured with either DRG neurons (around 3.6-fold) or OVA-specific cytotoxic CD8+ T cells (around 270-fold), and when exposed to both populations (around 150-fold) (Fig. 2a,b and Extended Data Fig. 5a–e). We also found that B16F10-mCherry-OVA cells, when co-cultured with naive DRG neurons and OVA-specific cytotoxic CD8+ T cells, increased the secretion of SLPI into the culture medium, with this effect being maximal after 48 h (around 200-fold; Fig. 2c).

In addition to protecting epithelial cells from the activity of serine proteases, SLPI enhances the regeneration of transected retinal ganglion cell axons[23] and the proliferation of neural stem cells[24]. Although these data provide evidence of the effect of SLPI on neurons, its role in nociception is unclear. To address this, we measured whether SLPI directly activates cultured DRG neurons using calcium microscopy. We found that SLPI (0.01–10 ng ml$^{-1}$) activates around 20% of DRG neurons and that—consistent with these neurons being nociceptors—SLPI-sensitive neurons were mostly small (with a mean area of 151 μm$^2$) capsaicin-responsive (around 90%) neurons (Fig. 2d,e and Extended Data Fig. 5f–i). Given that SLPI triggered calcium influx, we investigated whether this is the means by which B16F10 cells drive the release of CGRP from neurons (Fig. 1c). SLPI, when used at a concentration similar to that secreted by melanoma cells (Fig. 2c), induced the release of CGRP from cultured naive DRG neurons (Fig. 2f). Finally, we sought to test whether SLPI can drive pain hypersensitivity in vivo. When administered into the right hindpaw of naive mice, SLPI generated transient thermal hypersensitivity (Extended Data Fig. 5j).

Melanoma-secreted SLPI acts on nociceptors to trigger calcium influx, neuropeptide release and thermal hypersensitivity, which indicates that these sensory neurons detect and react to the presence of cancer cells. Whether this gives the malignant cells a functional advantage over the host cells remains unknown. To assess this, we implanted B16F10-mCherry-OVA cells (intradermally (i.d.), $2 \times 10^5$ cells) into the hindpaw of eight-week-old

male and female mice. We found that mice with larger tumours had a higher proportion of intratumoral PD-1+LAG3+TIM3+ CD8+ T cells and greater hypersensitivity to thermal pain (not shown). Notably, heightened sensitivity to thermal pain positively correlated ($n = 60$; $R^2 = 0.55$, $P < 0.0001$) with increased frequency in intratumoral PD-1+LAG3+TIM3+ CD8+ T cells (Fig. 3a; measured on day 13 after implantation).

## Melanoma-innervating nociceptors control tumour growth

The expression of adrenergic and cholinergic axon markers in tumours correlates with poor clinical outcome[2]. Gastric tumour denervation limits growth and patients who have undergone vagotomy have lower rates of mortality from intestinal cancer[16,25,26]. To investigate the nature of the three-way interaction between cancer, nociceptors and CD8+ T cells, we next used a syngeneic mouse model of triple-negative melanoma, which is an established model of immunosurveillance[9]. B16F10-mCherry-OVA cells were inoculated (i.d., $5 \times 10^5$ cells) into eight-week-old male and female nociceptor-ablated (*Trpv1cre::DTAfl/WT*) or intact (littermate control; *Trpv1WT::DTAfl/WT*) mice. In nociceptor-ablated male and female mice, the median length of survival increased by 2.5-fold (evaluated until day 22; Fig. 3b). In another set of mice that were analysed 16 days after tumour inoculation, we found that genetic ablation of nociceptors reduced tumour growth (Fig 3c). In addition, nociceptor-ablated mice showed an increase in the total number and relative frequency of cytotoxic (IFNγ+, TNF+ or IL-2+) tumour-infiltrating CD8+ T cells, but a reduced proportion of PD-1+LAG3+TIM3+ CD8+ T cells (Fig. 3d,e and Extended Data Fig. 6a,b).

Up to this point, our data suggest that nociceptor neurons are an upstream driver of intratumoral PD-1+LAG3+TIM3+ CD8+ T cells. To assess whether this is indeed the case, we mapped out the kinetics of thermal pain hypersensitivity, increased frequency in intratumoral PD-1+LAG3+TIM3+ CD8+ T cells and tumour growth. When compared to their baseline threshold and to that of sensory-neuron-ablated mice (*Trpv1cre::DTAfl/WT*; $n = 19$), eight-week-old littermate control mice (*Trpv1WT::DTAfl/WT*; $n = 96$) that were inoculated with B16F10-mCherry-OVA (left hindpaw, i.d., $2 \times 10^5$ cells) showed significant thermal hypersensitivity on day 7, an effect that peaked on day 21 (Extended Data Fig. 6c). In these mice, the intratumoral frequency of PD-1+LAG3+TIM3+ (Extended Data Fig. 6d) or IFNγ+ (Extended Data Fig. 6e) CD8+ T cells was significantly increased 12 days after tumour inoculation and peaked on day 19. Finally, B16F10-mCherry-OVA tumour volume peaked on day 22 (Extended Data Fig. 6f). Altogether, these data show that thermal hypersensitivity precedes any significant exhaustion of intratumoral CD8+ T cells by around five days and that pain hypersensitivity develops before the tumour is measurable using a digital caliper (Extended Data Fig. 6g).

Blocking the activity of immune checkpoint proteins releases a cancer-cell-induced 'brake' on the immune system, thereby increasing its ability to eliminate tumours[6,8–10]. Immune checkpoint inhibitors (ICIs), including those that target PD-L1, improve clinical outcomes in patients with metastatic melanoma[8]; however, the efficacy of ICIs varies considerably among patients, half of whom will not benefit[27]. We set out to assess whether the presence (*Trpv1WT::DTAfl/WT*) or absence (*Trpv1cre::DTAfl/WT*) of tumour-innervating nociceptor neurons would affect responsiveness to treatment with anti-PD-L1. Anti-PD-L1 (intraperitoneally (i.p.), days 7, 10, 13 and 16) was given either to mice whose tumour cells (B16F10-mCherry-OVA, i.d., $5 \times 10^5$ cells) were inoculated on the same day, or to mice with established tumours (around 85 mm$^3$; achieved by inoculating *Trpv1cre::DTAfl/WT* around 3 days before). In both scenarios, ablation of nociceptors increased the anti-PD-L1-mediated reduction in tumours and the infiltration of tumour-specific CD8+ T cells (Extended Data Fig. 6h–k).

To test whether the reduction in tumour growth that was observed in the absence of nociceptor neurons depends on their action on

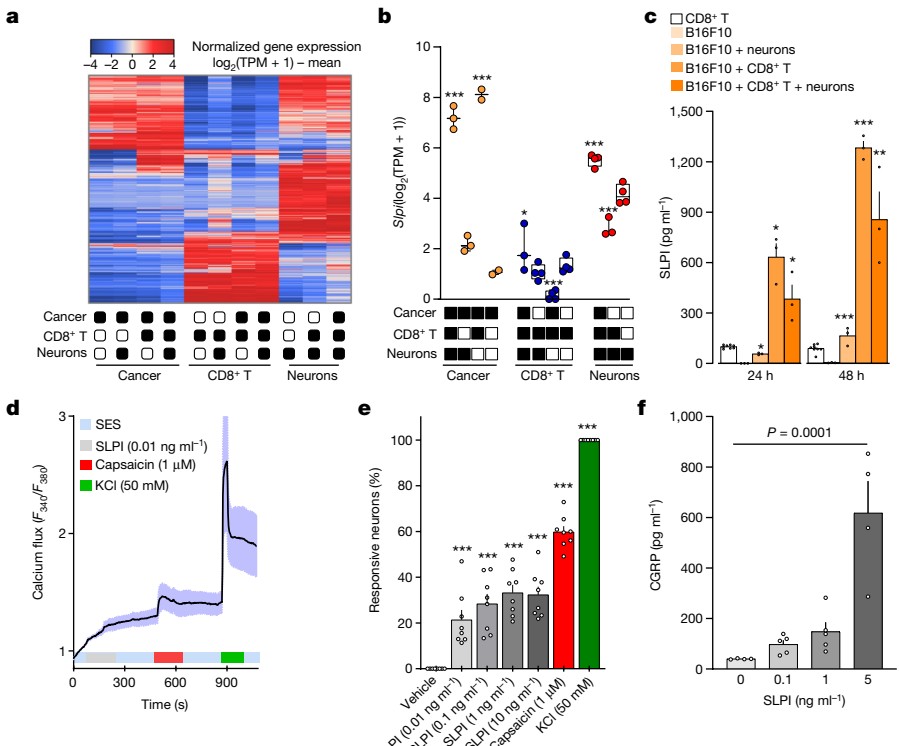

**Fig. 2 | Cancer-secreted SLPI drives the release of CGRP by nociceptor neurons. a–c**, Naive DRG neurons (*Trpv1^cre^::-CheRiff-eGFP^fl/WT^*), B16F10-mCherry-OVA cells and OVA-specific cytotoxic CD8⁺ T cells were cultured alone or in combination. After 48 h, the cells were collected, FACS purified and RNA sequenced. **a**, Hierarchical clustering of sorted neuron molecular profiles depicts distinct groups of transcripts enriched in each group. **b**, DEGs were calculated, and *Slpi* was found to be overexpressed in cancer cells when co-cultured with OVA-specific cytotoxic CD8⁺ T cells, DRG neurons or both populations. **c**, SLPI is secreted by B16F10-mCherry-OVA cells when co-cultured (24 h or 48 h) with naive DRG neurons and OVA-specific cytotoxic CD8⁺ T cells, with a maximal effect after 48 h. **d–f**, Using calcium microscopy, we found that SLPI (10 pg ml⁻¹–10 ng ml⁻¹) activated around 20% of cultured naive DRG neurons (**d,e**). Activation of cultured neurons (3 h) with SLPI also leads to significant release of CGRP (**f**). Data are shown as a heat map showing normalized gene expression (log₂(1 + TPM) − mean (**a**), as box-and-whiskers plots (as defined in Fig. 1b,c) (**b**) or as mean ± s.e.m. (**c–f**). *n* as follows: **a,b**: *n* = 2–4 per groups; **c**: *n* = 3 for all groups except CD8⁺ T cells (*n* = 8); **d**: *n* = 17; **e**: *n* = 8 per group; **f**: 0 ng ml⁻¹ (*n* = 4), 0.1 ng ml⁻¹ (*n* = 5), 1 ng ml⁻¹ (*n* = 5), 5 ng ml⁻¹ (*n* = 4). Experiments in **c–f** were independently repeated three times with similar results. The sequencing experiment was not repeated (**a,b**). *P* values were determined by one-way ANOVA with post-hoc Bonferroni (**b,e,f**) or two-sided unpaired Student's *t*-test (**c**). **P* ≤ 0.05, ***P* ≤ 0.01, and ****P* ≤ 0.001.

immune cells, we compared the respective effects of nociceptors on the growth of an immunogenic and a non-immunogenic isogenic melanoma model. YUMMER1.7 is a highly immunogenic derivative of the *Braf^V600E^Cdkn2a^−/−^Pten^−/−^* cell line modified by ultraviolet (UV) exposure, and provides a clinically relevant model of melanoma[28]. As in the case of B16F10-OVA, ablation of nociceptors decreased the growth of tumours (Extended Data Fig. 6l) and reduced their frequency in intratumoral PD-1⁺LAG3⁺TIM3⁺ CD8 T cells. while increasing their number and cytotoxic potential (IFNγ⁺ or TNF⁺; not shown). By contrast, YUMM1.7 (the parental and non-immunogenic[29] counterpart of YUMMER1.7) showed similar tumour growth (Extended Data Fig. 6m) and a similar frequency of intratumoral PD-1⁺LAG3⁺TIM3⁺ CD8⁺ T cells in both the presence and the absence of nociceptors (not shown).

Next, we assessed whether these differences were due to nociceptor neurons directly modulating intratumoral T cells. We observed no major changes in tumour growth between nociceptor-intact and nociceptor-ablated mice after systemic depletion of CD8⁺ (Fig. 3f) or CD3⁺ (Extended Data Fig. 6n) T cells. Although chemoablation of nociceptor neurons with resiniferatoxin (RTX) reduced tumour growth in B16F10-inoculated wild-type mice (Extended Data Fig. 6o), we found that naive OT-I CD8⁺ T cells enhanced tumour shrinkage when transplanted in RTX-exposed *Rag1^−/−^* mice (Fig. 3g). In doing so, the chemoablation of nociceptor neurons shielded the naive OT-I CD8⁺ T cells from undergoing exhaustion (Fig. 3h). These data imply that the

slower tumour growth found in *Trpv1^cre^::DTA^fl/WT^* and RTX-exposed mice depends on the modulation of CD8⁺ T cells by nociceptors.

Optogenetic activation of skin nociceptor neurons triggers the antidromic release of neuropeptides that mediate anticipatory immunity against microorganisms[30] and potentiate skin immunity[31]. We used transdermal illumination to stimulate tumour-innervating Na_V1.8⁺ channelrhodopsin-expressing neurons (*Nav1.8^cre^::ChR2^fl/WT^*). Daily stimulation with blue light enhanced the growth of B16F10 when exposure began in mice bearing visible (around 20 mm³) or well-established (around 200 mm³) tumours (Extended Data Fig. 6p). This increase in tumour volume was also linked to an increase in the intratumoral levels of CGRP, confirming the engagement of pain-transmitting neurons (Extended Data Fig. 6q). Laser exposure had no effect on tumour growth in light-insensitive mice (*Nav1.8^WT^::ChR2^fl/WT^*; not shown).

The neonatal or embryonic ablation of neuronal subsets may lead to compensatory changes. To circumvent this possibility, we silenced neurons using botulinum neurotoxin A (BoNT/A), a neurotoxic protein produced by *Clostridium botulinum*, which acts by cleaving SNAP25 (ref. [32]). BoNT/A causes a long-lasting (20 days) abolition of neurotransmitter release from skin-innervating neurons[33]. BoNT/A reduces tumour growth in prostate cancer[2] and blocks nociceptor-mediated modulation of neutrophils during skin infection[33]. BoNT/A does not affect the function of cultured B16F10 or CD8⁺ T cells in vitro (Extended Data Fig. 7a–f). When BoNT/A (25 pg μl⁻¹, 50 μl, five i.d. sites) was administered one and three days before the B16F10-OVA cell inoculation, it reduced

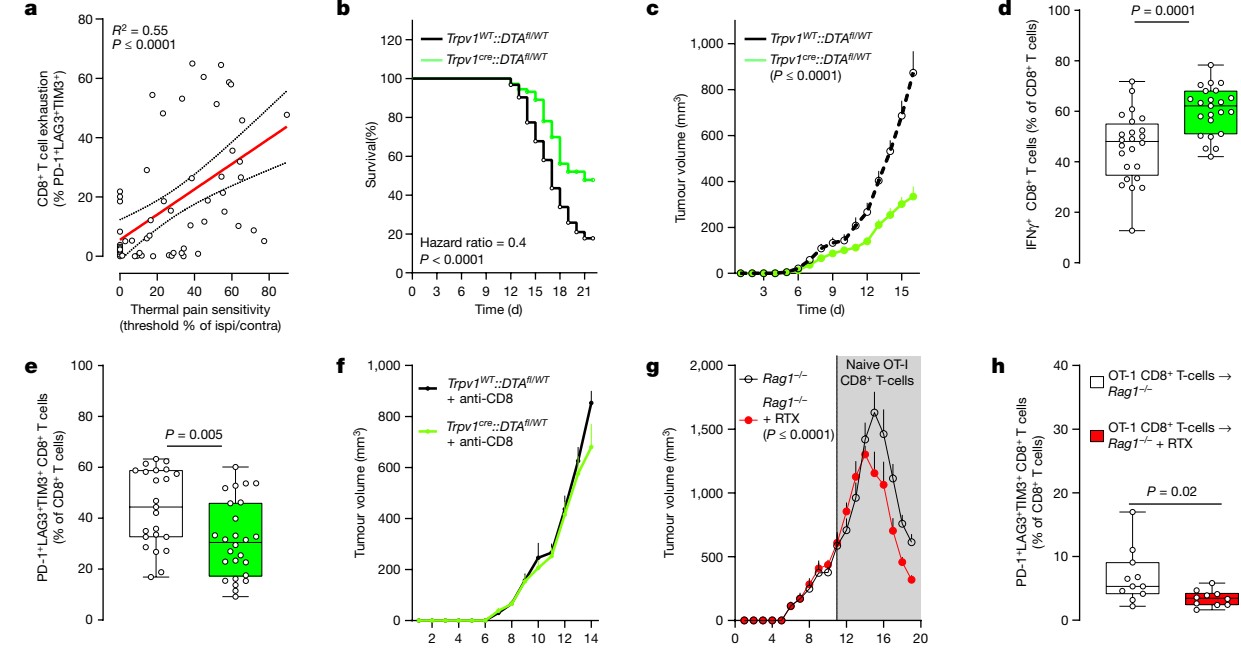

**Fig. 3 | Genetic ablation of nociceptors safeguards anti-tumour immunity.**
**a**, Orthotopic B16F10-mCherry-OVA cells ($2 \times 10^5$ cells, i.d.) were injected into the left hindpaw of wild-type mice. As measured on day 13 after tumour inoculation, intratumoral CD8[+] T cell exhaustion positively correlated with thermal hypersensitivity ($R^2 = 0.55$, $P \leq 0.0001$). The thermal pain hypersensitivity represents the withdrawal latency ratio of the ipsilateral paw (tumour-inoculated) to the contralateral paw. **b**, Orthotopic B16F10-mCherry-OVA ($5 \times 10^5$ cells, i.d.) were inoculated into the flank of eight-week-old male and female mice with sensory neurons intact ($Trpv1^{WT}::DTA^{fl/WT}$) or ablated ($Trpv1^{cre}::DTA^{fl/WT}$). The median length of survival was increased by around 250% in nociceptor-ablated mice (measured until 22 days after inoculation). **c–f**, Sixteen days after tumour inoculation, sensory-neuron-ablated mice have reduced tumour growth (**c**) and increased tumour infiltration of IFNγ[+] CD8[+] T cells (**d**), and the proportion of PD-1[+]LAG3[+]TIM3[+] CD8[+] T cells is decreased (**e**). This reduction in B16F10-mCherry-OVA ($5 \times 10^5$ cells, i.d.) tumour volume was absent in nociceptor-ablated mice whose CD8[+] T cells were systemically depleted (**f**; assessed until day 14; anti-CD8, 200 µg per mouse, i.p., every 3 days). **g,h**, To chemically

deplete their nociceptor neurons, $Rag1^{-/-}$ mice were injected with RTX. Twenty-eight days later, the mice were inoculated with B16F10-mCherry-OVA ($5 \times 10^5$ cells, i.d.). RTX-injected mice that were adoptively transferred with naive OVA-specific CD8[+] T cells (i.v., $1 \times 10^6$ cells, when tumour reached around 500 mm³) showed reduced tumour growth (**g**; assessed until day 19) and exhaustion (**h**) compared to vehicle-exposed $Rag1^{-/-}$ mice. Data are shown as a linear regression analysis ± s.e. (**a**), as a Mantel–Cox regression (**b**), as mean ± s.e.m. (**c,f,g**) or as box-and-whisker plots (as defined in Fig. 1b,c), for which individual data points are given (**d,e,h**). $n$ as follows: **a**: $n = 60$; **b**: intact ($n = 62$), ablated ($n = 73$); **c**: intact ($n = 20$), ablated ($n = 25$); **d**: intact ($n = 24$), ablated ($n = 23$); **e**: intact ($n = 23$), ablated ($n = 26$); **f**: intact + anti-CD8 ($n = 10$), ablated + anti-CD8 ($n = 8$); **g**: vehicle ($n = 12$), RTX ($n = 10$); **h**: vehicle ($n = 11$), RTX ($n = 10$). Experiments were independently repeated two (**a,f–h**) or six (**b–e**) times with similar results. $P$ values were determined by simple linear regression analysis (**a**), Mantel–Cox regression (**b**), two-way ANOVA with post-hoc Bonferroni (**c,g**) or two-sided unpaired Student's $t$-test (**d,e,h**).

subsequent tumour growth and preserved the cytotoxic potential of intratumoral CD8[+] T cells (Extended Data Fig. 7g–n; as measured 18 days after inoculation). Pre-treatment with BoNT/A also reduced the growth of YUMMER1.7 tumours and enhanced anti-PD-L1-mediated tumour regression (Extended Data Fig. 7o,p). When administered to mice with established tumours (around 200 mm³), BoNT/A had limited efficacy (Extended Data Fig. 7g–n). BoNT/A also did not affect tumour growth when given to mice in which TRPV1[+] nociceptor neurons were genetically ablated (Extended Data Fig. 7o), which suggests that its anti-tumour effectiveness depends on the presence of tumour-innervating nociceptor neurons.

We next tested the anti-tumour efficacy of a proven nociceptor-selective silencing strategy[34]. This protocol uses large-pore ion channels (TRPV1) as cell-specific drug-entry ports to deliver QX-314—a charged and membrane-impermeable form of lidocaine—to block voltage-gated sodium (Na$_V$) channels. During inflammation, similar to what we observed in tumour microenvironments, these large-pore ion channels open, which allows QX-314 to permeate the neurons and results in a long-lasting electrical blockade[17]. Although QX-314 did not affect cultured B16F10-mCherry-OVA cells or CD8[+] T cell function in vitro (Extended Data Fig. 8a–f), we confirmed that it silences tumour-innervating nociceptors in vivo, as shown by reduced B16F10-induced release of CGRP

and pain hypersensitivity (Extended Data Fig. 8g–i). We found that vehicle-exposed B16F10-mCherry-OVA-bearing mice succumbed at a 2.7-fold higher rate ($P \leq 0.02$) than QX-314-exposed mice (Extended Data Fig. 8j; measured until day 19). As observed 17 days after tumour inoculation, QX-314-mediated silencing of sensory neurons (0.3%; daily i.d., surrounding the tumour) reduced melanoma growth and limited the exhaustion of intratumoral CD8[+] T cells (Extended Data Fig. 8k–n). Nociceptor silencing also increased the intratumoral numbers of CD8[+] T cells and preserved their cytotoxic potential (IFNγ[+] or TNF[+]) as well as their proliferative capacity (IL-2[+]; Extended Data Fig. 8o–r). Similar to what was observed in nociceptor-ablated mice (Extended Data Fig. 6j–j), silencing tumour-innervating neurons with QX-314 enhanced anti-PD-L1-mediated tumour regression (Extended Data Fig. 8s,t). When administered to mice with an established (around 200 mm³) B16F10-mCherry-OVA tumour, QX-314 still reduced tumour growth and preserved the anti-tumour capacity of CD8[+] T cells (Extended Data Fig. 8k–r), suggesting that it could be used as a therapeutic agent in cancer.

## CGRP attenuates the activity of RAMP1[+] CD8[+] T cells

In breast cancer, tumour-specific sympathetic denervation downregulates the expression of PD-L1, PD-1 and FOXP3 in TILs[15]. Human and

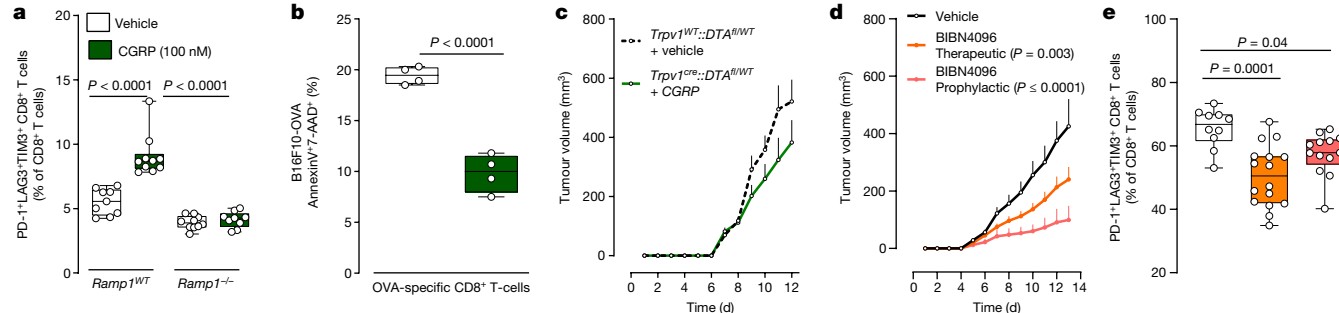

**Fig. 4 | CGRP modulates the activation of CD8⁺ T cells. a,b,** Splenocyte CD8⁺ T cells from wild-type (**a**), *Ramp1⁻/⁻* (**a**) or naive OT-I (**b**) mice were cultured under $T_{c1}$-stimulating conditions (ex-vivo-activated by CD3 and CD28, IL-12 and anti-IL4) for 48 h to generate cytotoxic CD8⁺ T cells. In the presence of IL-2 (10 ng ml⁻¹), the cells were stimulated with CGRP (100 nM; challenged once every two days) for 96 h. Wild-type cytotoxic CD8⁺ T cells showed an increased proportion of PD-1⁺LAG3⁺TIM3⁺ cells; this effect was absent when treating cytotoxic CD8⁺ T cells that were collected from *Ramp1⁻/⁻* mice (**a**). In co-culture (48 h), CGRP (100 nM; once daily) also reduced the ability of OT-I cytotoxic CD8⁺ T cells (4 × 10⁵ cells) to eliminate B16F10-mCherry-OVA cancer cells (**b**). **c,** Orthotopic B16F10-mCherry-OVA cells (5 × 10⁵ cells, i.d.) were inoculated into eight-week-old female mice with sensory neurons intact or ablated. In nociceptor-ablated mice, peritumoral recombinant CGRP injection (100 nM, i.d., once daily) rescues B16F10 growth (assessed until day 12). **d,e,** Orthotopic B16F10-mCherry-OVA cells (5 × 10⁵ cells, i.d.) were inoculated into eight-week-old male and female mice. Starting one day after inoculation (defined as prophylactic),

the RAMP1 antagonist BIBN4096 (5 mg kg⁻¹) was administered systemically (i.p.) once every two days. In another group of mice, BIBN4096 (5 mg kg⁻¹, i.p., every two days) injections were started once the tumour reached a volume of around 200 mm³ (defined as therapeutic). Prophylactic or therapeutic BIBN4096 treatments decreased tumour growth (**d**) and reduced the proportion of intratumoral PD-1⁺LAG3⁺TIM3⁺ CD8⁺ T cells (**e**; assessed until day 13). Data are shown as box-and-whisker plots (as defined in Fig. 1b, c), for which individual data points are given (**a,b,e**), or as mean ± s.e.m. (**c,d**). *n* as follows: **a**: *Ramp1ᵂᵀ* CD8 + vehicle (*n* = 9), *Ramp1ᵂᵀ* CD8 + CGRP (*n* = 10), *Ramp1⁻/⁻* CD8 + vehicle (*n* = 10), *Ramp1⁻/⁻* CD8 + CGRP (*n* = 9); **b**: *n* = 4 per group; **c**: intact + vehicle (*n* = 15), ablated + CGRP (*n* = 11); **d**: vehicle (*n* = 13), BIBN prophylactic (*n* = 16), BIBN therapeutic (*n* = 18); **e**: vehicle (*n* = 10), BIBN prophylactic (*n* = 13), BIBN therapeutic (*n* = 16). Experiments were independently repeated three times with similar results. *P* values were determined by one-way ANOVA with post-hoc Bonferroni (**a,e**), two-sided unpaired Student's *t*-test (**b**) or two-way ANOVA with post-hoc Bonferroni (**c,d**).

mouse cytotoxic CD8⁺ T cells express multiple neuropeptide receptors (10 or more), including the CGRP receptor RAMP1 (Extended Data Figs. 1b and 3b). Given that nociceptors readily interact with CD8⁺ T cells in culture and that the neuropeptides they release block anti-bacterial immunity[33,35–37], we aimed to test whether these mediators drive the expression of immune checkpoint receptors in CD8⁺ T cells. First, splenocyte-isolated CD8⁺ T cells were cultured under type 1 ($T_{c1}$) CD8⁺ T cell-stimulating conditions for two days and then co-cultured with DRG neurons for an additional four days. We found that nociceptor stimulation with capsaicin increased the proportion of PD-1⁺LAG3⁺TIM3⁺-expressing CD8⁺ T cells but decreased the levels of IFNγ⁺, TNF⁺ and IL-2⁺. Capsaicin had no measurable effect on CD8⁺ T cells in the absence of DRG neurons (Extended Data Fig. 9a,b). When $T_{c1}$-activated CD8⁺ T cells were exposed to fresh conditioned medium (1:2 dilution) collected from KCl (50 mM)-stimulated DRG neurons, this treatment increased the proportion of PD-1⁺LAG3⁺TIM3⁺ cytotoxic CD8⁺ T cells and reduced that of IFNγ⁺ cells (Extended Data Fig. 9c,d; measured after four days of co-culture). These effects were prevented when the cytotoxic CD8⁺ T cells were challenged (1:2 dilution) with fresh KCl-induced conditioned medium from BoNT/A-silenced neurons (50 pg per 200 µl) or when they were co-exposed to the RAMP1 blocker CGRP₈₋₃₇ (2 µg ml⁻¹; Extended Data Fig. 9c,d). To confirm that nociceptor-released neuropeptides drive T cell exhaustion, we exposed $T_{c1}$-activated CD8⁺ T cells to CGRP. CGRP-treated cells expressing wild-type RAMP1 showed increased exhaustion and limited cytotoxic potential. These effects were absent in CGRP-exposed CD8⁺ T cells that were collected from CGRP-receptor-knockout (*Ramp1⁻/⁻*) mice (Fig. 4a and Extended Data Fig. 9e,f).

We then assessed whether neuropeptides released by nociceptor neurons blunt the anti-tumour responses of cytotoxic CD8⁺ T cells through exhaustion. OT-I cytotoxic T cells induced robust apoptosis of cultured B16F10-mCherry-OVA cells (AnnexinV⁺7AAD⁺ B16F10-mCherry-OVA; Extended Data Fig. 9g–i). However, this apoptosis of B16F10-mCherry-OVA cells was decreased when the T cells were exposed to capsaicin- or KCl-stimulated neuron-derived conditioned medium, or when the cells were stimulated with CGRP (Fig. 4b and Extended Data Fig. 9g–i). OT-I

cytotoxic T cells did not eliminate cultured B16F10-mCherry-OVA when co-exposed to KCl-induced neuron-conditioned medium supplemented with the RAMP1 blocker CGRP₈₋₃₇ (2 µg ml⁻¹; Extended Data Fig. 9h). When taken together with previous evidence that CGRP limits the activity of CD8⁺ T cells[12,38], our data suggest that, through the CGRP–RAMP1 axis, nociceptors lead to the functional exhaustion of CD8⁺ T cells, as defined by a simultaneous loss of expression of cytotoxic molecules (that is, IFNγ and TNF) and proliferative capacity (IL-2), increased co-expression of several exhaustion markers (PD-1⁺LAG3⁺TIM3⁺) and a reduced capacity to eliminate malignant cells.

Nociceptor-produced neuropeptides reduce immunity against bacteria[37] and fungi[39], and promote cytotoxic CD8⁺ T cell exhaustion (Fig. 4a,b and Extended Data Fig. 9). Given that nociceptor-released CGRP is increased when cultured with B16F10 cells (Fig. 1c) or exposed to SLPI (Fig. 2f), and that tumour-infiltrating nociceptor neurons over-express *Calca* (Fig. 1d,e), we next sought to test whether the intratumoral levels of CGRP correlate with CD8⁺ T cell exhaustion. To do this we used an *Nav1.8ᶜʳᵉ* driver to ablate most mechano- and thermosensitive nociceptors with diphtheria toxin (*Nav1.8ᶜʳᵉ::DTAᶠˡ/ᵂᵀ*)[17,37]. When compared with melanoma-bearing littermate controls (*Nav1.8ᵂᵀ::DTAᶠˡ/ᵂᵀ*), the ablation of Na_v1.8⁺ sensory neurons preserved the functionality of intratumoral CD8⁺ T cells (Extended Data Fig. 10a–d). In both groups of mice, the proportion of intratumoral CGRP directly correlated with the frequency of PD-1⁺LAG3⁺TIM3⁺ CD8⁺ T cells (Extended Data Fig. 10e).

We then set out to rescue CGRP levels (by daily intratumoral injection) in sensory-neuron-ablated mice and measured the effect on tumour growth and TIL exhaustion. At 11 days after inoculation, CGRP-treated sensory-neuron-ablated mice (*Trpv1ᶜʳᵉ::DTAᶠˡ/ᵂᵀ*) showed similar tumour growth and CD8⁺ T cell exhaustion to that of nociceptor-intact mice (Fig. 4c and Extended Data Fig. 10f). Next, we treated tumour-bearing mice with the selective RAMP1 antagonist BIBN4096 (5 mg kg⁻¹, i.p., once every two days). The latter was previously found to block neuro–immune interactions during microorganism infections and rescues host anti-bacterial activity[35]. BIBN4096-exposed mice succumb at a rate 2.6-fold lower (*P* ≤ 0.02) than that of vehicle-exposed B16F10-bearing mice (Extended Data Fig. 10g; measured until day 19). As measured

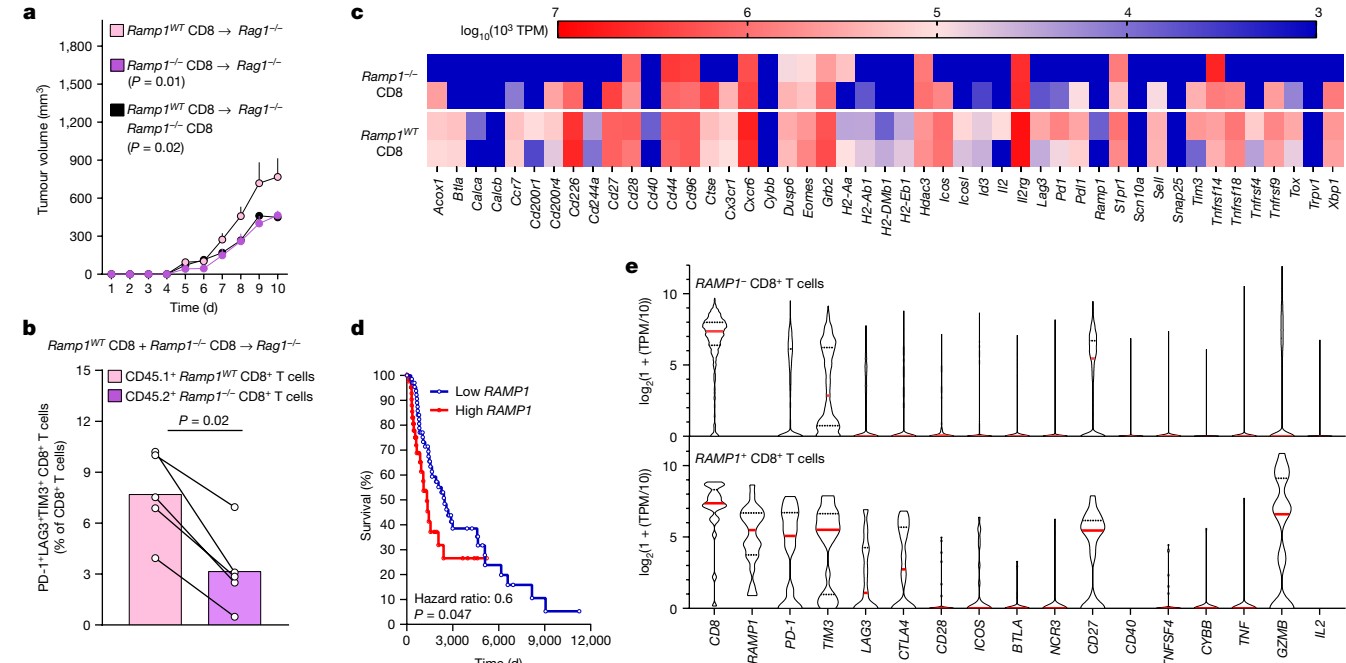

**Fig. 5 | CGRP attenuates the anti-tumour immunity of RAMP1⁺ CD8⁺ T cells.**
**a–c**, Splenocyte CD8⁺ T cells were FACS purified from *Ramp1^WT* (CD45.1⁺) or *Ramp1^−/−* (CD45.2⁺) mice, expanded and stimulated (CD3 and CD28 + IL-2) in vitro. Eight-week-old female *Rag1^−/−* mice were transplanted (i.v., 2.5 × 10⁶ cells) with activated *Ramp1^−/−* or *Ramp1^WT* CD8⁺ T cells or a 1:1 mix of *Ramp1^−/−* and *Ramp1^WT* CD8⁺ T cells. One week after transplantation, the mice were inoculated with B16F10-mCherry-OVA cells (5 × 10⁵ cells, i.d.). Ten days after B16F10 inoculation, we observed greater tumour growth (**a**) in *Ramp1^WT* transplanted mice. Intratumoral *Ramp1^−/−* (CD45.2⁺) and *Ramp1^WT* (CD45.1⁺) CD8⁺ T cells were FACS purified, immunophenotyped (**b**) and RNA sequenced (**c**). *Ramp1^−/−* CD8⁺ T cells showed a lower proportion of PD-1⁺LAG3⁺TIM3⁺ CD8⁺ T cells (**b**) as well as reduced transcript expression of exhaustion markers (**c**). **d**, In silico analysis of The Cancer Genome Atlas (TCGA) data[40] was used to correlate the survival rate of 459 patients with melanoma with the relative *RAMP1* expression (primary biopsy bulk RNA sequencing). In comparison to patients with low *RAMP1* expression, higher *RAMP1* levels correlate with decreased patient survival. **e**, In silico analysis of single-cell RNA sequencing of human melanoma[41] reveals that intratumoral *RAMP1*-expressing CD8⁺ T cells strongly overexpress several immune checkpoint receptors (*PD-1* (also known as *PDCD1*) *TIM3, LAG3, CTLA4*) in comparison to *Ramp1*-negative CD8⁺ T cells. Data are shown as mean ± s.e.m. (**a**), slopegraph (**b**), as a heat map showing normalized gene expression (log₁₀(10³ × TPM) (**c**), as a Mantel−Cox regression (**d**) or as a violin plot (**e**). *n* as follows: **a–c**: *n* = 5 per group; **d**: high (*n* = 45), low (*n* = 68); **e**: *RAMP1⁻* CD8 (*n* = 1,732), *RAMP1⁺* CD8 (*n* = 25). Experiments were independently repeated two (**a,b**) times with similar results. The sequencing experiment was not repeated (**c**). *P* values were determined by two-way ANOVA with post-hoc Bonferroni (**a**), two-sided unpaired Student's *t*-test (**b**) or Mantel−Cox regression (**d**).

on day 13, BIBN4096 (5 mg kg⁻¹, i.p., every other day) reduced B16F10 growth, tumour weight and frequency of PD-1⁺LAG3⁺TIM3⁺ CD8⁺ T cells (Fig. 4d-e and Extended Data Fig. 10h–m). As BIBN4096 showed no effect when administered to nociceptor-ablated mice and did not affect cultured B16F10 cells or CD8⁺ T cell function in vitro (Extended Data Fig. 10n–t), we conclude that the anti-tumour property of BIBN4096 relies on the presence of active nociceptor neurons.

To directly address whether RAMP1 is the main driver of CD8⁺ T cell exhaustion, we transplanted *Rag1^−/−* mice with *Ramp1^−/−* or *Ramp1* wild-type (*Ramp1^WT*) CD8⁺ T cells (intravenously (i.v.), 2.5 × 10⁶) or a 1:1 mixture of both. Although we retrieved similar numbers of CD8⁺ T cells across all three groups (Extended Data Fig. 10u), limited B16F10-OVA tumour growth (Fig. 5a) was found in mice that received the *Ramp1^−/−* CD8⁺ T cells− which are not responsive to CGRP. The relative proportion of intratumoral PD-1⁺LAG3⁺TIM3⁺ CD8⁺ T cells was also lower in *Ramp1^−/−*-transplanted *Rag1^−/−* mice (Extended Data Fig. 10v). In *Rag1^−/−* mice co-transplanted with RAMP1-expressing and -non-expressing CD8⁺ T cells, we found that within the same tumour, the relative proportion of intratumoral PD-1⁺LAG3⁺TIM3⁺ CD8⁺ T cells was lower in *Ramp1^−/−* CD8⁺ T cells (Fig 5b and Extended Data Fig. 10w). Next, we RNA sequenced FACS-purified *Ramp1^WT* and *Ramp1^−/−* CD8⁺ T cells from these tumours. Compared to their *Ramp1^WT* counterparts, we found that intratumoral *Ramp1^−/−* CD8⁺ T cells expressed fewer pro-exhaustion transcription factors (*Tox* and *Eomes*) and markers (*Pdcd1* (encoding PD-1), *Lag3* and *Tim3* (also known as *Havcr2*); Fig. 5c). Overall, CGRP-unresponsive *Ramp1^−/−* CD8⁺ T cells are

protected against undergoing nociceptor-induced exhaustion, which safeguards their anti-tumour responses.

When compared with benign nevi, patient melanomas showed increased expression of *Calca* (Extended Data Fig. 1d). Along with other markers of nociceptor neurons, overexpression of *RAMP1* in these biopsies[40] correlates (*P* ≤ 0.05) with reduced patient survival (Fig. 5d and Extended Data Fig. 11a–l). Whether RAMP1 does this by affecting intratumoral CD8⁺ T cell exhaustion is unknown. To answer this, we analysed two independent unbiased single-cell RNA-sequencing datasets of human melanomas[41,42], and found that around 1.5% of tumour-infiltrating CD8⁺ T cells expressed *RAMP1*. The melanoma-infiltrating *RAMP1⁺* CD8⁺ T cells of the patients over-expressed the immune checkpoint receptors *PD-1* (also known as *PDCD1*), *TIM3* (*HAVCR2*), *LAG3, CTLA4* and *CD27* (Fig. 5e and Extended Data Fig. 11m). This analysis also revealed that tumour-infiltrating CD8⁺ cells collected from patients who were resistant to ICIs markedly overexpressed *RAMP1* (Extended Data Fig. 11n–p). Such an expression profile resembles the functional exhaustion of effector CD8⁺ T cells and suggests that the CGRP receptor RAMP1 influences CD8⁺ T cell exhaustion and the clinical response to ICI in patients with melanoma.

Overall, the genetic ablation of nociceptor neurons decreases the growth of B16F10 tumours by preventing CD8⁺ T cells from undergoing exhaustion, whereas exogenous administration of CGRP has the oppo-site effect. These effects are restricted to immunogenic tumours and are not present in the absence of CD8 T cells. Similar to the pre-clinical

modelling in mice, human data imply that RAMP1-expressing CD8⁺ T cells are more prone to exhaustion and are associated with lower responsiveness to ICIs.

Tumour-innervating nociceptors dampen the immune response to melanoma by upregulating multiple immune checkpoint receptors on cytotoxic CD8⁺ T cells. Blocking the CGRP–RAMP1 axis attenuates this immunomodulatory action of the nervous system on CD8⁺ T cells, thereby safeguarding the anti-tumour immunity of the host (Extended Data Fig. 12) and providing potential therapeutic opportunities by interrupting pro-cancerous neuro–immune links.

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

# Methods

## Secondary use of biopsies as research specimens

The ten melanoma samples used in this study were collected by Sanford Health and classified by a board-certified pathologist. Their secondary use as research specimens (fully de-identified formalin-fixed, paraffin-embedded (FFPE) blocks) was approved under Sanford Health IRB protocol 640 (titled 'Understanding and improving cancer treatment of solid tumours'). As part of this Institutional Review Board (IRB)-approved retrospective tissue analysis, and in accordance with the US Department of Health and Human Services (HHS) secretary's advisory committee on human research protections, no patient consent was necessary as these secondary use specimens were free of linkers or identifiers and posed no more than minimal risk to the human individuals.

## Immunohistochemistry and scoring

In compliance with all the relevant ethical regulations and as approved by Sanford Health IRB protocol 640, ten fully de-identified FFPE melanoma blocks were randomly selected for secondary use as research specimens. The notes of a board-certified pathologist on these specimens are provided in Supplementary Table 1. The specimens were stained using a BenchMark XT slide staining system (Ventana Medical Systems). The Ventana iView DAB detection kit was used as the chromogen, and slides were counterstained with haematoxylin and anti-TRPV1 (Alomone Labs, ACC-030; 1:100). Haematoxylin and eosin (H&E) staining followed standard procedures. TRPV1 immunohistochemistry-stained specimens were analysed on an Olympus BX51 bright-field microscope. Sections were viewed under 20× magnification. Five random fields per sample for both tumour and adjacent normal tissue were analysed and scored on a scale from 0 to 3. Scores were averaged. A score of 0 indicates no appreciated nerve fibres in the evaluated field; +1 indicates sparse nerve fibres; +2 indicates 5–20 nerve fibres; +3 indicates more than 20 nerve fibres.

## IACUC approval

The Institutional Animal Care and Use Committee (IACUC) of Boston Children's Hospital and of the Université de Montréal (Comité de Déontologie de l'Expérimentation sur les Animaux; #21046; 21047) approved all animal procedures.

## Housing of mice

Mice were housed in standard environmental conditions (12-h light–dark cycle; 23 °C; food and water ad libitum) at facilities accredited by the Canadian Council of Animal Care (Université de Montréal) or the Association for Assessment and Accreditation of Laboratory Animal Care (Boston Children's Hospital).

## IACUC end-points

As per our IACUC-approved protocol, the following end-points were used in all of the experiments and were not exceeded. Along with excessive body weight loss (maximum of 10%), the end-points include excessive tumour volume (10% of the mouse's body weight; around 17 mm × 17 mm), skin ulceration, necrosis, bleeding, infection and self-inflicted injury, prostration, lethargy, unresponsiveness to stimulation and/or lack of grooming.

## Mouse lines

Six-to-twenty-week-old male and female C57BL6J (Jax, 000664); CD45.1[+] C57BL6J (Jax, 002014), $Ramp1^{-/-}$ (Jax, 031560), $Rag1^{-/-}$ (Jax, 002216), OT-I (Jax, 003831)[43], $Trpv1^{cre}$ (Jax, 017769)[44], $ChR2^{fl/fl}$ (Jax, 012567)[45], $tdTomato^{fl/fl}$ (Jax, 007908)[46], $DTA^{fl/fl}$ (Jax, 009669)[47] or $DTA^{fl/fl}$ (Jax, 010527), $QuasAr2$-dark $mOrange2$-CheRiff-eGFP$^{fl/fl}$ (referred to in the text as CheRiff-eGFP$^{fl/fl}$; Jax, 028678)[48] mice were purchased from the Jackson Laboratory. $Nav1.8^{cre}$ mice[49] were supplied by R. Kuner and

J. Wood. Excluding CD45.1[+] mice, all other lines were backcrossed for more than six generations on a C57BL6/J background (H-2Kb). Although Capecchi's $DTA^{fl/fl}$ (Jax, 010527) was created on a mixed C57BL6J/129 background, both haplotypes are H-2Kb. All these mice are therefore fully compatible with being transplanted with B16F10-derived cells (C57BL6/J background (H-2Kb)).

We used the Cre–lox toolbox to engineer the various mice lines used ($Trpv1^{cre}::DTA^{fl/WT}$, $Trpv1^{cre}::CheRiff-eGFP^{fl/WT}$, $Trpv1^{cre}::tdTomato^{fl/WT}$, $Nav1.8^{cre}::DTA^{fl/WT}$, $Nav1.8^{cre}::tdTomato^{fl/WT}$; $Nav1.8^{cre}::ChR2^{fl/WT}$ and littermate control) by crossing heterozygote Cre mice with homozygous loxP mice. Mice of both sexes were used for these crosses. All Cre driver lines used were viable and fertile, and abnormal phenotypes were not detected. Offspring were tail-clipped and tissue was used to assess the presence of the transgene by standard PCR, as described by The Jackson Laboratory or the donating investigators. Offspring of both sexes were used at 6–20 weeks of age.

## Cell lines

B16F0[50] (ATCC, CRL-6322), B16F10[51] (ATCC, CRL-6475), B16F10-mCherry-OVA[52] (M. F. Krummel, UCSF), B16F10-eGFP (Imanis, CL053), YUMM1.7[29] (ATCC, CRL-3362), and non-tumorigenic keratinocytes (CellnTEC, MPEK-BL6100) were cultured in complete Dulbecco's modified Eagle's medium high glucose (DMEM; Corning, 10-013-CV) supplemented with 10% fetal bovine serum (FBS; Seradigm, 3100) and 1% penicillin–streptomycin (Corning, MT-3001-Cl), and maintained at 37 °C in a humidified incubator with 5% $CO_2$. YUMMER1.7[28] (M. Bosenberg, Yale University) cells were cultured in DMEM F12 (Gibco, 11320033) supplemented with 10% FBS, 1% penicillin–streptomycin (Corning, MT-3001-Cl) and MEM nonessential amino acids (Corning, 25-025CI), and maintained at 37 °C in a humidified incubator with 5% $CO_2$.

All the cell lines tested negative for mycoplasma, and none are listed by the International Cell Line Authentication Committee registry (v.11). Non-commercial cell lines (B16F10-OVA, B16F10-OVA-mCherry and B16F10-eGFP) were authenticated using antibodies (against OVA, eGFP and mCherry) and/or imaging as well as morphology and growth properties. Commercial cell lines were not further authenticated.

## Cancer inoculation and volume measurement

Cancer cells were resuspended in phosphate buffered saline (PBS; Corning, 21040CV) and injected into the mouse's skin in the right flank ($5 \times 10^5$ cells, i.d., 100 μl) or hindpaw ($2 \times 10^5$ cells, i.d., 50 μl). Growth was assessed daily using a handheld digital caliper and tumour volume was determined by the formula ($L \times W^2 \times 0.52$) (ref. [53]), in which $L$ = length and $W$ = width.

## BoNT/A

BoNT/A[35] (List Biological Labs, 130B) was injected (25 pg μl$^{-1}$, i.d., five neighbouring sites injected with 20 μl) into the skin three days and one day before tumour inoculation (defined as prophylactic). BoNT/A (25 pg μl$^{-1}$; i.d., five neighbouring sites injected with 20 μl) was injected around the tumour one day and three days after the tumour reached a volume of around 200 mm³ (defined as therapeutic) in other groups of C57BL/6J mice.

## QX-314

Starting one day after tumour inoculation (defined as prophylactic), QX-314 (ref. [34]; Tocris, 2313; 0.3%) was injected (i.d., 100 μl) daily at five points around the tumour. In another group of mice, QX-314 daily injection started once the tumour reached a volume of around 200 mm³ (defined as therapeutic).

## BIBN4096

Starting one day after tumour inoculation, BIBN4096 (ref. [33]; Tocris, 4561; 5 mg kg$^{-1}$) was administered systemically (i.p., 50 μl) on alternate days to eight-week-old male and female mice (defined as prophylactic).

In another group of mice, BIBN4096 (5 mg kg$^{-1}$) was administered systemically (i.p., 50 µl) on alternate days once the tumour reached a volume of around 200 mm$^3$ (defined as therapeutic).

## RTX

RTX (ref. [33]; Alomone Labs, R-400) was injected (subcutaneously; s.c.) in three dosages (30, 70 and 100 µg kg$^{-1}$) into the right flank of *Rag1*$^{-/-}$ and C57BL/6J mice of around three weeks of age. Denervation was confirmed 28 days after RTX by an absence of pain withdrawal reflex (paw flinching) when exposed to heat (see 'Thermal hypersensitivity' for details on the test).

## Survival

In specific groups of mice, orthotopic B16F10-mCherry-OVA (5 × 10$^5$ cells, i.d.) cells were administered to intact and nociceptor-ablated mice and survival was measured until day 22 and determined by the tumour reaching a volume of 1,000 mm$^3$ or greater, or according to the ethical end-points described above. In B16F10-mCherry-OVA-inoculated mice treated with QX-314 or BIBN4096, survival was measured until day 19 and determined by the tumour reaching a volume of 800 mm$^3$ or greater, or according to the ethical end-points described above. As the survival analysis of vehicle-injected, QX-314-treated and BIBN4096-treated mice was performed simultaneously, the same group of vehicle-injected mice is shown in the respective panels for QX-314 and BIBN4096.

## iDISCO imaging

Whole-mount immunohistochemistry of tumours was performed using an iDISCO protocol[54,55] with methanol pre-treatment optimized for tumours. In brief, adult mice (eight weeks old) were perfused with 25 ml of PBS (HyClone) and 25 ml of 4% paraformaldehyde (PFA; Sigma) sequentially at room temperature. Tumours were post-fixed with 4% PFA for 6 h at 4 °C. For methanol pre-treatment, fixed tumours were washed sequentially in 50% methanol (in PBS) for 1 h and 100% methanol for 1 h, and then bleached in 5% H$_2$O$_2$ in 20% DMSO and methanol overnight at 4 °C. Tumours were subsequently rehydrated in 100% methanol for 1 h twice, 20% DMSO and methanol for 1 h twice, 50% methanol in PBS for 1 h, PBS for 1 h twice and PBS and 0.2% Triton X-100 for 1 h twice at room temperature. Tumours were then left in PBS, 0.2% Triton X-100, 20% DMSO and 0.3 M glycine (Sigma) overnight at room temperature and blocked in PBS, 0.2% Triton X-100, 10% DMSO, 6% donkey serum (Jackson ImmunoResearch) and anti-CD16/CD32 (Fc block; Bio X Cell) overnight at room temperature. Tumours were subsequently washed in PBS, 0.2% Tween-20 and 10 mg ml$^{-1}$ heparin (PTwH; Sigma) for 1 h twice at room temperature before incubation with antibody mix (GFP (Aves Labs) at 1:500, mCherry (OriGene) at 1:500, in PTwH, 5% DMSO, 3% donkey serum and Fc block 1:100 for four days at room temperature). Tumours were extensively washed in PTwH at least six times over the course of one day at room temperature. Tumours were further incubated with a secondary panel of species-specific anti-IgG (H+L) Alexa Fluor 488 or 546-conjugated antibodies (Invitrogen or Jackson ImmunoResearch), all at 1:500, in PTwH, 5% DMSO, 3% donkey serum and Fc block 1:100 for three more days at room temperature. Tumours were washed in the same way as after primary antibody incubation for one day. Immunolabelled tumours were then processed for clearing, which included sequential incubation with 50% methanol for 1 h at room temperature, 100% methanol for 1 h three times at room temperature, and a mixture of one part benzyl alcohol (Sigma):two parts benzyl benzoates (Sigma) overnight at 4 °C. For tdTomato and GFP immunolabelling, mCherry and GFP antibodies were preabsorbed against tumours from *tdTomato*$^-$ mice overnight at room temperature before use. Cleared whole-mount tissues were imaged in BABB between two cover glasses using Olympus FV3000 confocal imaging system.

## Tumour and tumour-draining lymph node digestion

Mice were euthanized when the tumour reached a volume of 800–1,500 mm$^3$ (refs. [50,51,56]). Tumours and their draining lymph nodes were collected. Tumours were enzymatically digested in DMEM + 5% FBS (Seradigm, 3100) + 2 mg ml$^{-1}$ collagenase D (Sigma, 11088866001) + 1 mg ml$^{-1}$ collagenase IV (Sigma, C5138-1G) + 40 µg ml$^{-1}$ DNAse I (Sigma, 10104159001) under constant shaking (40 min, 37 °C). The cell suspension was centrifuged at 400$g$ for 5 min. The pellet was resuspended in 70% Percoll gradient (GE Healthcare), overlaid with 40% Percoll and centrifuged at 500$g$ for 20 min at room temperature with acceleration and deceleration at 1. The cells were aspirated from the Percoll interface and passed through a 70-µm cell strainer. Tumour-draining lymph nodes were dissected in PBS + 5% FBS, mechanically dissociated using a plunger, strained (70 µm) and washed with PBS.

## Immunophenotyping

Single cells were resuspended in FACS buffer (PBS, 2% fetal calf serum and EDTA), and stained with ZombieAqua (15 min, room temperature; BioLegend, 423102) or a Viability Dye eFluor 780 (15 min, 4 °C; eBioscience, 65-0865-14). The cells were washed and Fc-blocked (0.5 mg ml$^{-1}$, 15 min, 4 °C; BD Biosciences, 553141). Finally, the cells were stained (30 min, 4 °C) with one of anti-CD45–BV421 (1:100, BioLegend, 103134), anti-CD45.1–BV421 (1:100, BioLegend, 110732), anti-CD45.2–BV650 (1:100, BioLegend, 109836), anti-CD45-Alexa Fluor 700 (1:100, BioLegend, 103128), anti-CD11b-APC/Cy7 (1:100, BioLegend, 101226), anti-CD8-AF700 (1:100, BioLegend, 100730), anti-CD8-BV421 (1:100, BioLegend, 100753), anti-CD8–PerCP/Cyanine5.5 (1:100, BioLegend, 100734), anti-CD8–Pacific Blue (1:100, BioLegend, 100725), anti-CD4–PerCP/Cyanine5.5 (1:100, BioLegend, 100540), anti-CD4-FITC (1:100, BioLegend, 100406), anti-PD-1–PE-Cy7 (1:100, BioLegend, 109110), anti-LAG3–PE (1:100, BioLegend, 125208), anti-LAG3–PerCP/Cyanine5.5 (1:100, BioLegend, 125212) or anti-TIM3–APC (1:100, BioLegend, 119706), washed and analysed using a LSRFortessa or FACSCanto II (Becton Dickinson). Antigen-specific CD8$^+$ T cells were stained with H-2Kb/OVA257-264 (15 min, 37 °C; NIH tetramer core facility), washed and stained with surface markers. Cytokine expression was analysed after in vitro stimulation (PMA–ionomycin; see 'Intracellular cytokine staining').

## Intracellular cytokine staining

Cells were stimulated (3 h) with phorbol-12-myristate 13-acetate (PMA; 50 ng ml$^{-1}$, Sigma-Aldrich, P1585), ionomycin (1 µg ml$^{-1}$, Sigma-Aldrich, I3909) and Golgi Stop (1:100, BD Biosciences, 554724). The cells were then fixed and permeabilized (1:100, BD Biosciences, 554714) and stained with anti-IFNγ–APC (1:100, BioLegend, 505810), anti-IFNγ–FITC (1:100, BioLegend, 505806), anti-TNF–BV510 (1:100, BioLegend, 506339), anti-TNF–BV5711 (1:100, BioLegend, 506349), anti-TNF–PE (1:100, BioLegend, 506306), anti-IL2–Pecy7 (1:100, BioLegend, 503832), anti-IL-2–Pacific Blue (1:100, BioLegend, 503820), anti-IL-2–BV510 (1:100, BioLegend, 503833), and analysed using a LSRFortessa or FACSCanto II (Becton Dickinson).

## In vivo depletion of CD3 or CD8

Anti-mouse CD3 (200 µg per mouse, Bio X Cell, BE0001-1) or anti-mouse CD8 (200 µg per mouse, Bio X Cell, BP0061) were injected (i.p.) three days before B16F10-mCherry-OVA inoculation (5 × 10$^5$ cells; i.d.) and continued every three days. Blood samples were taken twice weekly to confirm depletion, and tumour growth was measured daily.

## In vivo CGRP rescue experiment

*Trpv1*-ablated mice were injected (i.d.) once daily with recombinant CGRP (100 nM) at five points around the tumour (treatment began once the tumour was visible), and tumour growth was measured daily by a handheld digital caliper. Mice were euthanized, and tumour-infiltrating

CD8[+] cell exhaustion was immunophenotyped by flow cytometry using an LSRFortessa or a FACSCanto II (Becton Dickinson).

## Anti-PD-L1 treatment

Orthotopic B16F10-mCherry-OVA cells ($5 \times 10^5$ cells, i.d.) were inoculated into eight-week-old male and female sensory-neuron-intact or -ablated mice. On days 7, 10, 13 and 16 after tumour inoculations, the mice were treated with anti-PD-L1[14,57] (Bio X Cell, BE0101, 6 mg kg$^{-1}$; i.p., 50 µl) or isotype control. Nineteen days after tumour inoculation, the effect of anti-PD-L1 on tumour growth was analysed and TIL exhaustion was immunophenotyped using an LSRFortessa or a FACSCanto II (Becton Dickinson).

## Anti-PD-L1 treatment in mice with similar tumour sizes

Orthotopic B16F10-mCherry-OVA cells ($5 \times 10^5$ cells; i.d.) were injected into a cohort of nociceptor neuron-ablated mice three days before nociceptor-intact mice were injected. Mice from each group with a similar tumour size (around 85 mm$^3$) were selected and exposed to anti-PD-L1[14,57] (Bio X Cell, BE0101, 6 mg kg$^{-1}$, i.p., 50 µl) or isotype control once every three days for a total of nine days. The effect of anti-PD-L1 treatment on tumour growth was analysed until day 18.

One and three days before tumour inoculation, the skin of eight-week-old male and female mice was injected with BoNT/A (25 pg µl$^{-1}$, i.d., five neighbouring sites injected with 20 µl) or vehicle. One day after the last injection, orthotopic B16F10-mCherry-OVA cells ($5 \times 10^5$ cells, i.d.) were inoculated into the area pre-exposed to BoNT/A. On days 7, 10, 13 and 16 after tumour inoculation, the mice were exposed to anti-PD-L1 (6 mg kg$^{-1}$, i.p.) or isotype control. Eighteen days after tumour inoculation, we found that neuron silencing using BoNT/A potentiated anti-PD-L1-mediated tumour reduction.

Orthotopic B16F10-mCherry-OVA ($5 \times 10^5$ cells, i.d.) were injected into mice treated with QX-314 (0.3%, i.d.) two to three days before being given to vehicle-exposed mice. Mice from each group with a similar tumour size (around 100 mm$^3$) were selected and exposed to anti-PD-L1[14] (Bio X Cell, BE0101, 6 mg kg$^{-1}$, i.p.) or isotype control once every three days for a total of nine days. Eighteen days after tumour inoculation, the effect of anti-PD-L1 on tumour growth was analysed, and TIL exhaustion was immunophenotyped using an LSRFortessa or a FACSCanto II (Becton Dickinson).

## Adoptive transfer of *Ramp1*[WT] or *Ramp1*[−/−] CD8 T cells

Total CD8[+] T cells were isolated from the spleen of wild-type (CD45.1[+]) or *Ramp1*[−/−] (CD45.2[+]) mice, expanded and stimulated *in vitro* using a mouse T cell Activation/Expansion Kit (Miltenyi Biotec. #130-093-627). CD8[+] cells from *Ramp1*[−/−] and *Ramp1*[WT] were injected separately or 1:1 mix through tail vein of *Rag1*[−/−] mice. One week after, the mice were inoculated with B16F10-mCherry-OVA cancer cells ($5 \times 10^5$ cells; i.d.), and tumour growth was measured daily using a handheld digital caliper. On day 10, tumours were collected and *Ramp1*[−/−] (CD45.2[+]) and *Ramp1*[WT] (CD45.1[+]) CD8[+] T cells were immunophenotyped using a FACSCanto II (Becton Dickinson) or FACS purified using a FACSAria IIu cell sorter (Becton Dickinson).

## RNA sequencing of adoptive transferred *Ramp1*[WT] or *Ramp1*[−/−] CD8 T cells

For FACS-purified cells, *Ramp1*[−/−] and *Ramp1*[WT] CD8[+] T cell RNA-sequencing libraries were constructed using KAPA Hyperprep RNA ($1 \times 75$ bp) following the manufacturer's instructions. Nextseq500 (0.5 Flowcell High Output; 200 M defragments; 75 cycles single-end read) sequencing was performed on site at the Institute for Research in Immunology and Cancer (IRIC) genomic centre. Sequences were trimmed for sequencing adapters and low-quality 3′ bases using Trimmomatic v.0.35 and aligned to the reference mouse genome version GRCm38 (gene annotation from Gencode v.M23, based on Ensembl 98) using STAR v.2.5.1b (ref. [58]). Gene expression levels were obtained both as a read count

directly from STAR and computed using RSEM to obtain normalized gene and transcript level expression, in TPM values, for these stranded RNA libraries. DESeq2 v.1.18.1 (ref. [59]) was then used to normalize gene read counts. Individual cell data are shown as a $\log_{10}$ of (TPM × 1,000). These data have been deposited in the National Center for Biotechnology Information (NCBI)'s Gene Expression Omnibus (GEO)[60] (GSE205863).

## Adoptive T cell transfer in mice treated with RTX

CD8[+] T cells were isolated from OT-I mice spleens and magnet sorted (StemCell; 19858). Naive CD8[+] T cells (CD8[+]CD44[low]CD62L[hi]) cells were then purified by FACS using an FACSAria IIu cell sorter (Becton Dickinson) and injected ($1 \times 10^6$ cells, i.v., tail vein) into vehicle- or RTX-exposed *Rag1*[−/−] mice.

## Mechanical hypersensitivity

B16F10-mCherry-OVA ($2 \times 10^5$ cells, i.d.) or non-cancerous keratinocytes (MPEK-BL6; ($2 \times 10^5$ cells, i.d.) were inoculated intradermally in the left hindpaw of the mice. On alternate days, mechanical sensitivity was evaluated using von Frey filaments (Ugo Basile, 52-37450-275). To do so, the mice were placed in a test cage with a wire mesh floor and allowed to acclimatize (three consecutive days: 1 h per session). Von Frey filaments of increasing size (0.008–2 g) were applied to the plantar surface and the response rate was evaluated using the up-down test paradigm[61].

## Thermal hypersensitivity

To measure thermal sensitivity, the mice were placed on a glass plate of a Hargreaves's apparatus (Ugo Basile)[62] and stimulated using radiant heat (infrared beam). The infrared beam intensity was set at 44 and calibrated to result in a withdrawal time of around 12 seconds in acclimatized wild-type mice. An automatic cut-off was set to 25 s to avoid tissue damage. The radiant heat source was applied to the dorsal surface of the hindpaw and latency was measured as the time for the mouse to lift, lick or withdraw the paw[62].

Before any treatment, the mice were allowed to acclimatize in the apparatus (minimum of three consecutive days: 1 h per session) and three baseline measurements were taken on the following day. In some instances, B16F10-mCherry-OVA ($2 \times 10^5$ cells; i.d.) or non-cancerous keratinocytes (MPEK-BL6; ($2 \times 10^5$ cells; i.d.) were inoculated intradermally to the mouse's left hindpaw and thermal pain hypersensitivity was measured on alternate days (10:00). In other instances, SLPI (1 µg per 20 µl) or saline (20 µl) were injected in the left and right hindpaw, respectively, and thermal hypersensitivity was measured in both hindpaws at 1, 3 and 6 h after treatment.

## Kinetics of pain and intratumoral CD8 T cell exhaustion

We implanted B16F10-mCherry-OVA ($2 \times 10^5$ cells, i.d.) in several groups of littermate control (*Trpv1*[WT]*::DTA*[fl/WT]; $n = 96$) and nociceptor-ablated (*Trpv1*[cre]*::DTA*[fl/WT]; $n = 18$) mice. We then evaluated the level of thermal hypersensitivity (daily), tumour size (handheld digital caliper), and intratumoral CD8[+] T cell exhaustion (flow cytometry) at the time of euthanasia (days 1, 4, 7, 8, 12, 13, 14, 19 and 22). We processed these data by determining the percentage change of each data point to the maximal value obtained in the pain, CD8[+] T cell exhaustion and tumour size datasets, and then presented these data as percentages of the maximum (100%).

## Optogenetic stimulation

Orthotopic B16F10-mCherry-OVA cells ($5 \times 10^5$ cells, i.d.) were inoculated into the left flank of eight-week-old transgenic male mice expressing the light-sensitive protein channelrhodopsin 2 under the control of the *Nav1.8* promoter (*Nav1.8*[cre]*::ChR2*[fl/WT]). Optogenetic stimulation (3.5 ms, 10 Hz, 478 nm, 60 mW, in a 0.39-NA fibre placed 5–10 mm from the skin, for 20 min) started either when the tumour was visible (around 20 mm$^3$; 5 days after inoculation) or when it reached a volume of 200 mm$^3$ (8 days after inoculation) and lasted up to 14 days after tumour inoculation. The control mice (*Nav1.8*[cre]*::ChR2*[fl/WT]) were tumour-injected but

not light-stimulated. Groups of littermate control ($Nav1.8^{WT}::ChR2^{fl/WT}$) mice were light-stimulated and showed no response (not shown).

## CGRP release from skin explant
Tumour-surrounding skin was collected using 10-mm punch biopsies from nociceptor-intact ($Nav1.8^{WT}::DTA^{fl/WT}$), nociceptor-ablated ($Nav1.8^{cre}::DTA^{fl/WT}$), light-sensitive nociceptor ($Nav1.8^{cre}::ChR2^{fl/WT}$) or wild-type mice 3 h after exposure to vehicle (100 μl), QX-314 (0.3%, 100 μl) or BoNT/A (25 pg μl$^{-1}$, 100 μl). The biopsies were transferred into 24-well plates and cultured in DMEM containing 1 μl ml$^{-1}$ of protease inhibitor (Sigma, P1860) and capsaicin (1 μM, Sigma, M2028). After a 30-min incubation (37 °C), the supernatant was collected and the release of CGRP was analysed using a commercial enzyme-linked immunosorbent assay (ELISA)[35] (Cayman Chemical, 589001).

## CGRP release triggered by SLPI
$1 \times 10^4$ naive DRG neurons were cultured for 24 h in complete DMEM (10% FBS, 1% penicillin–streptomycin, 1 μl ml$^{-1}$ protease inhibitor) and subsequently stimulated (3 h) with vehicle or SLPI (0.1–5.0 ng ml$^{-1}$). After stimulation, the supernatant was collected and CGRP levels were measured using a commercial ELISA kit (Cayman Chemical, 589001).

## Neuron culture
Mice were euthanized, and dorsal root ganglia were dissected out into DMEM medium (Corning, 10-013-CV), completed with 50 U ml$^{-1}$ penicillin and 50 μg ml$^{-1}$ streptomycin (Corning, MT-3001-Cl) and 10% FBS (Seradigm, 3100). Cells were then dissociated in HEPES buffered saline (Sigma, 51558) completed with 1 mg ml$^{-1}$ collagenase IV (Sigma, C0130) + 2.4 U ml$^{-1}$ dispase II (Sigma, 04942078001) and incubated for 80 min at 37 °C. Ganglia were triturated with glass Pasteur pipettes of decreasing size in supplemented DMEM medium, then centrifuged over a 10% BSA gradient and plated on laminin (Sigma, L2020)-coated cell-culture dishes. Cells were cultured with Neurobasal-A medium (Gibco, 21103-049) completed with 0.05 ng μl$^{-1}$ NGF (Life Technologies, 13257-019), 0.002 ng μl$^{-1}$ GDNF (PeproTech, 450-51-10), 0.01 mM AraC (Sigma, C6645) and 200 mM L-glutamine (VWR, 02-0131) and B-27 supplement (Gibco, #17504044).

## Calcium imaging
L3–L5 DRG neurons were collected and co-cultured with B16F10, B16F0 or MPEK-BL6 for 24–48 h. The cells were then loaded with 5 mM Fura-2 AM (BioVision, 2243) in complete Neurobasal-A medium for 30 min at 37 °C, washed in Standard Extracellular Solution (SES, 145 mM NaCl, 5 mM KCl, 2 mM CaCl$_2$, 1 mM MgCl$_2$, 10 mM glucose and 10 mM HEPES, pH 7.5), and the response to noxious ligands (100 nM capsaicin, 100 μM AITC or 1 μM ATP) was analysed at room temperature. Ligands were flowed (15 s) directly onto neurons using perfusion barrels followed by buffer washout (105-s minimum). Cells were illuminated by a UV light source (Xenon lamp, 75 watts, Nikon), 340-nm and 380-nm excitation alternated by an LEP MAC 5000 filter wheel (Spectra services), and fluorescence emission was captured by a Cool SNAP ES camera (Princeton Instruments). The 340/380 ratiometric images were processed, background-corrected and analysed (IPLab software, Scientific Analytics), and Microsoft Excel was used for post-hoc analyses. Responsiveness to a particular ligand was determined by an increase in fluorescence ($F_{340}/F_{380}$) of at least 5–10% above baseline recording (SES). To test neuronal sensitivity in mice inoculated with B16F10 or non-tumorigenic keratinocytes, the mice were euthanized two weeks after inoculation (left hindpaw, i.d.), and L3–L5 DRG neurons were collected and cultured (3 h). Calcium flux to noxious ligands (1 μM capsaicin or 10 μM ATP) was subsequently tested. For SLPI, the DRG neurons were cultured for 24 h, loaded with 5 mM Fura-2 AM in complete Neurobasal-A medium for 45 min at 37 °C and washed into SES, and the responses to noxious ligands (0–10 ng ml$^{-1}$ of mouse recombinant SLPI (LifeSpan BioSciences, LS-G13637-10), 1 μM capsaicin or 50 mM KCl) were analysed at room temperature.

## Immunofluorescence
A total of $2 \times 10^3$ DRG neurons were co-cultured with $2 \times 10^4$ B16F10-mCherry-OVA cells for 24–48 h. The cells were fixed (4% PFA; 30 min), permeabilized (0.1% Triton X-100; 20 min), and blocked (PBS, 0.1% Triton X-100 and 5% BSA; 30 min). The cells were rinsed (PBS), stained, and mounted with vectashield containing DAPI (Vector Laboratories, H-1000). Images were acquired using a Ti2 Nikon fluorescent microscope (IS-Elements Advanced Research v.4.5).

## Neurite length and ramification index
TRPV1$^+$ nociceptors ($Trpv1^{cre}::tdTomato^{fl/WT}$) were cultured alone ($2 \times 10^3$ cells) or co-cultured ($2 \times 10^4$ cells) with B16F10-GFP, B16F0 or non-tumorigenic keratinocytes (MPEK-BL6). After 48 h, cells were fixed (see 'Immunofluorescence'), and images were acquired using a Ti2 Nikon fluorescent microscope. The neurite length of TRPV1$^+$ (tdTomato) neurons was measured using a neurite tracer macro in ImageJ (Fiji, v.1.53c) developed by the Fournier laboratory[63], and the Schoenen ramification index (SRI) was measured by a Sholl analysis[64] macro in ImageJ (Fiji, v.1.53c).

## Isolation of CD8$^+$ T cells
Six-to-eight-week-old male and female mice were euthanized, and their spleens were collected in ice-cold PBS (5% FBS) and mechanically dissociated. The cells were strained (70 μm), RBC lysed (Life Technologies, A1049201; 2 min), and counted using a haemocytometer. Total CD8$^+$ T cells were magnet sorted (Stem Cell, 19853A) and cultured (DMEM + FBS 10%, penicillin–streptomycin 1% + nonessential amino acids (Corning, 25-025-Cl) + vitamin + β-mercaptoethanol (Gibco, 21985-023) + L-glutamine (VWR, 02-0131) + sodium pyruvate (Corning, 25-000-Cl)). Cell purity was systematically confirmed after magnet sorting and the numbers of CD8$^+$CD62L$^{hi}$ were immunophenotyped by flow cytometry.

To generate cytotoxic T lymphocytes, $2 \times 10^5$ CD8$^+$ T cells were seeded and stimulated for 48 h under $T_{c1}$ inflammatory conditions (2 μg ml$^{-1}$ plate bounded anti-CD3 and anti-CD28 (Bio X Cell, BE00011, BE00151) + 10 ng ml$^{-1}$ rIL-12 (BioLegend, 577008) + 10 μg ml$^{-1}$ of anti-IL-4 (Bio X Cell, BE0045).

## In vitro stimulation of cytotoxic CD8$^+$ T cells with neuron-conditioned medium
Naive or ablated DRG neurons were cultured (48 h) in Neurobasal-A medium supplemented with 0.05 ng μl$^{-1}$ NGF (Life Technologies, 13257-019) and 0.002 ng μl$^{-1}$ GDNF (PeproTech, 450-51-10). After 48 h, the neurobasal medium was removed, neurons were washed with PBS and 200 μl per well of T cell medium supplemented with 1 μl ml$^{-1}$ peptidase inhibitor (Sigma, P1860) and, in certain cases, capsaicin (1 μM) or KCl (50 mM) was added to DRG neurons. The conditioned medium or vehicle were collected after 30 min and added to $T_{c1}$ CD8$^+$ T cells for another 96 h. The expression of exhaustion markers (PD-1, LAG3 and TIM3) and cytokines (IFNγ, TNF and IL-2) by CD8$^+$ T cells was analysed by flow cytometry using an LSRFortessa or a FACSCanto II (Becton Dickinson). Cytokine expression levels were analysed after in vitro stimulation (PMA–ionomycin; see 'Intracellular cytokine staining').

## In vitro stimulation of cytotoxic CD8$^+$ T cells with CGRP
CD8$^+$ T cells were isolated and stimulated under $T_{c1}$ conditions in a 96-well plate. After 48 h, cells were treated with either CGRP (0.1 μM) or PBS in the presence of peptidase inhibitor (1 μM) for another 96 h. The expression of PD-1, LAG3 and TIM3, as well as IFNγ, TNF and IL-2, was immunophenotyped by flow cytometry using an LSRFortessa or a FACSCanto II (Becton Dickinson). Cytokine expression levels were analysed after in vitro stimulation (PMA–ionomycin; see 'Intracellular cytokine staining').

## In vitro silencing of DRG neurons with BoNT/A
Naive DRG neurons ($2 \times 10^4$) were seeded in a 96-well plate with neurobasal medium supplemented with NGF and GDNF. Neurons were

pre-treated with 50 pg ml$^{-1}$ of BoNT/A for 24 h. After 24 h, the culture medium was removed, neurons were washed with PBS and 200 µl per well of T cell medium supplemented with 1 µl ml$^{-1}$ peptidase inhibitor, and KCl (50 mM) was added to DRG neurons. The conditioned medium or vehicle were collected after 30 min and added to $T_{c1}$ CD8$^+$ T cells for another 96 h.

### In vitro RAMP1 blockade

CD8$^+$ T cells were treated with CGRP$_{8-37}$ (Tocris, 1169) 6 h before being exposed to the neuron-conditioned medium. In other instances, the neuron-conditioned medium was incubated for 1 h with 2 µg ml$^{-1}$ of CGRP$_{8-37}$ before being added to the CD8$^+$ T cells.

### Co-culture of CD8$^+$ T cells and DRG neurons

Naive DRG neurons ($2 \times 10^4$) were seeded in a 96-well-plate with T cell medium (supplemented with 0.05 ng µl$^{-1}$ NGF (Life Technologies, 13257-019) and 0.002 ng µl$^{-1}$ GDNF (PeproTech, 450-51-10)). One day after, $T_{c1}$ CD8$^+$ cells ($1 \times 10^5$) were added to the neurons in the presence of IL-2 (BioLegend, 575408). In some instances, co-cultures were stimulated with either capsaicin (1 µM) or KCl (50 mM). After 96 h, the cells were collected by centrifugation (5 min at 1,300 rpm), stained and immunophenotyped by flow cytometry using an LSRFortessa or a FACSCanto II (Becton Dickinson). Cytokine expression levels were analysed after in vitro stimulation (PMA–ionomycin; see 'Intracellular cytokine staining').

### RNA sequencing of triple co-cultures and data processing

A total of $1 \times 10^4$ naive *Trpv1$^{cre}$::CheRiff-eGFP$^{fl/WT}$* DRG neurons were co-cultured with $1 \times 10^5$ B16F10-mCherry-OVA overnight in T cell medium (supplemented with 0.05 ng µl$^{-1}$ NGF (Life Technologies, 13257-019), 0.002 ng µl$^{-1}$ GDNF (PeproTech, 450-51-10). One day after, $4 \times 10^5$ stimulated OVA-specific CD8$^+$ T cells under $T_{c1}$ conditions were added to the co-culture. After 48 h, the cells were detached and TRPV1 neurons (CD45$^-$eGFP$^+$mCherry$^-$), B16F10-mCherry-OVA (CD45$^-$eGFP$^-$mCherry$^+$) and OVA-specific CD8$^+$ T cells (eGFP$^-$mCherry$^-$CD45$^+$CD3$^+$CD8$^+$) were FACS purified using a FACSAria IIu cell sorter (Becton Dickinson), and the cell supernatant was collected for ELISAs.

RNA-sequencing libraries were constructed using the Illumina TruSeq Stranded RNA LT Kit (Illumina) following the manufacturer's instructions. Illumina sequencing was performed at Fulgent Genetics. Reads were aligned to the Mouse mm10 (GenBank assembly accession GCA_000001635.2) reference genome using STAR v.2.7 (ref. [58]). Aligned reads were assigned to genic regions using the featureCounts function from subread v.1.6.4 (ref. [65]). Gene expression levels were represented by TPM. Hierarchical clustering was computed using the heatmap.2 function (ward.D2 method) from the gplots R package (v.3.1.3). Differential gene expression analysis was performed using DeSeq2 v.1.28.1 (ref. [59]). These data have been deposited in the NCBI's GEO (ref. [60]) (GSE205864).

### RNA sequencing of cancer and neuron co-cultures and data processing

A total of $1 \times 10^4$ naive *Trpv1$^{cre}$::CheRiff-eGFP$^{fl/WT}$* DRG neurons were co-cultured with $5 \times 10^4$ B16F10-mCherry-OVA cells overnight in complete DMEM (Corning, 10-013-CV) supplemented with 10% FBS (Seradigm, 3100), 1% penicillin–streptomycin (Corning, MT-3001-Cl), 0.05 ng µl$^{-1}$ NGF (Life Technologies, 13257-019), 0.002 ng µl$^{-1}$ GDNF (PeproTech, 450-51-10). After 48 h, the cells were detached and TRPV1 neurons (eGFP$^+$mCherry$^-$) and B16F10-mCherry-OVA (eGFP$^-$mCherry$^+$) were FACS purified using a FACSAria IIu cell sorter (Becton Dickinson), and the cell supernatant was collected for ELISAs.

RNA-sequencing libraries were constructed using the Illumina TruSeq Stranded RNA LT Kit (Illumina) following the manufacturer's instructions. Illumina sequencing was performed at Fulgent Genetics. Reads were aligned to the mouse mm10 reference genome (GenBank assembly accession GCA_000001635.2) using STAR v.2.7 (ref. [58]). Aligned reads

were assigned to genic regions using the featureCounts function from subread v.1.6.4 (ref. [65]). Gene expression levels were represented by TPM. Hierarchical clustering was computed using the heatmap.2 function (ward.D2 method) from the gplots R package (v.3.1.3). Differential gene expression analysis was performed using DeSeq2 v.1.28.1[59]. These data have been deposited in the NCBI's GEO (ref. [60]) (GSE205865).

### ELISA on co-cultures of B16F10 cells and DRG neurons

A total of $1 \times 10^4$ naive DRG neurons were cultured (96 h) with and without $5 \times 10^4$ B16F10 cells in complete DMEM (10% FBS, 1% penicillin–streptomycin, 1 µl ml$^{-1}$ protease inhibitor). The cells were then challenged (30 min) with sterile PBS or KCl (40 mM) and the supernatant was collected. Neuropeptide releases were measured using commercial ELISAs for VIP (Antibodies Online, ABIN6974414), SP (Cayman Chemical, 583751) and CGRP (Cayman Chemical, 589001).

### ELISA on co-cultures of B16F10 cells, CD8$^+$ T cells and DRG neurons

Levels of SLPI (R&D Systems, DY1735-05) were measured in the cells' supernatant using a commercial ELISA.

### OT-I CD8$^+$ T cell-induced B16F10 elimination

A total of $2 \times 10^4$ naive *Trpv1$^{cre}$::CheRiff-eGFP$^{fl/WT}$* DRG neurons were co-cultured with $1 \times 10^5$ B16F10-mCherry-OVA cells overnight in T cell medium (supplemented with 0.05 ng µl$^{-1}$ NGF (Life Technologies, 13257-019) and 0.002 ng µl$^{-1}$ GDNF (PeproTech, 450-51-10)). One day after, $4 \times 10^5$ stimulated OVA-specific CD8$^+$ T cells under $T_{c1}$ conditions were added to the co-culture. After 48 h, the cells were detached by trypsin (Gibco, 2062476) and collected by centrifugation (5 min at 1,300 rpm), stained using anti-Annexin V, 7-AAD (BioLegend, 640930) and anti-CD8 for 20 min at 4 °C, and immunophenotyped by flow cytometry using a FACSCanto II (Becton Dickinson). Cytokine expression levels were analysed after in vitro stimulation (PMA/ionomycin; see 'Intracellular cytokine staining').

### Effect of neuron-conditioned medium on OT-I CD8$^+$ T cell-induced B16F10 elimination

A total of $4 \times 10^5$ stimulated OVA-specific CD8$^+$ T cells were added to $1 \times 10^5$ B16F10-mCherry-OVA and treated with fresh condition medium (1:2 dilution). After 48 h, cells were stained using anti-Annexin V, 7-AAD (BioLegend, 640930) and anti-CD8 for 20 min at 4 °C, and were immunophenotyped by flow cytometry using an LSRFortessa or a FACSCanto II (Becton Dickinson). For CGRP, $4 \times 10^5$ stimulated OVA-specific CD8$^+$ T cells were added to $1 \times 10^5$ B16F10- mCherry-OVA and treated with CGRP (100 nM). After 24 h, the cells were stained using anti-Annexin V, 7-AAD (BioLegend, 640930) and anti-CD8 for 20 min at 4 °C, and were immunophenotyped by flow cytometry using an LSRFortessa or a FACSCanto II (Becton Dickinson). Cytokine expression levels were analysed after in vitro stimulation (PMA–ionomycin; see 'Intracellular cytokine staining').

### Survival of B16F10 cells

A total of $1 \times 10^5$ B16F10 cells were cultured in six-well plates and challenged with BoNT/A (0–50 pg µl$^{-1}$) for 24 h, QX-314 (0–1%) for 72 h, BIBN4096 (1–8 µM) for 24 h or their vehicle. The survival of B16F10 cells was assessed using anti-Annexin V staining and measured by flow cytometry using an LSRFortessa or a FACSCanto II (Becton Dickinson), or counted using a haemocytometer.

### Effect of drugs on the function of CD8$^+$ T cells

Splenocyte-isolated CD8$^+$ T cells from naive C57BL6J mice were cultured under $T_{c1}$-stimulating conditions (ex-vivo-activated by CD3 and CD28, IL-12 and anti-IL4) in 24-well plates for 48 h. The cells were then exposed to QX-314 (50–150 µM), BoNT/A (10–50 pg µl$^{-1}$) or BIBN4096 (1–4 µM) for 24 h. Apoptosis, exhaustion and activation levels were measured by flow cytometry using an LSRFortessa or a FACSCanto II (Becton Dickinson).

## In silico analysis of neuronal expression profiles using RNA-sequencing and microarray datasets

Publicly available RNA gene expression data from seven datasets were downloaded from the NCBI GEO portal[66]. RNA gene expression values of genes of interest were extracted. Expression values from single-cell sequencing were averaged for all cells. To be able to compare expression from datasets that were generated using different techniques (single-cell RNA sequencing, bulk RNA sequencing and microarrays) and normalization methods (TPM, RPKM (reads per kilobase per million mapped reads), RMA (robust multiarray analysis) and UMI (unique molecular identifiers)), all genes of interest were ratioed over *TRPV1* expression, then multiplied by 100, and the $\log_{10}$ values of these values were plotted as a heat map[66]. Kupari et al.[67] used single-cell RNA sequencing of JNC neurons, whereas Usoskin et al.[68] and Li et al.[69]. used single-cell RNA sequencing of lumbar neurons. Chiu et al.[70] measured gene expression by microarrays of whole and FACS-sorted $Na_V1.8^+$ lumbar neurons. Goswami et al.[71] performed RNA sequencing of TRPV1[+] lumbar neurons, whereas Ray et al.[72] performed RNA sequencing of human lumbar neurons.

## In silico analysis of tumour expression profiles of patients with melanoma using single-cell RNA sequencing

Using the publicly available Broad Institute single-cell bioportal, we performed an in silico analysis of single-cell RNA sequencing of human melanoma biopsies. We assessed the gene profile of *RAMP1*-expressing and *RAMP1*-negative T cells in the tumours of patients with metastatic melanoma[42]. Similarly, we assessed the genetic program of *RAMP1*-expressing and *RAMP1*-negative CD8[+] T cells in patients with melanoma[41]. The latter dataset was also used to analyse the genetic profile of CD8[+] T cells in patients who were responsive to immune checkpoint blockers or unresponsive to such treatment, as well as the genetic profile of malignant melanoma cells (defined as CD90[−]CD45[−]) from the biopsies of ten different patients[41]. Individual cell data are shown as a $\log_2$-transformed $1+(TPM/10)$. Experimental details and cell clustering have been defined in previous studies[41,42].

## In silico analysis of the expression profiles of human immune cells

Publicly available RNA gene expression data from a previous study[73] were downloaded from the NCBI GEO portal. Read counts normalized to transcripts per million protein-coding genes (pTPM) values for genes of interest were extracted. Expression values from single-cell sequencing were averaged for all cells. Experimental details and cell clustering have been defined before[73].

## In silico analysis of the expression profiles of cultured B16F10 cells

Publicly available RNA gene expression data from a previous study[74] were downloaded from the NCBI GEO portal. Read counts normalized to TPM for genes of interest were extracted. Experimental details and cell clustering have been defined before[74].

## In silico analysis of the expression profiles of mouse immune cells using the ImmGen database

Using the publicly available ImmGen database, we performed an in silico analysis of RNA-sequencing data (DESeq2 data) of various mouse immune cells. As per ImmGen protocol, RNA-sequencing reads were aligned to the mouse genome GENCODE GRCm38/mm10 primary assembly (GenBank assembly accession GCA_000001635.2) and gene annotations vM16 with STAR 2.5.4a. The ribosomal RNA gene annotations were removed from the general transfer format file. The gene-level quantification was calculated by featureCounts. Raw read count tables were normalized by the median of ratios method with the DESeq2 package from Bioconductor and then converted to GCT and CLS format. Samples with fewer than 1 million uniquely mapped reads were automatically excluded from normalization. Experimental details

can be found at https://www.immgen.org/Protocols/ImmGenULI_RNAseq_methods.pdf.

## Oncomine

In silico analysis of the expression profiles of biopsies from patients with melanoma using bulk microarray sequencing. As described previously[19], samples from 45 cutaneous melanomas and 18 benign melanocytic skin nevus biopsies (around 5–20 μm) were collected and amplified, and their transcriptomes were profiled using Affymetrix U133A microarrays. Data were downloaded from the Oncomine database (https://www.oncomine.com/) as $\log_2$-transformed (median centred intensity) and genes of interest were shown as heat maps. Experimental details and cell clustering have been defined before[19].

## Survival analysis of patients with melanoma

OncoLnc (http://www.oncolnc.org/) contains survival data for 8,647 patients from 21 cancer studies performed by TCGA[40]. Using OncoLnc, we assessed the transcript expression of a user-defined list of 333 neuronal-enriched genes (neuronal membrane proteins, neural stem cell markers, transcription factors, ion channel receptors and neuropeptides) in 459 skin cancer (SKCM) tumour biopsies from the TCGA database. Of these genes, 206 were expressed, and 108 were selected on the basis of their negative Cox coefficient value, indicating a link between lower gene expression and improved patient survival. Kaplan–Meier curves show the survival of the patients after segregation into two groups defined by their low or high expression of a gene of interest. Details of patients can be found in TCGA[40] and computational analyses can be found at https://doi.org/10.7717/peerj-cs.67.

## Data collection and analysis

GraphPad Prism (v.9.0) and Microsoft Excel (v.2019) were used for data entry, graph construction and data analysis. Image analysis (neurite length and ramification index) was performed using ImageJ macros (Fiji, v.1.53c). Flow cytometry data were analysed using FlowJo (v.10.0.0). Calcium microscopy analysis was performed using a Nikon Eclipse Ti2 microscope (NIS-Elements Advanced Research v.4.5). Patient biopsy images were collected using an Olympus BX51 bright-field microscope and mouse tumour innervation images were acquired using an Olympus FV3000 confocal imaging system. For RNA sequencing, the reads were aligned to the mouse reference genome GRCm38/mm10 (GenBank assembly accession GCA_000001635.2) using STAR (versions used: 2.5.4a, 2.5.1b and 2.7). Aligned reads were assigned to genic regions using the featureCounts function from Subread (v.1.6.4 22). Hierarchical clustering was computed using the heatmap.2 function (ward.D2 method) from Gplots R package (v.3.1.3). Differential gene expression analysis was carried out by DeSeq2 (versions used: 1.18.1 and 1.28.1). TCGA data were accessed using Oncomine (https://www.oncomine.com/ for gene expression) and OncoLnc (http://www.oncolnc.org/ for survival). Single-cell RNA sequencing was analysed using the Broad Single-Cell Portal (https://singlecell.broadinstitute.org). Human and mouse immune cell gene profiles were respectively analysed using the Human Protein Atlas (https://www.proteinatlas.org/humanproteome/immune+cell) and Immunological Genome Project (https://www.immgen.org/).

## Sample size

Statistical methods were not used to predetermine sample size. The size of the cohort, based on similar studies in the field, was validated by pilot studies. All sample sizes are indicated in the figures and/or figure legends. All *n* values are indicated within the figure legends. In the only case in which a range is used (Fig. 2a,b), exact *n* values are provided in the source data files. For in vivo experiments, we used *n* > 8 mice. For in vitro experiments in which replicate samples were used, we repeated the experiments at least three independent times to confirm the findings. For other mouse experiments a minimum of five mice were used

to ensure that proper statistics could be used. We determined this to be sufficient as per our pilot data, use of internal controls and/or the observed variability between within experimental groups.

## Replication

The number of replicates is indicated in the figures, figure legends and/or methods. On the graphs, individual dots represent individual samples or mice used. For each experiment, all attempts at replication were successful and our findings showed comparable results.

## Randomization

Breeding pairs and their offspring (nociceptor-intact and -ablated mice) were co-housed and, in respect with the ARRIVE guidelines[75], were randomly allocated into each experimental group. For in vitro experiments, randomization was used for treatment selection. In some calcium microscopy experiments, the investigators performing the data collection were tasked to select all ligand-responsive cells for downstream analysis. In these rare cases. randomization was not used for cell selection.

## Blinding

Double blinding was used for all in vivo treatments. In calcium micros-copy experiments involving co-culture (for example, nociceptors and cancer cells), the differences in cell morphology are obvious and, therefore, the investigator performing the experiment was not blind. However, this investigator was always blinded to the treatment being applied to the cells and a second blinded investigator performed the downstream data analysis.

## Data exclusions

No data were excluded.

## Statistics

Statistical significance was determined using GraphPad Prism (Dot-matics, v.9) and calculated using simple linear regression analysis, Mantel–Cox regression, one-way or two-way ANOVA for multiple com-parisons and two-sided unpaired Student's $t$-test for single variable comparison. In calcium imaging experiments, the $P$ value is calculated on ligand-responsive neurons (calcium flux ≥ 5–10%). $P$ values < 0.05 were considered significant.

## Antibodies

All of the antibodies used in this study are also listed in Supplementary Table 2.

## Reporting summary

Further information on research design is available in the Nature Research Reporting Summary linked to this article.

## Data availability

All data are readily available online (https://www.talbotlab.com/nature) and from the corresponding author. The RNA-sequencing datasets have been deposited in the NCBI's GEO (GSE205863, GSE205864 and GSE205865). Source data are provided with this paper.

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

**Acknowledgements** S.T. is supported by the Canada Research Chair program (950-231859), the Canadian Institutes of Health Research (407016 (S.T. and M. Rangachari), 461274 and 461275), the Canadian Foundation for Innovation (37439), the Natural Sciences and Engineering Research Council of Canada (RGPIN-2019-06824), the Azrieli Foundation and the Brain Canada Foundation, as well as the Canada Cancer Research Society (840775; S.T., S.D.R., and J.T.). C.J.W. is supported by Dr. Miriam and Sheldon G. Adelson Medical Research Foundation and National Institutes of Health grant R35NS105076. P.D.V. is supported by the University of Pennsylvania Basser Center for BRCA (Innovation award), and the National Institutes of Health (National Institute of General Medical Science) grants 1U54GM128729-02 (scholar award) and 5P20GM103548-08. J.T. is supported by a Saputo Research Chair, the Canada Foundation for Innovation (30017), and the Canadian Institutes of Health Research (136802). M.B., M.A., and T.E. held a scholarship from the Fonds de Recherche Santé Quebec or CAPES/Print (88887.374124/2019-00). We thank the Molecular Pathology and Imaging cores at Sanford Research (supported by National Institutes of Health grants 5P20GM103548 and P20GM103620) for tissue processing. We thank P. B. D. Rosa for graphic design, A. C. Anderson for initial discussions and A. Regev, and L. Jerby-Arnon for discussing the in silico analysis of the single-cell RNA-sequencing datasets.

**Author contributions** M.B., M.A., T.E., S.H., C.J.S., H.D., K.-C.H., W.C.S., C.S.W., C.R.S., S.L.F., M. Rangachari, J.T., S.D.R., R.D., M. Rafei, N.G., P.D.V., C.J.W. and S.T. designed the study. A.E.T. and N.G. initiated the work on SLPI. E.S., S.C.T., H.M., A.P. and B.D. assisted with experimentation. M.B., M.A., T.E., A.A., A.M., Karine Roversi, Katiane Roversi, C.T.L., A.C.R., S.H., L.J., H.M., D.W.V. and S.T. conducted the experiments. M.B., M.A., P.D.V., C.J.W. and S.T. wrote the manuscript with input from all authors. M.B. performed the experiments in Figs. 1d,e, 2a,b,

3a,b,d,e,h, 4a,b,e and 5a–c and Extended Data Figs. 4e, 5a–e, 6a–g,k,q, 7a–f,h–n, 8a–e,l–r, 9 and 10a–f,h–w. M.A. performed the experiments in Figs. 1d,e, 2a,b, 3a–c, 4a–d and 5a and Extended Data Figs. 4a–c,e, 5a–e,h–j, 6l–n, 7g,o,p, 8g,j,k,s,t and 10e,g. T.E. performed the experiments in Fig. 2c–f and Extended Data Fig. 5f–j and did the in silico analysis for Fig. 5d,e and Extended Data Figs. 1a–d, 3 and 11. A.A. performed the experiments in Fig. 3g,h and Extended Data Figs. 6o and 9. A.M. performed the experiments in Fig. 3f. Karine Roversi performed the experiments in Fig. 3a and Extended Data Fig. 6c–g,p. Katiane Roversi performed the experiments in Fig. 3a. C.T.L. performed the experiments in Extended Data Fig. 2. A.C.R. performed the experiments in Extended Data Fig. 2. S.H. performed the experiments in Fig. 1a. L.J. did the bioinformatic analysis of the experiments in Figs. 1d,e and 2a,b and Extended Data Figs. 4e and 5a–e. D.W.V. performed the experiments in Extended Data Fig. 2. S.T. performed the experiments in Fig. 1b,c and Extended Data Figs. 4d and 8f,h,i.

**Competing interests** S.T. and C.J.W. have an equity stake in Nocion Therapeutics. S.T. and C.J.W. have deposited a provisional patent (WO 2021/173916) on the use of charged sodium channel blockers to silence nociceptor neurons as a means to safeguard host anti-tumour immunity.

**Additional information**
**Correspondence and requests for materials** should be addressed to Sebastien Talbot.

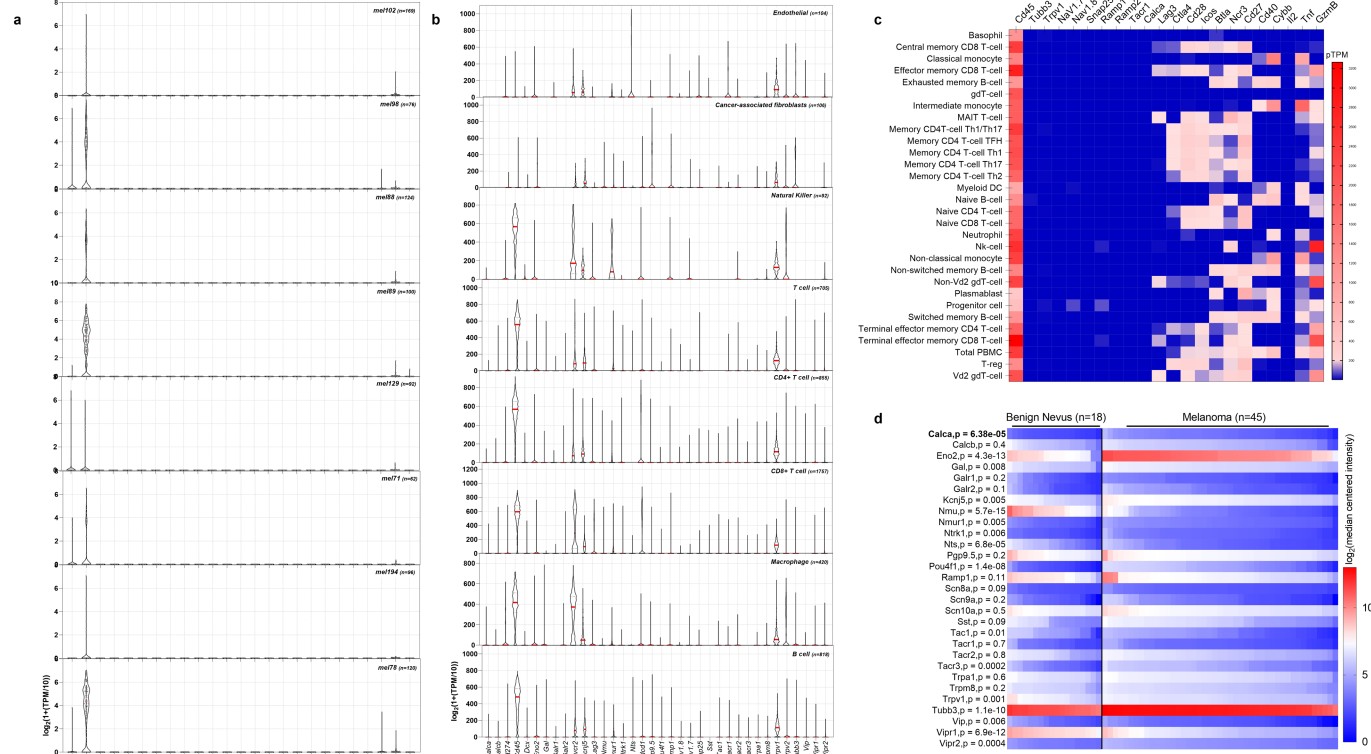

**Extended Data Fig. 1 | *TRPV1*, *NAV1.8*, *SNAP25* or *RAMP1* transcripts are expressed in patient melanoma biopsies but are not detected in human immune cells or malignant cells.** (**a**–**b**) In silico analysis of single-cell RNA sequencing of human melanoma-infiltrating cells revealed that *Trpv1*, *Nav1.8* (Scn10a), *Snap25* (the molecular target of BoNT/A), *Calca* (gene encoding for CGRP) transcripts are not detected in malignant melanoma cells (defined as CD90⁻CD45⁻) from ten different patients' biopsies (**a**) nor in cancer-associated fibroblasts, macrophage, endothelial, natural killer, T, and B cells (**b**). Individual cell data are shown as a log₂ of 1 + (transcript per million / 10). Experimental details and cell clustering were defined in Jerby-Arnon et al[41]. N are defined in the figures. (**c**) *In silico* analysis of human immune cells revealed their basal expression of *Cd45*. Using RNA sequencing approaches, *Calca*, *Snap25*, *Trpv1* or *NaV1.8* are not detected in these cells. Heat maps show the read counts normalized to transcripts per million protein-coding genes (pTPM) for each of the single-cell clusters. Experimental details and cell clustering were defined in Monaco et al[73]. (**d**) Forty-five cutaneous melanomas and 18 benign melanocytic skin nevus biopsies transcriptomes were profiled using Affymetrix U133A microarrays[19]. In silico analysis of this dataset revealed that cutaneous melanoma heightened expression levels of *Calca (1.4-fold)*, *Pouf4f1 (2-fold)*, *Eno2 (1.4-fold)*, and *Tubb3 (1.1-fold)*, as well as other neuronal-enriched genes. Heat map data are shown as log₂ (median centred intensity); two-sided unpaired Student's t-test; p-values and n are shown in the figure. Experimental details were defined in Haqq et al[19].

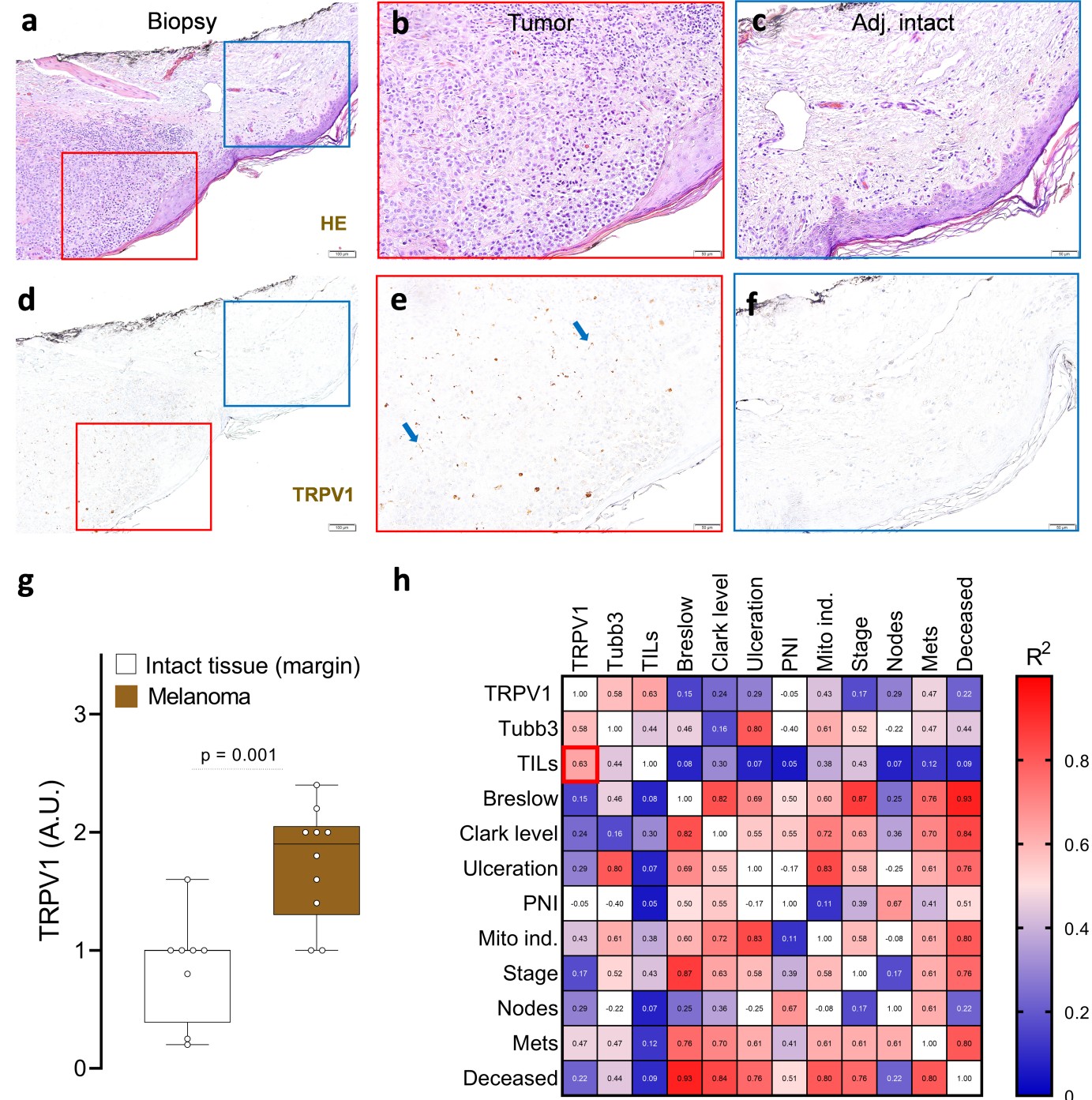

**Extended Data Fig. 2 | TRPV1⁺ neurons innervate patient melanomas.** Patients' melanoma sections were stained with hematoxylin eosin (**a**–**f**), and the presence of TRPV1 (**d**–**f**; brown) neurons was analysed by immunohistochemistry. Increased levels of TRPV1⁺ neurons (**g**) were found in the tumour (delimited by red square; **a**-**b**, **d**-**e**) compared to adjacent healthy skin (delimited by blue square; **a**,**c**,**d**,**f**). Increased TRPV1 immunolabelling in tumour sections primarily correlated with enhanced levels of tumour-infiltrating leukocytes (**h**) as scored from a retrospective correlation analysis performed on the patients' pathology reports. Data are shown as representative immunohistochemistry images (**a**–**f**), box-and-whisker plots (runs from minimal to maximal values; the box extends from 25th to 75th percentile and the middle line indicates the median), for which individual data points are given (**g**) or as a heat map (**h**) displaying Pearson's correlation ($R^2$). N are as follows: **a**–**f**: n=10, **g**: intact (n = 8), tumour (n = 10), **h**: n = 10. Slides were scored blindly by two experienced medical pathologists. P-values are shown in the figure and determined by two-sided unpaired Student's t-test (**g**). Scale = 100 µm (**a**,**d**), 50 µm (**b**,**c**,**e**,**f**).

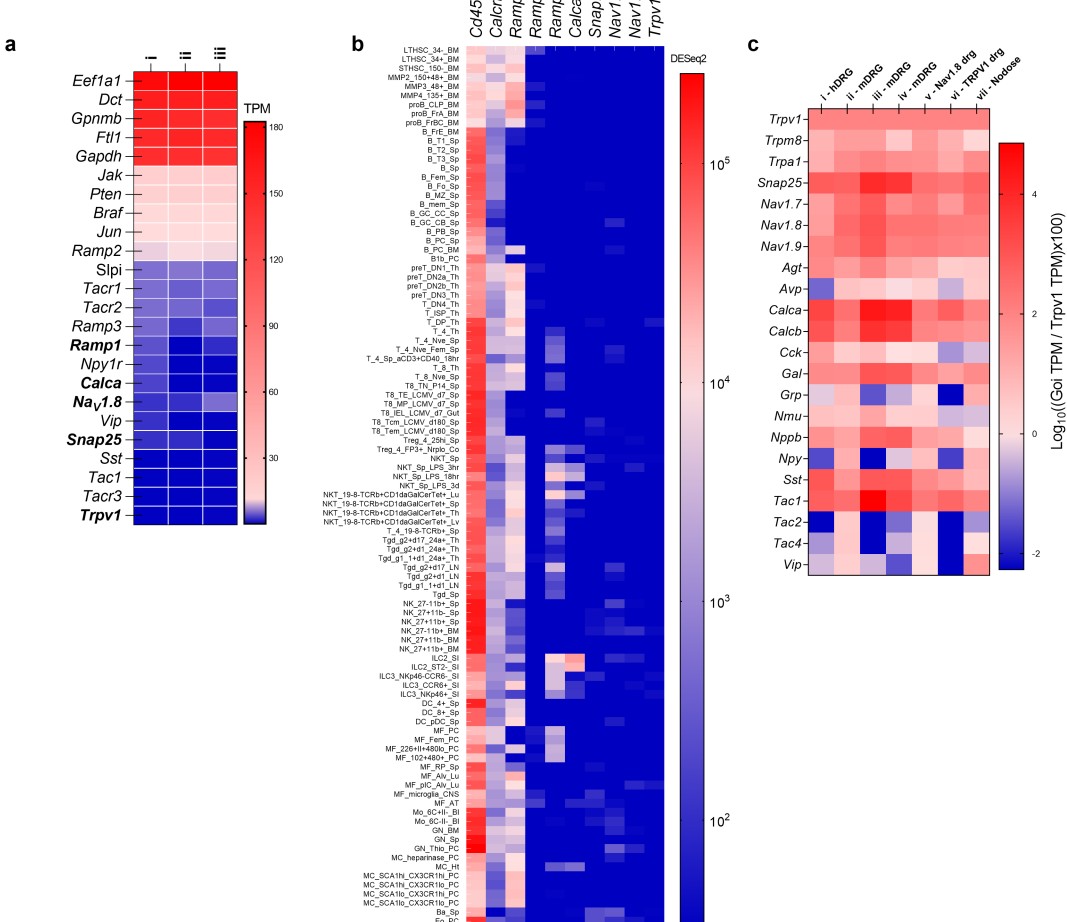

**Extended Data Fig. 3 | *Trpv1*, *Nav1.8*, *Snap25* or *Ramp1* transcripts are not detected in B16F10 cancer cells or mouse immune cells. (a)** *In silico* analysis of three different B16F10 cells cultures (labelled as i, ii, iii)[74] revealed their basal expression of *Braf* and *Pten*. In contrast, *Calca*, *Snap25*, *Trpv1* or *NaV1.8* transcripts are not detected in B16F10 cells. Heat map data are shown as transcript per million (TPM) on a linear scale. Experimental details were defined in Castle et al[74]. N=3/group. **(b)** ImmGen RNA sequencing of leukocyte subpopulations[76] reveals their basal expression of *Cd45* and *Ramp1*. In contrast, *Snap25*, *Trpv1*, or *Nav1.8* transcripts are not detected in mouse immune cells. Heat map data are shown as DESeq₂ on a logarithmic scale. **(c)** A meta-analysis

of seven published nociceptor neuron expression profiling datasets[66] revealed the basal expression of sensory neuron markers (*Trpv1*, *Trpa1*) and neuropeptides (*Sp*, *Vip*, *Nmu*, *Calca*). Expression across datasets was ratioed over Trpv1 and multiplied by 100. The log₁₀ of these values is presented as a heat map. i) RNA sequencing of human lumbar neurons;[72] ii) microarrays of mouse FACS-sorted Naᵥ1.8⁺ neurons;[70] iii) and iv) single-cell RNA sequencing of mouse lumbar neurons;[68,69] v) microarray profiling of mouse Naᵥ1.8⁺ DRG neurons;[70] vi) performed RNA sequencing of mouse TRPV1⁺ neurons;[71] and vii) single-cell RNA sequencing of mouse vagal ganglia[67].

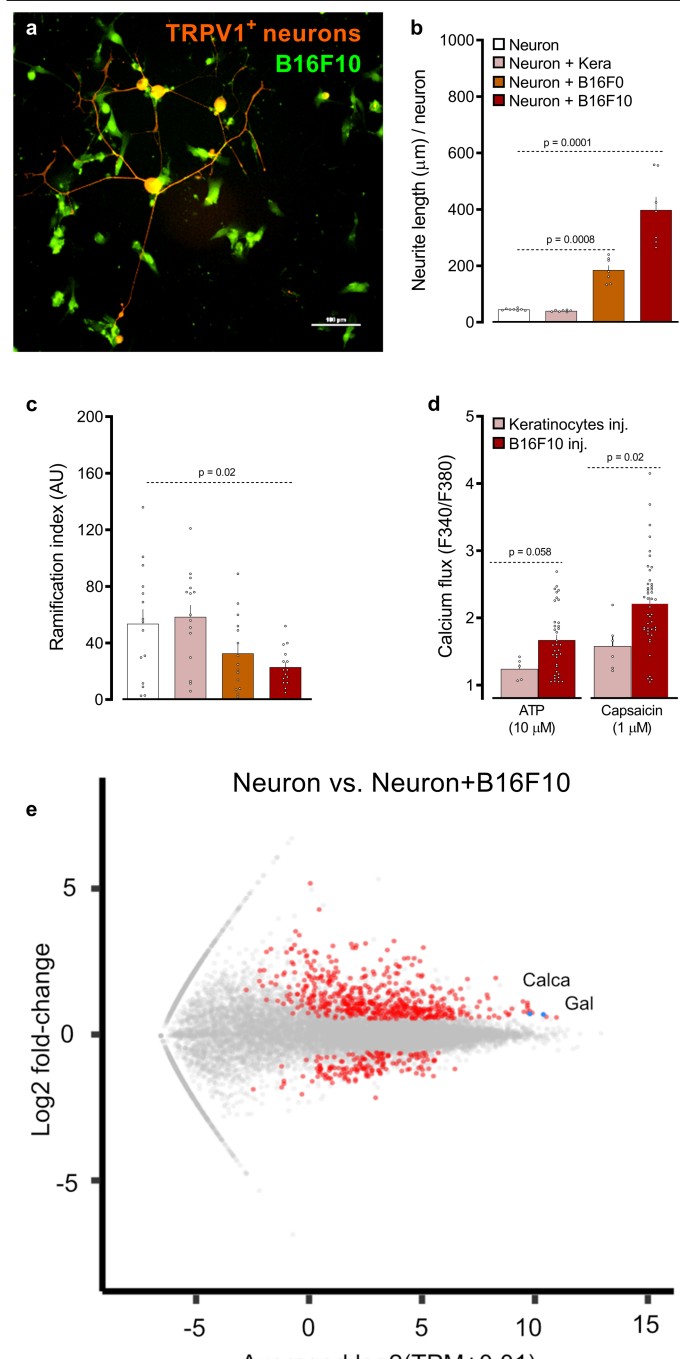

**Extended Data Fig. 4 | B16F10 cells interact with nociceptor neurons.**
(**a**–**c**) When co-cultured with B16F10-eGFP cells (green), TRPV1[+] nociceptor (*Trpv1^cre^::tdTomato^fl/WT^*; orange) neurons form neuro-neoplasic contacts (**a**), show longer neurites (**b**), and exhibit reduced arborization (**c**) than when cultured alone or with non-tumorigenic keratinocytes (**b**–**c**). (**d**) L3–L5 DRG neurons were collected from mice 2-weeks after they were inoculated (left hindpaw; i.d.) with B16F10- or non-tumorigenic keratinocytes, cultured and calcium flux to ligands tested (ATP (10 μM), and capsaicin (1 μM)). Compared to neurons from keratinocytes-injected mice, the one from tumour-bearing mice showed increased sensitivity to capsaicin. (**e**) Naive DRG neurons (*Trpv1^cre^::CheRiff-eGFP^fl/WT^*) were cultured alone or in combination with B16F10-mCherry-OVA. After 48h, the cells were collected, FACS purified, and RNA sequenced. Hierarchical clustering of sorted neuron DEG show distinct groups of transcripts enriched in TRPV1[+] neuron vs cancer-exposed TRPV1[+] neuron populations. Pairwise comparison of naive TRPV1[+] neuron vs cancer-exposed TRPV1[+] neuron populations showing differentially expressed transcripts as a volcano plot (p<0.05). Among others, *Calca* (gene encoding for CGRP) was overexpressed in TRPV1[+] (FACS-purified eGFP-expressing cells) neurons when co-cultured with B16F10-mCherry-OVA. Data are shown as representative image (**a**), mean ± S.E.M (**b**–**d**), or volcano plot (**e**). N are as follows: **a**: n = 4, **b**: neuron (n=8), neuron + keratinocytes (n = 7), neuron + B16F0 (n = 7), neuron + B16F10 (n = 7), **c**: n = 15/groups, **d**: keratinocytes inj. + ATP (n=5), B16F10 inj. + ATP (n=36), Keratinocytes inj. + caps (n = 6), B16F10 inj. + caps (n = 44), **e**: n = 4/ groups. Experiments were independently repeated two (**d**) or three (**a**–**c**) times with similar results. Sequencing experiment was not repeated (**e**). P-values are shown in the figure and determined by one-way ANOVA post-hoc Bonferroni (**b**–**c**) or two-sided unpaired Student's t-test (**d**). Scale bar = 100 μm (**a**).

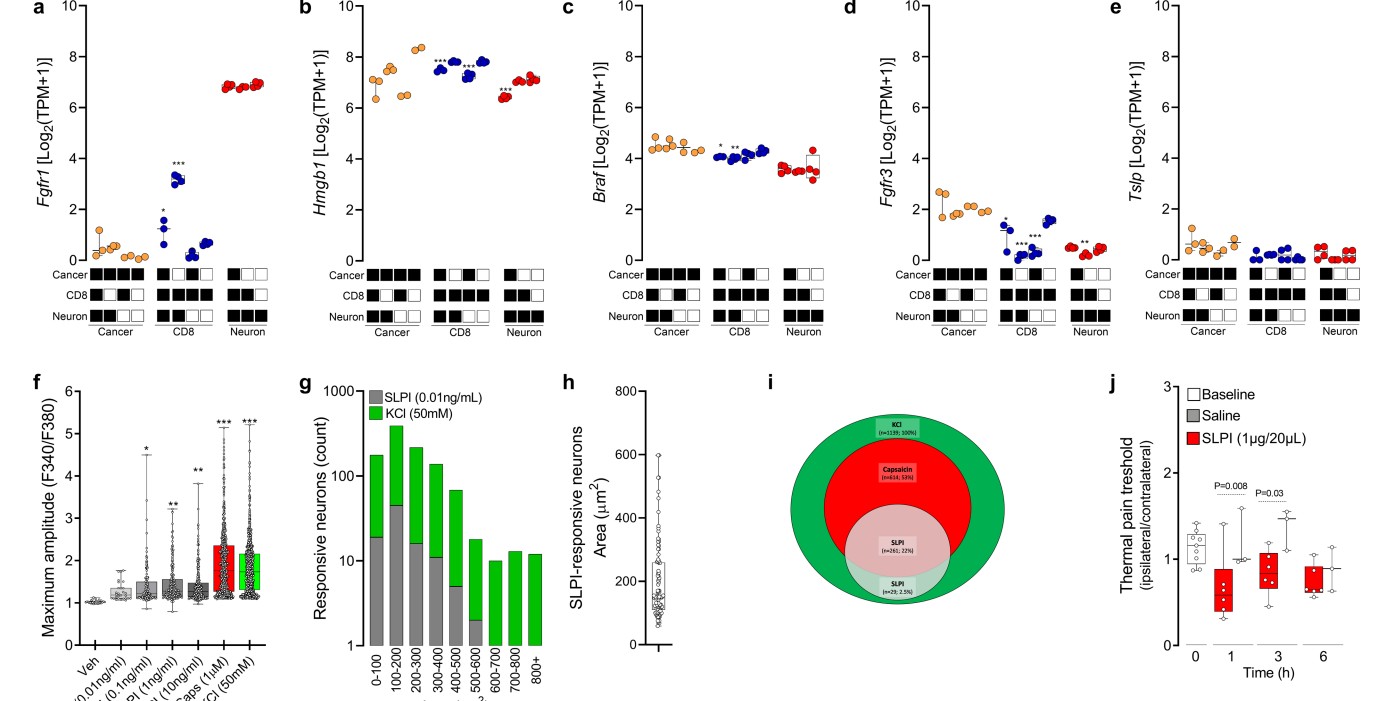

**Extended Data Fig. 5 | B16F10-secreted SLPI activates nociceptor neurons.**
(**a–e**) Naive DRG neurons (*Trpv1^cre^::CheRiff-eGFP^fl/WT^*), B16F10-mCherry-OVA, and OVA-specific cytotoxic CD8^+ T cells were cultured alone or in combination. After 48h, the cells were collected, FACS purified, and RNA sequenced. DEGs were calculated, and *Fgfr1* (fibroblast growth factor receptor 1) was found to be overexpressed in OVA-specific cytotoxic CD8^+ T cells when co-cultured with cancer cells and DRG neurons (**a**). Conversely, OVA-specific cytotoxic CD8^+ T cells downregulates the expression of the pro-nociceptive factor *Hmgb1* (High–mobility group box 1; **b**), *Braf* (**c**), as well as *Fgfr3* (**d**) when co-cultured with B16F10-mCherry-OVA and DRG neurons. *Tslp* expression level was not affected in any of tested groups (**e**). (**f–i**) Using calcium microscopy, we probed whether SLPI directly activates cultured DRG neurons. We found that SLPI (0.01-10 ng/mL) induces a significant calcium influx in DRG neurons (**f**). SLPI-responsive neurons are mostly small-sized neurons (**g-h**; mean area = 151 μm²) and largely capsaicin-responsive (**i**; ~42%). (**j**) The right hindpaw of naive mice was injected with saline (20 μL) or SLPI (i.d., 1 μg/20 μL), and the mice's noxious thermal nociceptive threshold was measured (0-6h). The ipsilateral paw injected with SLPI showed thermal hypersensitivity in contrast with the contralateral paw. Saline had no effect on the mice's thermal sensitivity. Data are shown as box-and-whisker plots (runs from minimal to maximal values; the box extends from 25^th to 75^th percentile and the middle line indicates the median), for which individual data points are given (**a–f**, **h**, **j**), stacked bar graph on a logarithmic scale (**g**), and Venn Diagram (**i**). N are as follows: **a–e**: n = 2–4/groups, **f**: vehicle (n = 28), 10pg/ml (n = 28), 100 pg/ml (n = 132), 1,000 pg/ml (n = 191), 10 ng/ml (n = 260), capsaicin (n = 613), KCl (n = 1,139), **g**: 0-100 (SLPI:19; KCl:177), 100-200 (SLPI: 45; KCl: 390), 200-300 (SLPI: 16; KCl:216), 300-400 (SLPI:11; KCl = 138), 400-500 (SLPI = 5; KCl = 68), 500-600 (SLPI=2, KCl = 18), 600-700 (SLPI = 0; KCl = 10), 700-800 (SLPI=0; KCl=13), 800+ (SLPI = 0; KCl = 12), **h**: n = 98, **i**: KCl^+=1139, KCl^+Caps^+=614, KCl^+Caps^+SLPI^+=261, KCl^+Caps^-SLPI^+=29, **j**: 0h (n = 9), SLPI at 1h (n = 6), saline at 1h (n = 3), SLPI at 3h (n = 6), saline at 3h (n=3), SLPI at 6h (n = 6), saline at 6h (n = 3). Experiments were independently repeated two (**j**) or three (**f–i**) times with similar results. Sequencing experiment was not repeated (**a–e**). P-values were determined by one-way ANOVA post-hoc Bonferroni (**a–f**); or two-sided unpaired Student's t-test (**j**). P-values are shown in the figure or indicated by * for p ≤ 0.05; ** for p ≤ 0.01; *** for p ≤ 0.001.

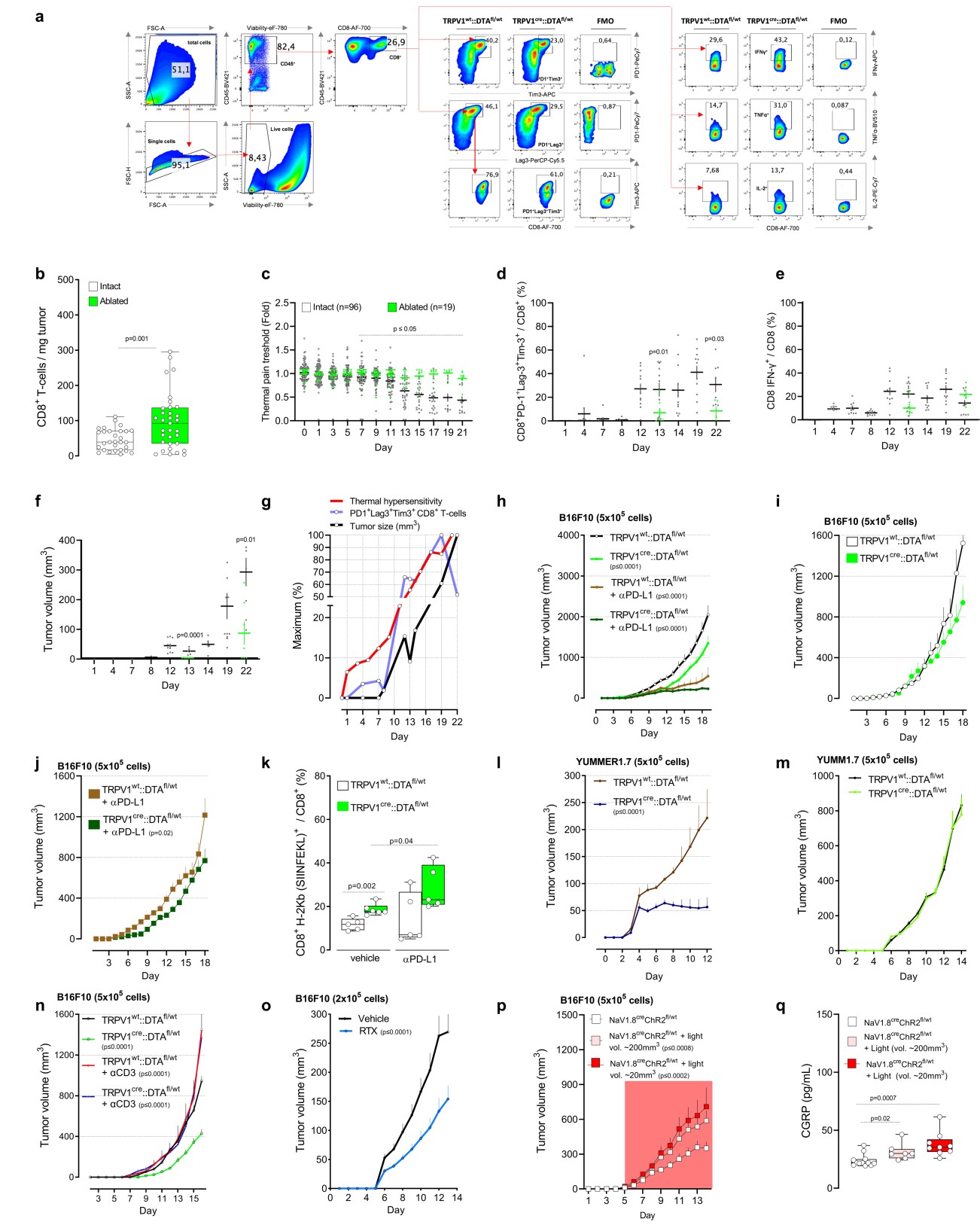

**Extended Data Fig. 6 |** See next page for caption.

**Extended Data Fig. 6 | Nociceptor ablation reduces the exhaustion of intratumoral CD8⁺ T cells.** (**a-b**) Orthotopic B16F10-mCherry-OVA (5x10⁵ cells; i.d.) cells were injected to nociceptor intact (*Trpv1^WT^::DTA^fl/WT^*) and ablated (*Trpv1^cre^::DTA^fl/WT^*) mice. Sixteen days post-B16F10-mCherry-OVA cells inoculation (5x10⁵ cells; i.d.), tumour-infiltrating CD8⁺ T cells were immunophenotyped (**a**) and were found to be more numerous in sensory neuron depleted tumours (**b**). (**c-g**) Orthotopic B16F10-mCherry-OVA (2x10⁵ cells; i.d.) cells were injected into the left hindpaw paw of nociceptor intact (n = 96; *Trpv1^WT^::DTA^fl/WT^*) or ablated (n = 18; *Trpv1^cre^::DTA^fl/WT^*) mice. When compared to their baseline threshold, littermate control mice showed significant thermal hypersensitivity on day 7, an effect that peaks on day 21 (**c**). In these mice, intratumoral frequency of PD-1⁺LAG3⁺TIM3⁺ (**d**) and IFNγ⁺ (**e**) CD8⁺ T cells increased 12 days post tumour inoculation, an effect that peaked on day 19. Finally, B16F10 tumour volume peaked on day 22 (**f**). When compared with littermate control mice, sensory neuron ablated mice inoculated with B16F10 cells showed no thermal pain hypersensitivity (**c**), reduced intratumoral frequency of PD-1⁺LAG3⁺TIM3⁺ CD8⁺ T cells (**d**) and tumour volume (**f**). In littermate control mice, thermal pain hypersensitivity (day 7) precedes the increase in intratumoral frequency of PD-1⁺LAG3⁺TIM3⁺ CD8⁺ T cells (day 12), and significant tumour growth (day 12; **g**). (**h**) Orthotopic B16F10-mCherry-OVA cells (5x10⁵ cells; i.d.) were inoculated into 8-week-old male and female sensory neuron intact or ablated mice. The mice were treated with αPD-L1 (6 mg/kg, i.p.; days 7, 10, 13, 16 post tumour inoculation) or its isotype control. On day 19, αPD-L1 potentiated the nociceptor ablation mediated reduction in B16F10-OVA tumour volume. (**i–k**) Orthotropic B16F10-mCherry-OVA cells (5x10⁵ cells, i.d.) were injected into a cohort of nociceptor neuron-ablated mice 3 days prior to the injection given to nociceptor intact mice. Mice from each group with similar tumour size (~85mm³) were selected and exposed to αPD-L1 (6 mg/kg, i.p.) once every 3 days for a total of 9 days. Eighteen days post tumour inoculation, we found that αPD-L1-reduced tumour growth was higher (~47%) in nociceptor-ablated mice than was observed in nociceptor-intact mice (~32%; **i–j**). In addition, nociceptor ablation increased the proportion of intratumoral tumour-specific (**k**; defined as H-2Kb⁺) CD8⁺ T cells. These differences were further enhanced by αPD-L1 treatment (**i–k**). (**l–m**) Sensory neurons ablation (*Trpv1^cre^::DTA^fl/WT^*) decreased growth of YUMMER1.7 cells (5x10⁵ cells; i.d.) an immunogenic version of a Braf^V600E^Cdkn2a^−/−^Pten^−/−^ melanoma cell line (**l**; assessed until day 12). The non-immunogenic YUMM1.7 cell line (5×10⁵ cells; i.d.; assessed until day 14) cells were injected to nociceptor intact (*Trpv1^WT^::DTA^fl/WT^*) and ablated mice (*Trpv1^cre^::DTA^fl/WT^*). Nociceptor ablation had no effect on YUMM1.7 growth (**m**). (**n**) Orthotopic B16F10-mCherry-OVA (5×10⁵ cells; i.d.) cells were injected to nociceptor intact (*Trpv1^WT^::DTA^fl/WT^*) and ablated mice (*Trpv1^cre^::DTA^fl/WT^*). The reduction in B16F10-mCherry-OVA (5×10⁵ cells; i.d.) tumour growth observed in nociceptors ablated mice was absent following systemic CD3 depletion (assessed until day 15; αCD3, 200 µg/mouse; i.p.; every 3 days). (**o**) To deplete their nociceptor neurons, C57BL6J mice were injected with RTX (s.c., 30, 70, 100 µg/kg) and were subsequently (28 days later) inoculated with B16F10-mCherry-OVA (2×10⁵ cells). RTX-injected mice showed reduced tumour growth when compared to vehicle-exposed mice (assessed until day 13). (**p–q**) Orthotopic B16F10-mCherry-OVA (5×10⁵ cells; i.d.) cells were injected to light-sensitive mice (*Nav1.8^cre^::ChR2^fl/WT^*). As opposed to unstimulated mice, the optogenetic activation (3.5 ms, 10Hz, 478nm, 60 mW, giving approx. 2-6 mW/mm² with a 0.39-NA fibre placed 5–10 mm from the skin, 20 min) of tumour-innervating nociceptor neurons, when started once B16F10 tumours were visible (~20 mm³) or well established (~200 mm³), resulted in enhanced tumour growth (**p**, as measured until day 14) and intratumoral CGRP release (**q**). Data are shown as FACS plot (**a**; depict the gating strategy used in fig. 3d,e), as box-and-whisker plots (runs from minimal to maximal values; the box extends from 25^th^ to 75^th^ percentile and the middle line indicates the median) for which individual data points are given (**b,k,q**), scatter dot plot (**c–f**), percentage change from maximal thermal hypersensitivity, intratumoral frequency of PD-1⁺LAG3⁺TIM3⁺ CD8⁺ T cells and tumour volume (**g**), or mean ± S.E.M (**h–j, l–p**). N are as follows: **a–b**: intact (n = 29), ablated (n = 33), **c**: intact (n = 96), ablated (n = 19), **d**: intact (n = 92), ablated (n = 15), **e**: intact (n = 96), ablated (n = 15), **f**: intact (n = 96), ablated (n = 16), **g**: n=96, **h**: intact (n = 9), ablated (n = 10), intact+αPD-L1 (n = 9), ablated+αPD-L1 (n = 8), **i**: intact (n = 14), ablated (n = 4), **j**: intact+αPD-L1 (n = 12), ablated+αPD-L1 (n = 12), **k**: intact (n = 5), ablated (n = 6), intact+αPD-L1 (n = 5), ablated+αPD-L1 (n = 5), **l**: intact (n = 8), ablated (n = 11), **m**: intact (n = 6), ablated (n = 13), **n**: intact (n = 5), ablated (n = 5), intact+αCD3 (n = 6), ablated+αCD3 (n = 5), **o**: vehicle (n = 11), RTX (n = 10), **p**: *Nav1.8^cre^::ChR2^fl/WT^* (n = 12), *Nav1.8^cre^::ChR2^fl/WT^*+Light (vol. ~200 mm³) (n = 8), *Nav1.8^cre^::ChR2^fl/WT^*+Light (vol. ~20 mm³) (n = 8), **q**: *Nav1.8^cre^::ChR2^fl/WT^* (n = 12), *Nav1.8^cre^::ChR2^fl/WT^*+Light (vol. ~200 mm³) (n = 7), *Nav1.8^cre^::ChR2^fl/WT^*+Light (vol. ~20 mm³) (n = 9). Experiments were independently repeated two (**c–g**), three (**h–q**) or six (**a,b**) times with similar results. P-values are shown in the figure and determined by two-sided unpaired Student's t-test (**b–f, k,q**), or two-way ANOVA post-hoc Bonferroni (**h–j, l–p**).

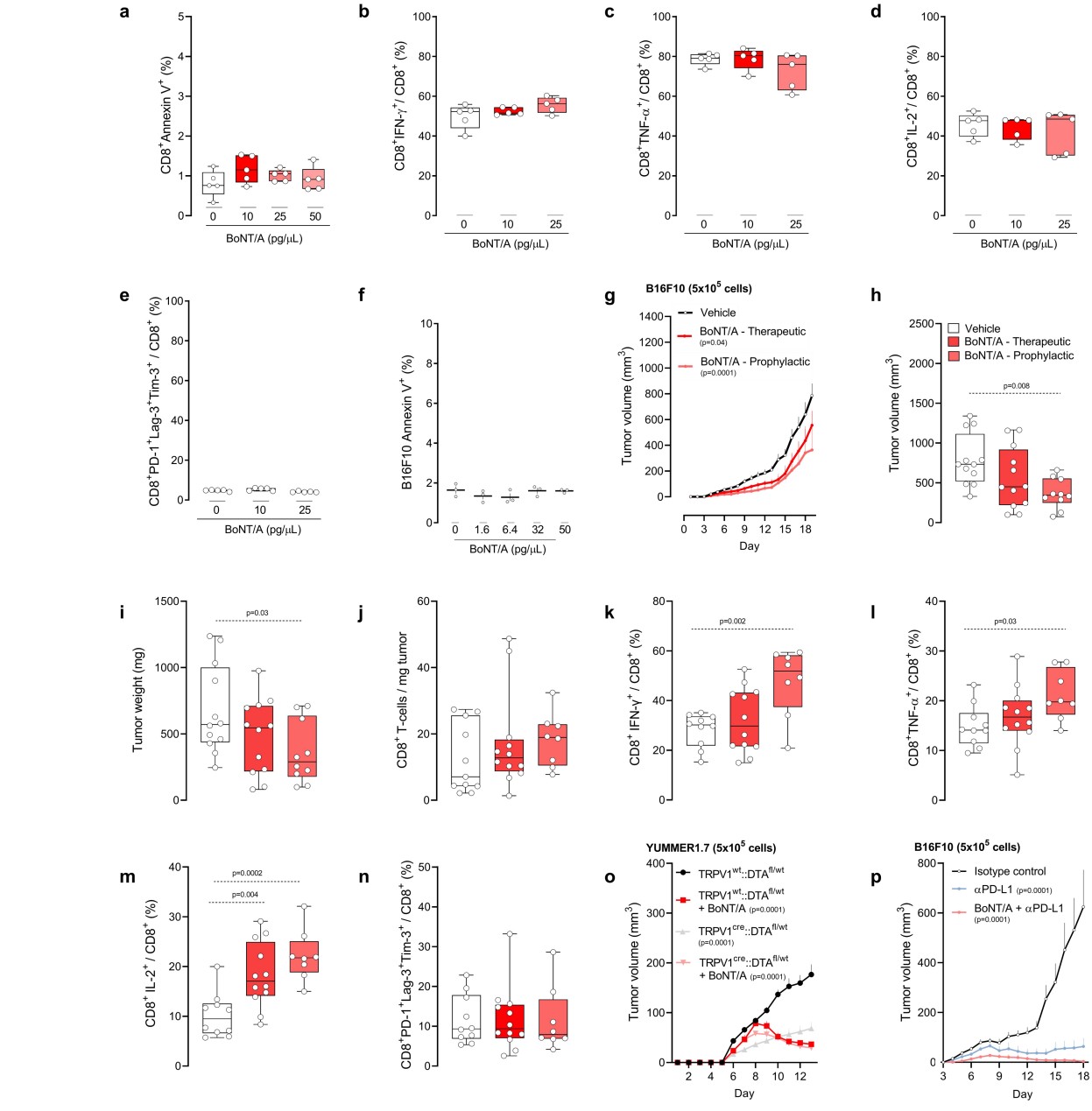

**Extended Data Fig. 7** | See next page for caption.

**Extended Data Fig. 7 | BoNT/A silencing of B16F10-innervating neurons decreases tumour growth.** (**a**–**e**) Splenocytes-isolated CD8$^+$ T cells from naive C57BL6J mice were cultured under $T_{c1}$-stimulating conditions (*ex vivo* activated by CD3 and CD28, IL-12, and anti-IL4) for 48h. The cells were then exposed to BoNT/A (10–50 pg/μL) for 24h; effects on apoptosis, exhaustion, and activation were measured by flow cytometry. When compared to vehicle-exposed cells, BoNT/A did not affect the survival (**a**) of cultured cytotoxic CD8$^+$ T cells, nor their relative expression of IFNγ$^+$ (**b**), TNF$^+$ (**c**), IL-2$^+$ (**d**) and PD-1$^+$LAG3$^+$TIM3$^+$ (**e**). (**f**) B16F10 (1x10$^5$ cells) were cultured for 24h and subsequently exposed to BoNT/A (1.6-50 pg/μL) or its vehicle for an additional 24h. BoNT/A did not trigger B16F10 cells apoptosis, as measured by the mean fluorescence intensity of Annexin V. (**g**–**n**) One and three days prior to tumour inoculation (*defined as prophylactic*), the skin of 8-week-old male and female mice was injected with BoNT/A (25 pg/μL; i.d.) or its vehicle. One day after the last injection, orthotopic B16F10-mCherry-OVA (5x10$^5$ cells; i.d.) were inoculated into the area pre-exposed to BoNT/A. In another group of mice, BoNT/A was administered (25 pg/μL; i.d.) one and three days after the tumour reached a volume of ~200mm3 (*defined as therapeutic*). The effect of neuron silencing on tumour size and tumour-infiltrating CD8 T cell exhaustion was measured. Nineteen days post tumour inoculation, we found that the tumour volume (**g**,**h**) and weight (**i**) were reduced in mice treated with BoNT/A (*Prophylactic group*). In parallel, we found that silencing tumour-innervating neurons increased the proportion of IFNγ$^+$ (**k**), TNF$^+$ (**l**), and IL-2$^+$ (**m**) CD8$^+$ T cells. BoNT/A had no effect on the total number of intratumoral CD8 T cells (**j**) or the relative proportion of PD-1$^+$LAG3$^+$TIM3$^+$ (**n**) CD8$^+$ T cells. (**o**) One and three days prior to tumour inoculation, the skin of 8-week-old male and female sensory neuron-intact or ablated mice was injected with BoNT/A (25 pg/μL; i.d.) or its vehicle. One day following the last injection, orthotopic YUMMER1.7 cells (5×10$^5$ cells; i.d.) were inoculated into the area pre-exposed to BoNT/A. The effects of nociceptor neuron ablation on tumour size and volume were measured. Thirteen days post tumour inoculation, we found that the tumour growth was lower in mice treated with BoNT/A or in sensory neuron-ablated mice. BoNT/A had no additive effects when administered to sensory neuron-ablated mice. (**p**) One and three days prior to tumour inoculation, the skin of 8-week-old male and female mice was injected with BoNT/A (25 pg/μL; i.d.) or its vehicle. One day following the last injection, orthotopic B16F10-mCherry-OVA cells (5×10$^5$ cells; i.d.) were inoculated into the area pre-exposed to BoNT/A. On days 7, 10, 13 and 16 post tumour inoculation, the mice were exposed to αPD-L1 (6 mg/kg, i.p.) or its isotype control. Eighteen days post tumour inoculation, we found that neuron silencing using BoNT/A potentiated αPD-L1-mediated tumour reduction. Data are shown as box-and-whisker plots (runs from minimal to maximal values; the box extends from 25$^{th}$ to 75$^{th}$ percentile and the middle line indicates the median), for which individual data points are given (**a**–**f**; **h**–**n**) or as mean ± S.E.M (**g**,**o**,**p**). N are as follows: **a-e**: n = 5/groups, **f**: n = 3/groups, **g**–**i**: vehicle (n = 12), BoNT/A therapeutic (n = 12), BoNT/A prophylactic (n = 10), **j**: vehicle (n = 11), BoNT/A therapeutic (n = 12), BoNT/A prophylactic (n = 8), **k**–**n**: vehicle (n = 10), BoNT/A therapeutic (n = 12), BoNT/A prophylactic (n = 8), **o**: intact + vehicle (n = 9), ablated + vehicle (n = 8), intact + BoNT/A (n = 10), ablated + BoNT/A (n = 8), **p**: vehicle (n = 7), αPD-L1 (n = 8), αPD-L1 + BoNT/A (n = 7). Experiments were independently repeated two (**a**–**f**, **o**–**p**) or four (**g**–**n**) times with similar results. P-values are shown in the figure and determined by one-way ANOVA posthoc Bonferonni (**a**–**f**, **h**–**n**) or two-way ANOVA post-hoc Bonferroni (**g**,**o**,**p**).

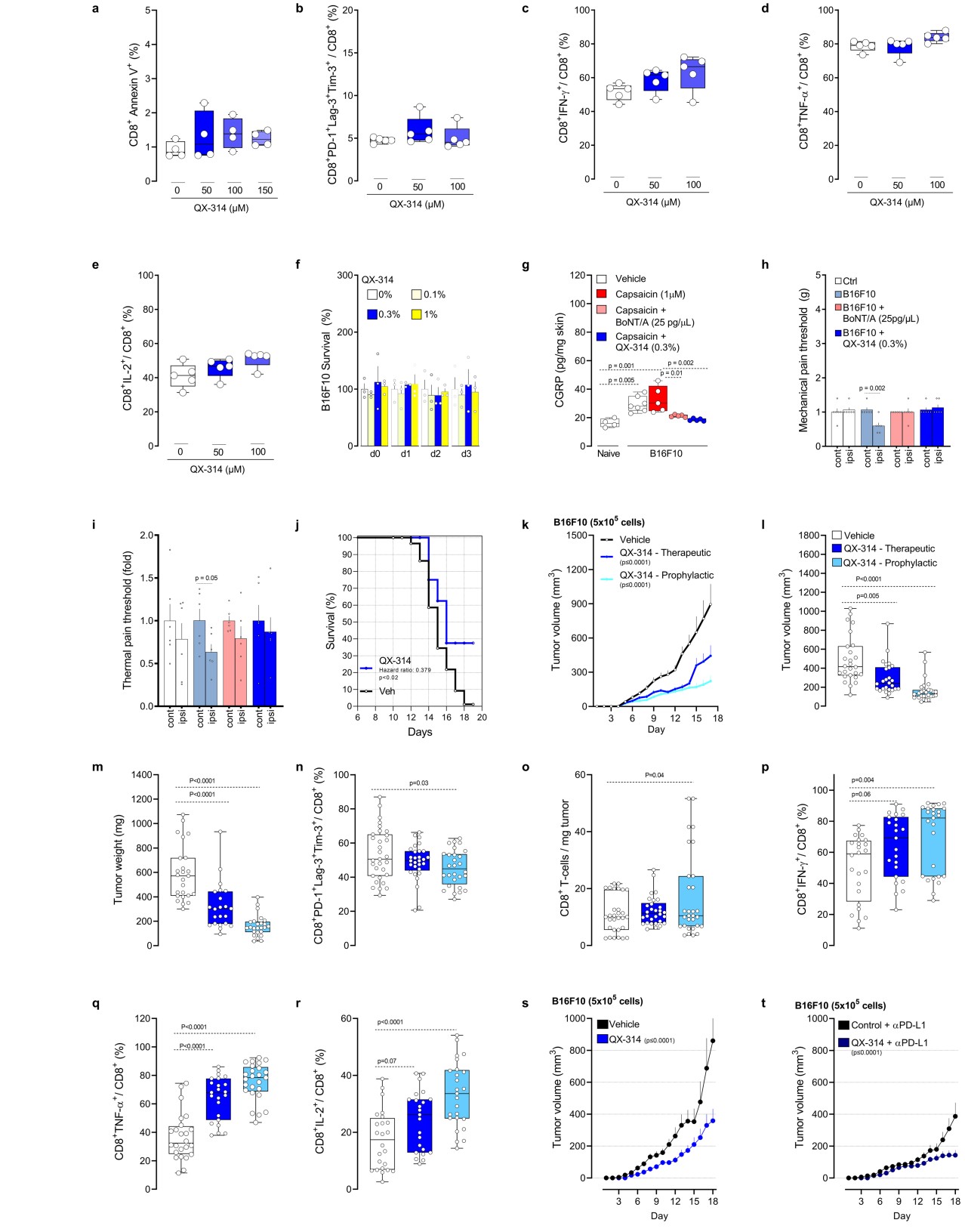

**Extended Data Fig. 8** | See next page for caption.

**Extended Data Fig. 8 | QX-314 silencing of B16F10-innervating neurons reduces tumour growth.** (**a**–**e**) Splenocytes-isolated CD8$^+$ T cells from naive C57BL6J mice were cultured under $T_{ct}$-stimulating conditions (*ex vivo* activated by CD3 and CD28, IL-12, and anti-IL4) for 48h. The cells were then exposed to QX-314 (50–150 μM) for 24h, effects on apoptosis, exhaustion and activation were measured by flow cytometry. When compared to vehicle-exposed cells, QX-314 did not affect the survival of cultured cytotoxic CD8$^+$ T cells (**a**), nor their relative expression of PD-1$^+$LAG3$^+$TIM3$^+$ (**b**), IFNγ$^+$ (**c**), TNF$^+$ (**d**) and IL-2$^+$ (**e**). (**f**) B16F10 (1x10$^5$ cells) were cultured for 24h. The cells were then exposed or not to QX-314 (0.1-1%) for an additional 24-72h, and cell count was analysed by bright-field microscopy. QX-314 did not affect B16F10 cells' survival, as measured by relative cell count changes (at each time point) in comparison to vehicle-exposed cells. (**g**–**i**) One and three days prior to tumour inoculation, 8-week-old male and female wild-type mice's right hindpaws or flanks were injected with BoNT/A (25 pg/μL; i.d.) or its vehicle. On the following day, orthotopic B16F10 cells (**g**: 5x10$^5$ cells; i.d.; **h**–**i**: 2x10$^5$ cells; i.d.) were inoculated into the area pre-exposed to BoNT/A. Starting one day post inoculation, QX-314 (0.3%) or its vehicle was administered (i.d.) once daily in another group of mice. The effects of sensory neuron silencing were tested on neuropeptide release (**g**), as well as mechanical (**h**) and thermal pain hypersensitivity (**i**). First, CGRP levels were increased in B16F10 tumour surrounding skin explant (assessed on day 15) in comparison to control skin; an effect further enhanced by capsaicin (1 μM; 3h) but was absent in skin pre-treated with BoNT/A (25 pg/μL) or QX-314 (0.3%; **g**). We also found that B16F10 injection induced mechanical (**h**) and thermal pain hypersensitivities (**i**) fourteen days post tumour inoculation. These effects were stopped by sensory neuron silencing with QX-314 or BoNT/A (**h**–**i**). (**j**) Orthotopic B16F10-mCherry-OVA cells (5x10$^5$ cells; i.d.) were inoculated into 8-week-old male and female mice. Starting one day post inoculation, QX-314 (0.3%; i.d.; 5 sites) was injected once daily around the tumour. The effect of nociceptor neuron silencing on tumour size and tumour-infiltrating CD8$^+$ T cell exhaustion was measured. We found that silencing tumour-innervating neurons increased the mice's median length of survival (-270% Mantel–Haenszel hazard ratio; measured on day 19). (**k**–**r**) Orthotopic

B16F10-mCherry-OVA cells (5x10$^5$ cells; i.d.) were inoculated into 8-week-old male and female mice. Starting one day post inoculation (*defined as prophylactic*), In other groups of mice, QX-314 daily injection started once the tumour reached a volume of -200mm$^3$ (*defined as therapeutic*). As measured seventeen days post tumour inoculation, silencing tumour innervation also decreased tumour volume (**k**,**l**) and weight (**m**), as well as the relative proportion of PD-1$^+$LAG3$^+$TIM3$^+$ (**n**) CD8$^+$ T cells. QX-314 treatment also increased the total number of intratumoral CD8$^+$ T cells (**o**), as well as relative proportion of IFNγ$^+$ (**p**), TNF$^+$ (**q**), and IL-2$^+$ (**r**) CD8$^+$ T cells. (**s**–**t**) Orthotropic B16F10-mCherry-OVA cells (5x10$^5$ cells, i.d.) were injected into mice treated with QX-314 (0.3%; i.d.) 2-3 days prior to being injected into vehicle-exposed mice. Mice from each group with similar tumour size (-100mm$^3$) were selected and exposed to αPD-L1 (6 mg/kg, i.p.) once every 3 days for a total of 9 days. Eighteen days post tumour inoculation, we found that αPD-L1-reduced tumour growth was higher (-61%) in nociceptor silenced mice than was observed in isotype vehicle-exposed mice (-49%; **s**-**t**). Data are shown as box-and-whisker plots (runs from minimal to maximal values; the box extends from 25$^{th}$ to 75$^{th}$ percentile and the middle line indicates the median), for which individual data points are given (**a**–**e**, **l**–**r**), as mean ± S.E.M (**f**,**h**,**i**,**k**,**s**,**t**), or as Mantel–Cox regression analysis (**j**). N are as follows: **a**: n = 4/groups, **b**–**e**: n = 5/groups, **f**: n = 3/groups, **g**: naïve (n = 4), vehicle (n = 7), B16F10+vehicle (n = 5), B16F10+BoNT/A (n = 5), B16F10+QX-314 (n = 5), **h**–**i**: n = 6/groups, **j**: vehicle (n = 89), QX-314 (n = 12), **k**: vehicle (n = 21), QX-314 prophylactic (n = 21), QX-314 therapeutic (n = 17), **l**: vehicle (n = 26), QX-314 therapeutic (n = 26), QX-314 prophylactic (n = 28), **m**: vehicle (n = 25), QX-314 therapeutic (n = 22), QX-314 prophylactic (n = 25), **n**: vehicle (n = 31), QX-314 therapeutic (n = 29), QX-314 prophylactic (n = 28), **o**: n = 30/groups, **p**–**r**: vehicle (n = 24), QX-314 therapeutic (n = 23), QX-314 prophylactic (n = 25), **s**: vehicle (n = 9), QX-314 (n = 13), **t**: vehicle + αPD-L1 (n = 18), QX-314 + αPLD1 (n = 13). Experiments were independently repeated two (**a**–**i**, **s**–**t**) or four (**j**–**r**) times with similar results. P-values are shown in the figure and determined by one-way ANOVA posthoc Bonferonni (**a**–**g**, **l**–**r**), two-sided unpaired Student's t-test (**h**–**i**), Mantel–Cox regression (**j**), or two-way ANOVA posthoc Bonferroni (**k**, **s**–**t**).

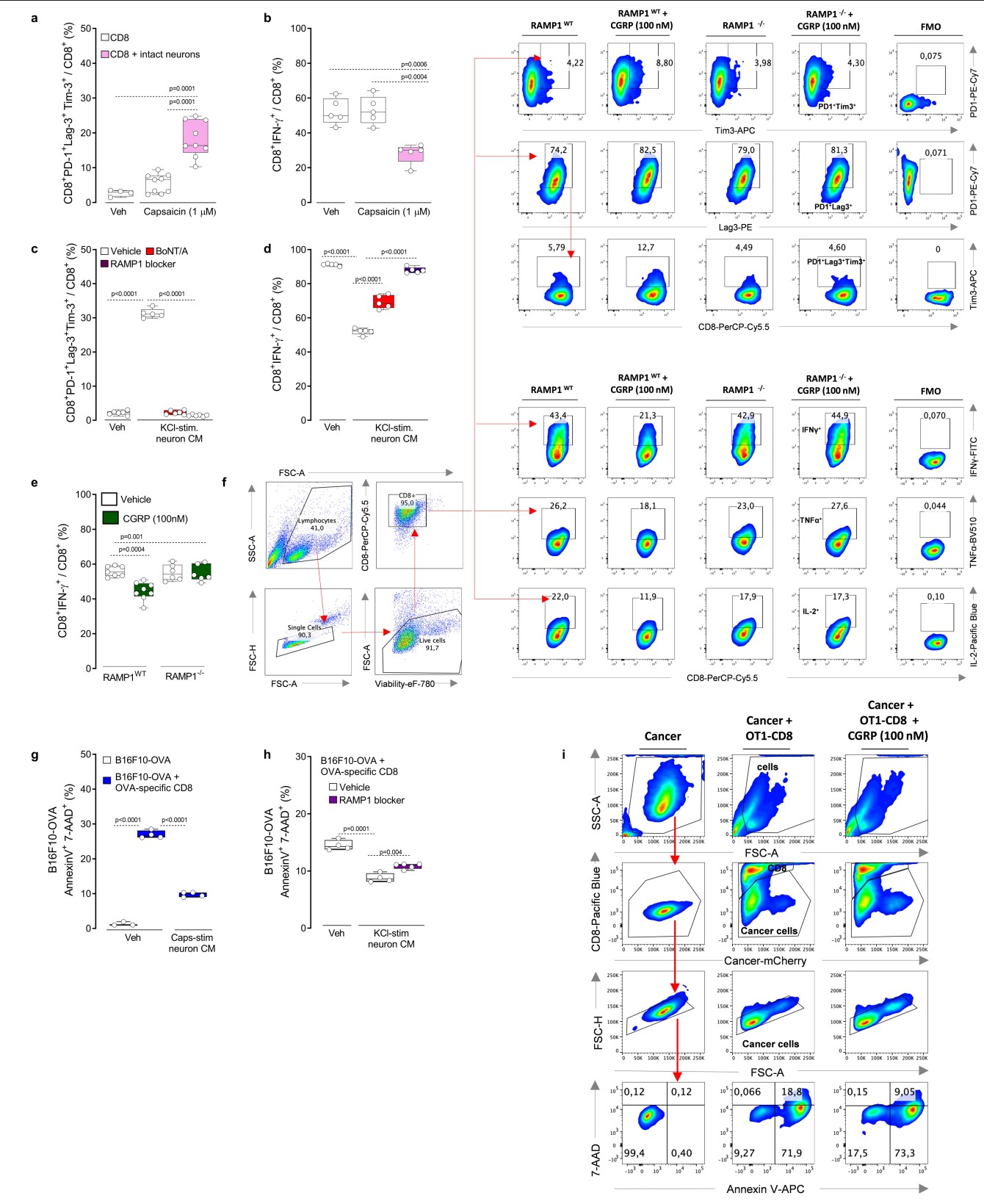

**Extended Data Fig. 9** | See next page for caption.

**Extended Data Fig. 9 | Nociceptor-released CGRP increases cytotoxic CD8⁺ T cell exhaustion.** (**a**–**b**) Splenocytes-isolated CD8⁺ T cells were cultured under $T_{c1}$-stimulating condition (*ex vivo* activated by CD3 and CD28, IL-12, and anti-IL4) for 48h. The cells were then cultured or not with wild-type DRG neurons and exposed to capsaicin (1 µM, challenged once every two days) or its vehicle. As measured after 4 days stimulation, capsaicin-stimulated intact neuron increased the proportion of PD-1⁺LAG3⁺TIM3⁺ (**a**) cytotoxic CD8⁺ T cells, while it decreased the one of IFNγ⁺ (**b**). (**c**–**d**) Splenocytes-isolated CD8⁺ T cells were cultured under $T_{c1}$-stimulating conditions (*ex vivo* activated by CD3 and CD28, IL-12, and anti-IL4) for 48h. In the presence of peptidase inhibitors (1 µL/mL), naive DRG neurons were cultured in the presence of BoNT/A (50 pg/mL) or its vehicle for 24h. The cells were then washed, stimulated (30 min) with KCl (50mM), and the conditioned medium collected. On alternate days for 4 days, the cytotoxic CD8⁺ T cells were exposed or not to a RAMP1 blocker (CGRP$_{8-37}$; 2 µg/mL) and challenge (1:2 dilution) with fresh KCl-induced conditioned medium from naive, or BoNT/A-silenced neurons. As measured after 4 days stimulation, KCl-stimulated neuron-conditioned medium increased the proportion of PD-1⁺LAG3⁺TIM3⁺ (**c**) cytotoxic CD8⁺ T cells, while it decreased the one of IFNγ⁺ (**d**). Such effect was absent when cytotoxic CD8⁺ T cells were co-exposed to the RAMP1 blocker CGRP$_{8-37}$ or challenged with the neuron conditioned medium collected from BoNT/A-silenced neurons (**c**–**d**). (**e**–**f**) Splenocytes-isolated CD8⁺ T cells from wild-type and *Ramp1⁻/⁻* mice were cultured under $T_{c1}$-stimulating conditions (*ex vivo* activated by CD3 and CD28, IL-12, and anti-IL4) for 48h. On alternate days for 4 days, the cytotoxic CD8⁺ T cells were exposed to CGRP (0.1 µM) or its vehicle. As measured after 4 days stimulation, representative flow cytometry plots (**f**) show that CGRP decrease *Ramp^{WT}* cytotoxic CD8⁺ T cells expression of IFNγ⁺ (**e**,**f**), TNF⁺ (**f**), and IL-2⁺ (**f**) when exposed to CGRP. Inversely, CGRP increase the proportion of PD-1⁺LAG3⁺TIM3⁺ in *Ramp1^{WT}* cytotoxic CD8⁺ T cells (**f**). *Ramp1⁻/⁻* cytotoxic CD8⁺ T cells were protected from the effect of CGRP (**e**–**f**). (**g**–**i**) Splenocytes-isolated CD8⁺ T cells from naive OT-I mice were cultured under Tc1-stimulating conditions (*ex vivo* activated by CD3 and CD28, IL-12, and anti-IL4) for 48h. B16F10-mCherry-OVA cells (1×10⁵ cells) were then cultured with or without OT-I cytotoxic CD8⁺ T cells (4×10⁵ cells). Tc1-stimulated OT-I-CD8⁺ T cells lead to B16F10-OVA cell apoptosis (AnnexinV⁺7AAD⁺; **g**, measured after 48h; **h**–**i**, measured after 24h). B16F10-mCherry-OVA cells elimination by cytotoxic CD8⁺ T cells was reduced when the co-cultures were challenged (1:2 dilution; once daily for two consecutive days) with fresh conditioned medium collected from capsaicin (1 µM)-stimulated naive DRG neurons (**g**; measured after 48h). Similarly, KCl (50mM)-stimulated naive DRG neurons conditioned medium (1:2 dilution) reduced B16F10-mCherry-OVA apoptosis (**h**; measured after 24h). This effect was blunted when the cells were co-exposed to the RAMP1 blocker CGRP$_{8-37}$ (**h**; 2 µg/mL; measured after 24h). CGRP (0.1 µM) challenges also reduced OT-I cytotoxic CD8⁺ T cells elimination of B16F10-OVA cell (**i**; measured after 24h). Data are shown as box-and-whisker plots (runs from minimal to maximal values; the box extends from 25^{th} to 75^{th} percentile and the middle line indicates the median), for which individual data points are given (**a**–**e**, **g**–**i**), or representative FACS plot (**f**, **i**). N are as follows: **a**: CD8 + vehicle (n = 4), CD8 + capsaicin (n = 9), CD8 + neuron + capsaicin (n = 9), **b**: n = 5/groups, **c**: CD8 (n = 6), CD8 + KCl-induced neurons CM (n = 5), CD8 + KCl-induced neurons CM + CGRP$_{8-37}$ (n = 6), CD8 + KCl-induced neurons CM + BoNT/A (n = 6), **d**: n = 5/ groups, **e**: *Ramp1^{WT}* CD8 + vehicle (n = 7), *Ramp1^{WT}* CD8 + CGRP (n = 8), *Ramp1⁻/⁻* CD8 + vehicle (n = 6), *Ramp1⁻/⁻* CD8 + CGRP (n = 6), **g**: B16F10 (n = 3), B16F10 + OT-I CD8 (n = 4), B16F10 + OT-I CD8 + KCl-induced neuron CM (n = 4), **h**: B16F10 + OT-I CD8 (n = 4), B16F10 + OT-I CD8 + KCl-induced neuron CM (n = 4), B16F10 + OT-I CD8 + KCl-induced neuron CM + CGRP$_{8-37}$ (n = 5). Experiments were repeated a minimum of three independent times with similar results. P-values are shown in the figure and determined by one-way ANOVA posthoc Bonferroni (**a**–**e**, **g**–**h**).

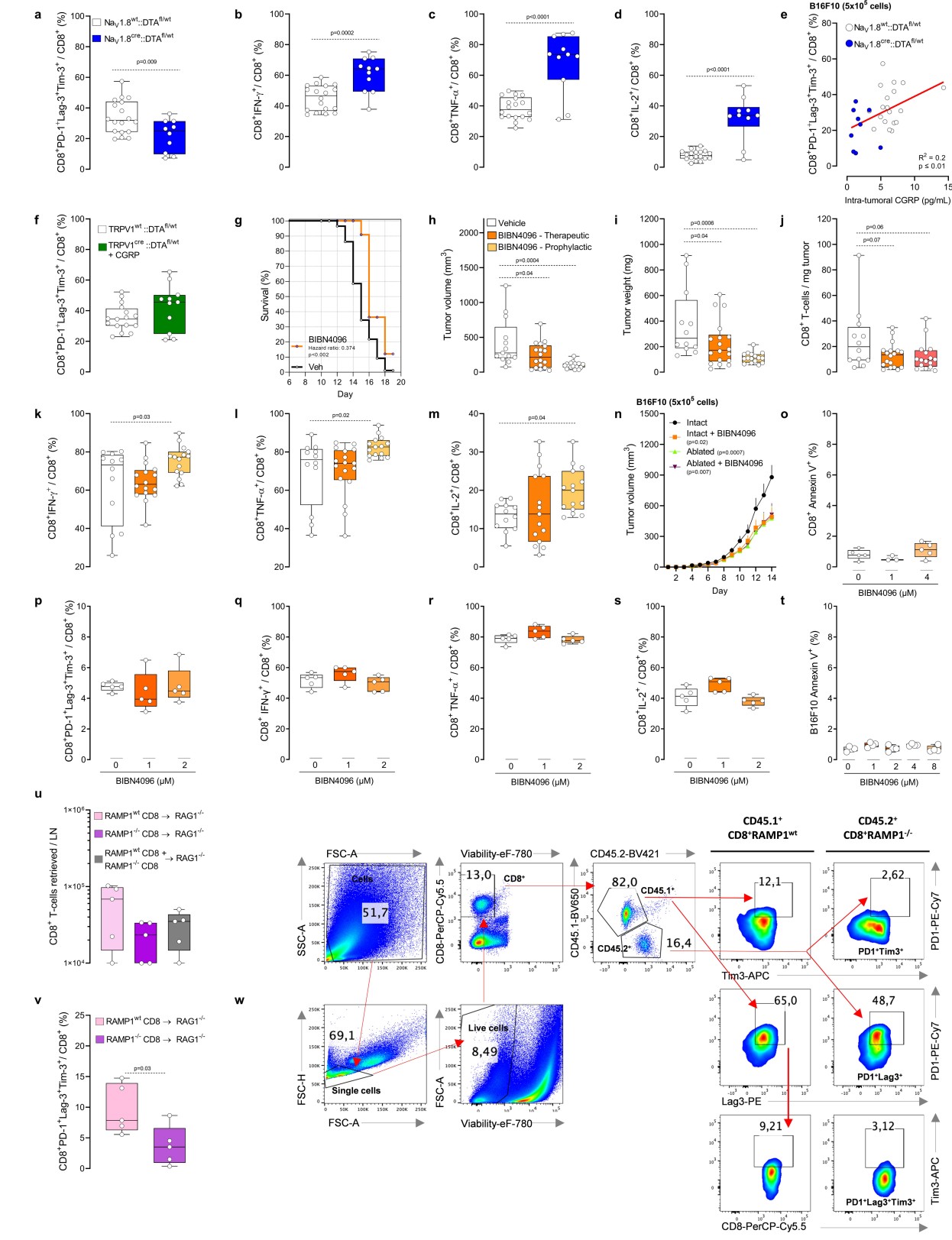

**Extended Data Fig. 10** | See next page for caption.

**Extended Data Fig. 10 | The CGRP–RAMP1 axis promotes intratumoral CD8⁺ T cell exhaustion.** (**a**–**e**) Orthotopic B16F10-mCherry-OVA (5x10⁵ cells; i.d.) cells were injected to nociceptor intact (*Nav1.8^{WT}::DTA^{fl/WT}*) and ablated mice (*Nav1.8^{cre}::DTA^{fl/WT}*). As measured fifteen days post inoculation, Na$_v$1.8⁺ nociceptor-ablated mice had lower proportion of PD-1⁺LAG3⁺TIM3⁺ (**a**) CD8⁺ T cells, but increased levels of IFNγ⁺ (**b**), TNF⁺ (**c**), IL-2⁺ (**d**) CD8⁺ T cells. B16F10-mCherry-OVA (5x10⁵ cells; i.d.)-tumour surrounding skin was also collected and capsaicin-induced CGRP release assessed by ELISA. Intratumoral CGRP levels positively correlate with the proportion of PD-1⁺LAG3⁺TIM3⁺ CD8⁺ T cells (**e**). (**f**) Orthotopic B16F10-mCherry-OVA cells (5x10⁵ cells; i.d.) were inoculated into 8-week-old female sensory neuron intact or ablated mice. In nociceptor-ablated mice, recombinant CGRP injection (100nM, i.d., once daily) rescues intratumoral CD8⁺ T cells exhaustion (PD-1⁺LAG3⁺TIM3⁺). (**g**) Orthotopic B16F10-mCherry-OVA cells (5x10⁵ cells; i.d.) were inoculated into 8-week-old male and female mice. Starting one day post inoculation, the RAMP1 antagonist BIBN4096 (5 mg/kg, i.p., every other day) was administered systemically. We found that blocking the action of CGRP on RAMP1-expressing cells, increased the mice's median length of survival (-270% Mantel−Haenszel hazard ratio; measured on day 19). (**h**–**m**) Orthotopic B16F10-mCherry-OVA cells (5x10⁵ cells; i.d.) were inoculated into 8-week-old male and female mice. Starting one day post inoculation (*defined as prophylactic*), the RAMP1 antagonist BIBN4096 (5 mg/kg, i.p., every other day) was administered systemically. In another group of mice, BIBN4096 (5 mg/kg, i.p., every other day) injections were started once the tumour reached a volume of -200mm³ (*defined as therapeutic*). The effect of nociceptor neuron-silencing on tumour size and tumour-infiltrating CD8⁺ T cell exhaustion was measured. As assessed thirteen days post tumour inoculation, BIBN4096 decreased tumour volume (**h**) and weight (**i**) but increased the relative proportion of IFNγ⁺ (**k**), TNF⁺ (**l**), and IL-2⁺ (**m**) CD8⁺ T cells. BIBN4096 had no effect on the number of intratumoral CD8⁺ T cells (**j**). When administered as therapeutic, BIBN4096 reduced tumour volume (**h**) and weight (**i**) but had limited effect on CD8⁺ T cells' cytotoxicity (**j**–**m**). (**n**) Orthotopic B16F10-mCherry-OVA cells (5x10⁵ cells; i.d.) were inoculated into 8-week-old male and female sensory neuron-intact (*Trpv1^{WT}::DTA^{fl/WT}*) and ablated (*Trpv1^{cre}::DTA^{fl/WT}*) mice. Starting one day post inoculation, BIBN4096 (5 mg/kg) or its vehicle was administered (i.p.) on alternate days; effects on tumour volume were measured. Fourteen days post tumour inoculation, we found that tumour growth was reduced in sensory neuron-ablated mice and in BIBN4096-treated mice. BIBN4096 had no additive effect when given to sensory neuron-ablated mice. (**o**–**s**) Splenocytes-isolated CD8⁺ T cells from naïve C57BL6J mice were cultured under Tc1-stimulating conditions (*ex vivo* activated by CD3 and CD28, IL-12, and anti-IL4) for 48h. The cells were then exposed to BIBN4096 (1–4 μM) for 24h; effects on apoptosis, exhaustion and activation were measured by flow cytometry. When compared to vehicle-exposed cells, BIBN4096 did not affect the survival (**o**) of cultured cytotoxic CD8⁺ T cells, nor their relative expression of PD-1⁺LAG3⁺TIM3⁺ (**p**), IFNγ⁺ (**q**), TNF⁺ (**r**), and IL-2⁺ (**s**). (**t**) B16F10 cells (1x10⁵ cells) were cultured for 24h. The cells were then exposed (or not) to BIBN4096 (1-8 μM) for an additional 24h; effects on apoptosis were measured by flow cytometry. BIBN4096 did not trigger B16F10 cells apoptosis, as measured by the mean fluorescence intensity of Annexin V. (**u-w**) Naive splenocyte CD8⁺ T cells were FACS purified from *Ramp1^{WT}* (CD45.1⁺) or *Ramp1^{−/−}* (CD45.2⁺) mice, expanded and stimulated (*CD3 and CD28 + IL-2*) *in vitro*. 8-week-old female *Rag1^{−/−}* mice were transplanted (i.v., 2.5x10⁶ cells) with either *Ramp1^{−/−}* or *Ramp1^{WT}* CD8⁺ T cells or 1:1 mix of *Ramp1^{−/−}* and *Ramp1^{WT}* CD8⁺ T cells. One week post transplantation, the mice were inoculated with B16F10-mCherry-OVA cells (5x10⁵ cells; i.d.). Ten days post tumour inoculation, we retrieved a similar number of tumours draining lymph node CD8⁺ T cells across the three tested groups (**u**). The relative proportion of intra-tumour PD-1⁺LAG3⁺TIM3⁺ CD8⁺ T cells was lower in *Ramp1^{−/−}* transplanted mice (**v**). Within the same tumour, intratumoral CD8⁺ T cell exhaustion was immunophenotyped by flow cytometry (*representative panel shown in **w***) and showed that the relative proportion of PD-1⁺LAG3⁺TIM3⁺ CD8⁺ T cells was -3-fold lower in *Ramp1^{−/−}* CD8⁺ T cells than in *Ramp1^{WT}* CD8⁺ T cells (**w**). Data are shown as box-and-whisker plots (runs from minimal to maximal values; the box extends from 25th to 75th percentile and the middle line indicates the median), for which individual data points are given (**a**–**d**, **f**, **h**–**m**, **o**–**v**), linear regression (**e**), Mantel−Cox regression (**g**), mean ± S.E.M (**n**), or as FACS plot (**w**). N are as follows **a**–**e**: *Nav1.8^{WT}::DTA^{fl/WT}* (n = 18), *Nav1.8^{cre}::DTA^{fl/WT}* (n = 10), **f**: *Trpv1^{WT}::DTA^{fl/WT}* (n = 16), *Trpv1^{cre}::DTA^{fl/WT}* +CGRP (n = 11), **g**: vehicle (n = 89), BIBN4096 (n = 16), **h**–**m**: Vehicle (n = 13), BIBN4096 therapeutic (n = 18), BIBN4096 prophylactic (n = 16), **n**: *Trpv1^{WT}::DTA^{fl/WT}* + vehicle (n = 8), *Trpv1^{WT}::DTA^{fl/WT}* + BIBN4096 (n = 9), *Trpv1^{cre}::DTA^{fl/WT}* + vehicle (n = 7), *Trpv1^{cre}::DTA^{fl/WT}* + BIBN4096 (n = 7), **o**: vehicle (n = 5), 1μM BIBN4096 (n = 3), 4 μM BIBN4096 (n = 5), **p**–**s**: n = 5/groups, **t**: n = 4/groups, **u**–**w**: n = 5/groups. Experiments were independently repeated twice (**a**–**f**, **n**–**w**) or four (**g**–**m**) times with similar results. P-values are shown in the figure and determined by two-sided unpaired Student's t-test (**a**–**d**, **f**,**v**), simple linear regression analysis (**e**), Mantel−Cox regression (**g**), by one-way ANOVA posthoc Bonferroni (**h**–**m**; **o**–**u**), or two-way ANOVA post-hoc Bonferroni (**n**).

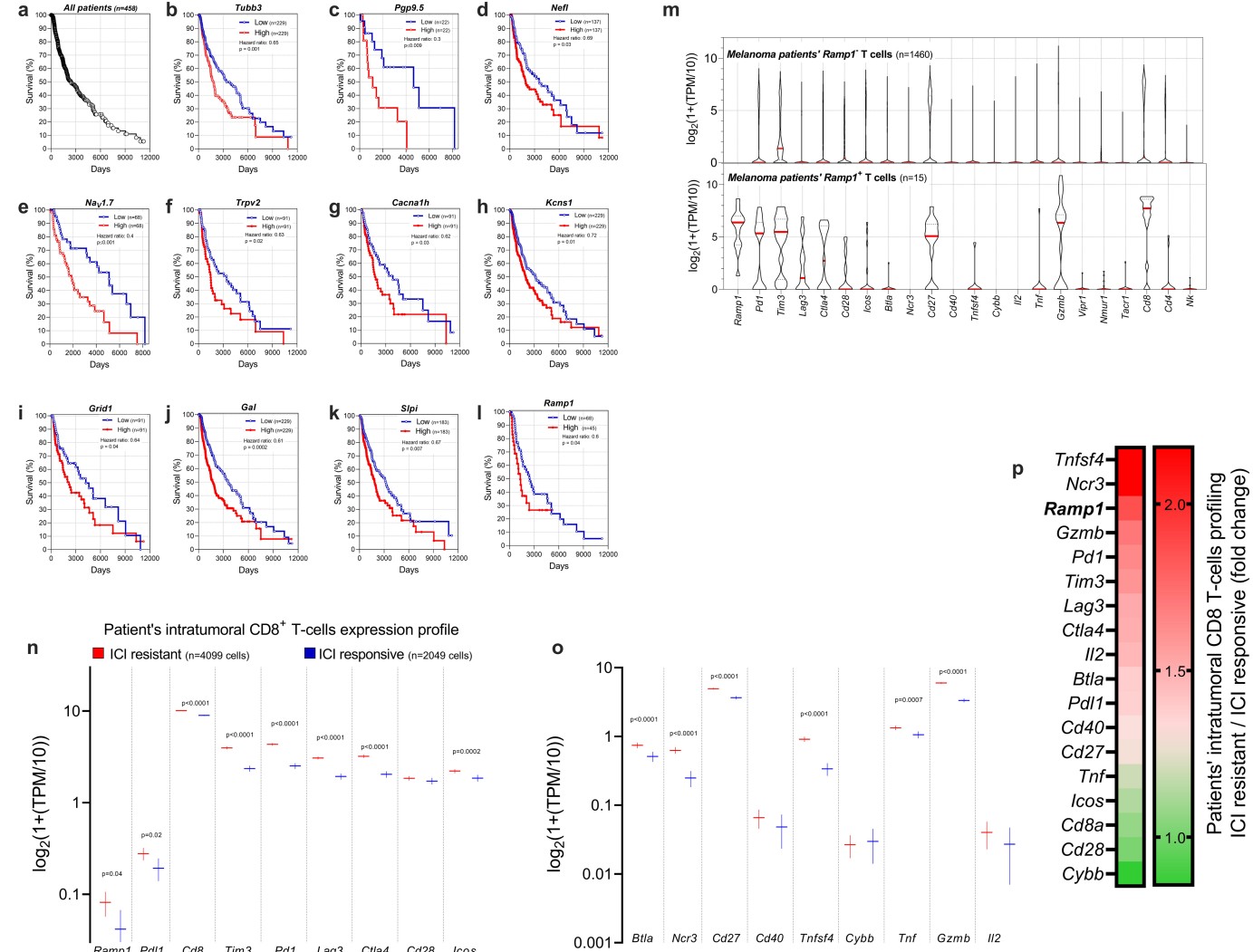

**Extended Data Fig. 11 | *RAMP1* expression in patient melanoma-infiltrating T cells correlates with worsened survival and poor responsiveness to ICIs.** (**a–l**) In silico analysis of Cancer Genome Atlas (TCGA) data linked the survival rate among 459 patients with melanoma with their relative expression levels of various genes of interest (determined by bulk RNA sequencing of tumour biopsy). Kaplan–Meier curves show the patients' survival after segregation in two groups defined by their low or high expression of a gene of interest. Increased gene expression (labelled as high; red curve) of *TUBB3* (**b**), *PGP9.5* (**c**), *Nav1.7* (**E**), *SLPI* (**k**) and *RAMP1* (**l**) in biopsy correlate with decreased patient survival (p≤0.05). The mantel–Haenszel hazard ratio and number of patients included in each analysis are shown in the figure (**a–l**). Experimental details were defined in Cancer Genome Atlas (TCGA)[40]. (**m**) In silico analysis of single-cell RNA sequencing of human melanoma-infiltrating T cells revealed that *RAMP1*⁺ T cells downregulated *Il-2* expression and strongly overexpressed

several immune checkpoint receptors (*PD-1*, *TIM3*, *LAG3*, *CTLA4*, *CD28*, *ICOS*, *BTLA*, *CD27*) in comparison to *RAMP1*⁻ T cells. Individual cell data are shown as a log₂ of 1+ (transcript per million / 10). Experimental details and cell clustering were defined in Tirosh et al[42]. N are defined in each panel. (**n–p**) On the basis of the clinical response of patients with melanoma to immune checkpoint blocker, patients were clustered into two groups defined as ICI-responsive or ICI-resistant[41]. *In silico* analysis of single-cell RNA sequencing of patients' biopsies revealed that tumour-infiltrating CD8⁺ T cells from patients who were resistant to ICIs significantly overexpressed *RAMP1* (2.0-fold), *PD-1* (1.7-fold), *LAG3* (1.6-fold), *CTLA4* (1.6-fold), and *TIM3* (1.7-fold; **n–p**). Individual cell data are shown as a log₂(1+(transcript per million/10). Experimental details and cell clustering were defined in Jerby-Arnon et al[41]. P-values are shown in the figure and determined by two-sided unpaired Student's t-test. N are defined in each panel (**n–o**).

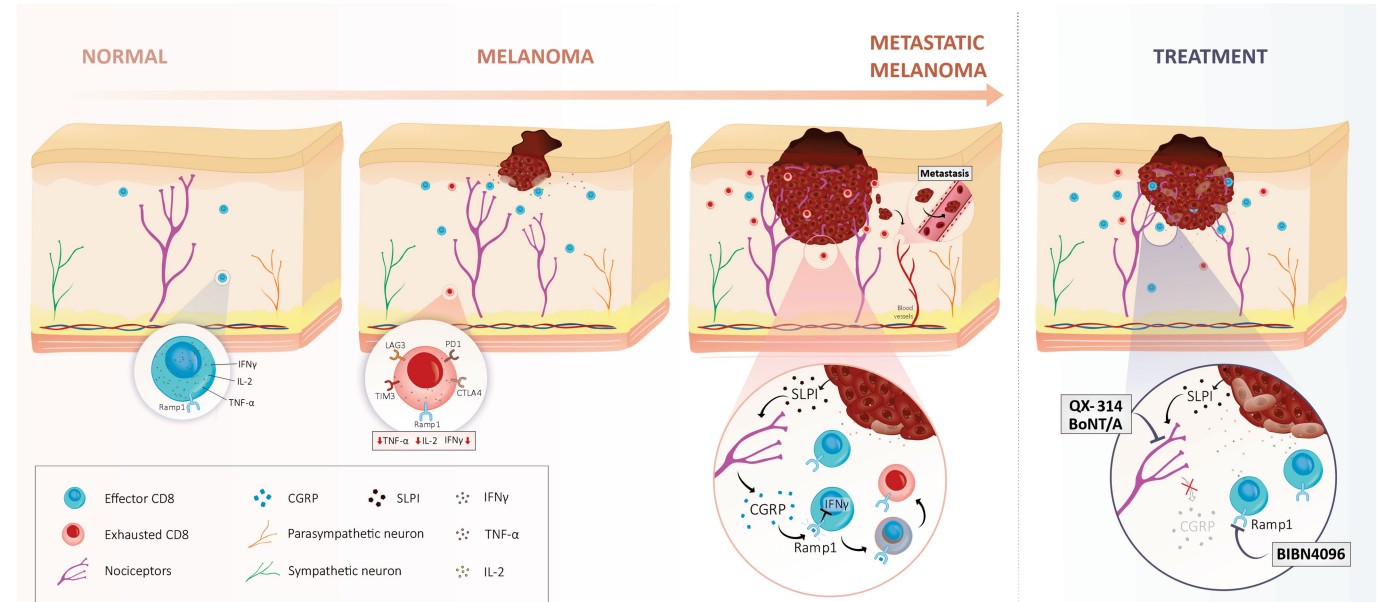

**Extended Data Fig. 12 | Melanoma-innervating nociceptors attenuate cancer immunosurveillance.** Melanoma growth sets off anti-tumour immune responses, including the infiltration of effector CD8 T cells and their subsequent release of cytotoxic cytokines (i.e., IFNγ, TNF, Granzyme B). By acting on tissue-resident nociceptor neurons, melanoma-produced SLPI promotes pain hypersensitivity, tweaks the neurons' transcriptome, and drives neurite outgrowth. These effects culminate in dense melanoma innervation by nociceptors and abundant release of immunomodulatory neuropeptides. CGRP, one such peptide, acts on tumour-infiltrating effector CD8⁺ T cells that express the CGRP receptor RAMP1, increasing their expression of immune checkpoint receptors (i.e., PD-1, LAG3, TIM3). Therefore, along with the immunosuppressive environment present in the tumour, nociceptor-produced CGRP leads to the functional exhaustion of tumour-infiltrating CD8⁺ T cells, which opens the door to unchecked proliferation of melanoma cells. Genetically ablating (i.e., TRPV1 lineage) or pharmacologically silencing (i.e., QX-314, BoNT/A) nociceptor neurons as well as blocking the action of CGRP on RAMP1 using a selective antagonist (i.e., BIBN4096) prevents effector CD8⁺ T cells from undergoing exhaustion. Therefore, targeting melanoma-innervating nociceptor neurons constitutes a novel strategy to safeguard host anti-tumour immunity and stop tumour growth.

# Reporting Summary

## Statistics

For all statistical analyses, confirm that the following items are present in the figure legend, table legend, main text, or Methods section.

| n/a | Confirmed | |
|---|---|---|
| ☐ | ☒ | The exact sample size (*n*) for each experimental group/condition, given as a discrete number and unit of measurement |
| ☐ | ☒ | A statement on whether measurements were taken from distinct samples or whether the same sample was measured repeatedly |
| ☐ | ☒ | The statistical test(s) used AND whether they are one- or two-sided *Only common tests should be described solely by name; describe more complex techniques in the Methods section.* |
| ☐ | ☒ | A description of all covariates tested |
| ☐ | ☒ | A description of any assumptions or corrections, such as tests of normality and adjustment for multiple comparisons |
| ☐ | ☒ | A full description of the statistical parameters including central tendency (e.g. means) or other basic estimates (e.g. regression coefficient) AND variation (e.g. standard deviation) or associated estimates of uncertainty (e.g. confidence intervals) |
| ☐ | ☒ | For null hypothesis testing, the test statistic (e.g. *F*, *t*, *r*) with confidence intervals, effect sizes, degrees of freedom and *P* value noted *Give P values as exact values whenever suitable.* |
| ☒ | ☐ | For Bayesian analysis, information on the choice of priors and Markov chain Monte Carlo settings |
| ☒ | ☐ | For hierarchical and complex designs, identification of the appropriate level for tests and full reporting of outcomes |
| ☐ | ☒ | Estimates of effect sizes (e.g. Cohen's *d*, Pearson's *r*), indicating how they were calculated |

*Our web collection on statistics for biologists contains articles on many of the points above.*

## Software and code

Policy information about availability of computer code

| | |
|---|---|
| Data collection | Cell were immunophenotyped using a LSRFortessa or FACSCanto II (Becton Dickinson). Calcium microscopy analysis was performed using Nikon eclipse ti2 microscope (NIS-Elements Advanced Research version 4.5). Patient biopsy images were collected using an Olympus BX51 bright-field microscope. Mouse tumor innervation images were acquired using an Olympus FV3000 confocal imaging system. GraphPad Prism v9 and Microsoft Excel (version 2019) was used for data entry, graph construction and data analysis. |
| Data analysis | GraphPad Prism (Version 9.0) and Microsoft Excel (version 2019) were used for data entry, graph construction, and data analysis. Image analysis (neurite length, ramification index) was performed using ImageJ macros (Fiji; version 1.53c). Calcium microscopy analysis was performed using Nikon eclipse ti2 microscope (NIS-Elements Advanced Research version 4.5). Flow cytometry data were analyzed using FlowJo (version 10.0.0). TCGA data were accessed via Oncomine (www.oncomine.com for gene expression) and OncoLnc (www.oncolnc.org for survival). Single-cell RNA sequencing was analyzed using the Broad single-cell portal (https://singlecell.broadinstitute.org). Human and mouse immune cell gene profiles were respectively analyzed using the human protein atlas (https://www.proteinatlas.org/humanproteome/immune+cell) and Immunological genome project (https://www.immgen.org/). |

For RNA sequencing, the reads were aligned to the mouse reference genome GRCm38/mm10 (GenBank assembly accession GCA_000001635.2) using STAR (version used: 2.5.4a, 2.5.1b, 2.7).

Aligned reads were assigned to genic regions using the featureCounts function from Subread (version 1.6.4 22).

Hierarchical clustering was computed using the heatmap.2 function (ward.D2 method) from Gplots R package (version 3.1.3).

Differential gene expression analysis was carried out by DeSeq2 (version used: 1.18.1 or 1.28.1).

For manuscripts utilizing custom algorithms or software that are central to the research but not yet described in published literature, software must be made available to editors and reviewers. We strongly encourage code deposition in a community repository (e.g. GitHub). See the Nature Portfolio guidelines for submitting code & software for further information.

## Data

Policy information about availability of data

All manuscripts must include a data availability statement. This statement should provide the following information, where applicable:
- Accession codes, unique identifiers, or web links for publicly available datasets
- A description of any restrictions on data availability
- For clinical datasets or third party data, please ensure that the statement adheres to our policy

All data are readily available online (https://www.talbotlab.com/nature), from the corresponding author (email: sebas.talbot@gmail.com) and within the paper and its Supplementary Information files.

The RNA sequencing datasets have been deposited in the National Center for Biotechnology Information's Gene Expression Omnibus (#GSE205863, #GSE205864, #GSE205865).

GSE205863 is available at: https://www.ncbi.nlm.nih.gov/geo/query/acc.cgi?acc=GSE205863

GSE205864 is available at: https://www.ncbi.nlm.nih.gov/geo/query/acc.cgi?acc=GSE205864

GSE205865 is available at: https://www.ncbi.nlm.nih.gov/geo/query/acc.cgi?acc=GSE205865

Mouse reference genome GRCm38/mm10 is available at www.ncbi.nlm.nih.gov/assembly/GCF_000001635.20

Oncomine is available at: www.oncomine.com

OncoLnc is available at: www.oncolnc.org

Broad single-cell bioportal is available at : https://singlecell.broadinstitute.org
Tirosh et al., is available at: https://singlecell.broadinstitute.org/single_cell/study/SCP11/melanoma-intra-tumor-heterogeneity
Jerby-Arnon is available at: https://singlecell.broadinstitute.org/single_cell/study/SCP109/melanoma-immunotherapy-resistance#study-summary

Human immune cell gene profiles are accessible at: www.proteinatlas.org/humanproteome/immune+cell

Mouse immune cell gene profiles are available at: www.immgen.org/

Meta-analysis of nociceptor neurons gene profiles are available at: www.talbotlab.com/dataset

# Field-specific reporting

Please select the one below that is the best fit for your research. If you are not sure, read the appropriate sections before making your selection.

☒ Life sciences ☐ Behavioural & social sciences ☐ Ecological, evolutionary & environmental sciences

For a reference copy of the document with all sections, see nature.com/documents/nr-reporting-summary-flat.pdf

# Life sciences study design

All studies must disclose on these points even when the disclosure is negative.

| Sample size | Statistical methods were not used to predetermine sample size. The size of the cohort, based on similar studies in the field, was validated by pilot studies. All sample sizes are indicated in the figures and/or figure legends. All n values are clearly indicated within the figure legends. In the only case where a range is used (figure 2a/b), exact n values are provided in the source data files. For in vivo experiments we used n>8 animals. For in vitro experiments where replicate samples were used, we repeated the experiments at least 3 independent times to confirm the findings. For other mouse experiments a minimum of 5 mice were used to ensure proper statistics could be utilized. We determined this to be sufficient as per our pilot data, use of internal controls and/or the observed variability between within experimental groups. |
|---|---|
| Data exclusions | No data were excluded. |

| Replication | This is indicated in the figures, figure legends and/or methods. On the graphs individual dots represent individual samples/mice used. For each experiment, all attempts at replication were successful and our findings showed comparable results. |
| --- | --- |
| Randomization | Animals in a particular cohort were generated from one breeding pair and all offspring (e.g. nociceptor intact and ablated mice) were co-housed and, in respect with the ARRIVE guidelines, were randomly allocated into each experimental group. For in vitro experiments, randomization was used for treatment selection. In some calcium microscopy experiments, the investigators performing the data collection were tasked to select all ligands responsive cells for downstream analysis. In these rare cases. randomization was not used for cell selection. |
| Blinding | Double blind was used for all in vivo treatments. In calcium microscopy involving co-culture (e.g. nociceptors and cancer cells), the differences in cell morphology are obvious and, therefore, the investigator performing the experiment was not blind. However, these investigators were always blinded to the treatment being applied to the cells and a second blinded investigator performed the downstream data analysis. |

# Reporting for specific materials, systems and methods

We require information from authors about some types of materials, experimental systems and methods used in many studies. Here, indicate whether each material, system or method listed is relevant to your study. If you are not sure if a list item applies to your research, read the appropriate section before selecting a response.

## Materials & experimental systems

| n/a | Involved in the study |
| --- | --- |
| ☐ | ☒ Antibodies |
| ☐ | ☒ Eukaryotic cell lines |
| ☒ | ☐ Palaeontology and archaeology |
| ☐ | ☒ Animals and other organisms |
| ☐ | ☒ Human research participants |
| ☒ | ☐ Clinical data |
| ☒ | ☐ Dual use research of concern |

## Methods

| n/a | Involved in the study |
| --- | --- |
| ☒ | ☐ ChIP-seq |
| ☐ | ☒ Flow cytometry |
| ☒ | ☐ MRI-based neuroimaging |

## Antibodies

| Antibodies used | Antibody (company, catalog number, dilution, clone):<br><br>anti-AnnexinV-APC and 7-AAD (BioLegend), cat no: 640930, Dilution: 1:100<br>anti-CD11b-APC/Cy7 (BioLegend), cat no: 101226, Dilution: 1:100<br>anti-CD16/32 (Biolegend), cat no: 156604, Dilution: 1:100<br>anti-CD28 (Bio X Cell), cat no: BE0015-5, Dilution: 1:4000<br>anti-CD3 (Bio X Cell), cat no: BE0001-1, Dilution: 200μg/mouse<br>anti-CD3 (Bio X Cell), cat no: BE00011, Dilution: 1:3000<br>anti-CD45.1-BV421 (BioLegend), cat no: 110732, Dilution: 1:100<br>anti-CD45.2-BV650 (BioLegend), cat no: 109836, Dilution: 1:100<br>anti-CD45-Alexa Fluor 700 (BioLegend), cat no: 103128, Dilution: 1:100<br>anti-CD45-BV421 (BioLegend), cat no: 103134, Dilution: 1:100<br>anti-CD4-FITC (BioLegend), cat no: 100406, Dilution: 1:100<br>anti-CD4-PerCP/Cyanine5.5 (BioLegend), cat no: 100540, Dilution: 1:100<br>anti-CD8 (Bio X Cell), cat no: BP0061, Dilution: 200μg/mouse<br>anti-CD8-AF700 (BioLegend), cat no: 100730, Dilution: 1:100<br>anti-CD8-BV421 (BioLegend), cat no: 100753, Dilution: 1:100<br>anti-CD8-Pacific Blue (BioLegend), cat no: 100725, Dilution: 1:100<br>anti-CD8-PerCP/Cyanine5.5 (BioLegend), cat no: 100734, Dilution: 1:100<br>anti-GFP (Aves Labs), cat no: GFP-1020, Dilution: 1:500<br>anti-H-2Kb/OVA257-264 (NIH tetramer core facility), IEDB ID: 58560, Dilution: 1:100<br>anti-IFN-γ-APC (BioLegend), cat no: 505810, Dilution: 1:100<br>anti-IFN-γ-FITC (BioLegend), cat no: 505806, Dilution: 1:100<br>anti-IgG(H+L)-AF488 (Invitrogen), cat no: A28175, Dilution: 1:500<br>anti-IL-2-BV510 (BioLegend), cat no: 503833, Dilution: 1:100<br>anti-IL-2-Pacific Blue (BioLegend), cat no: 503820, Dilution: 1:100<br>anti-IL2-Pecy7 (BioLegend), cat no: 503832, Dilution: 1:100<br>anti-IL4 (Bio X Cell), cat no: BE0045, Dilution: 0.1:1000 - 1:1000<br>anti-Lag3-PE (BioLegend), cat no: 125208, Dilution: 1:100<br>anti-Lag3-PerCP/Cyanine5.5 (BioLegend), cat no: 125212, Dilution: 1:100<br>anti-mCherry (OriGene), cat no: AB0040-200, Dilution: 1:500<br>anti-PD1-PE-Cy7 (BioLegend), cat no: 329917, Dilution: 1:100<br>anti-PDL1 (Bio X Cell), cat no: BE0101, Dilution: 6mg/kg<br>anti-Tim3-APC (BioLegend), cat no: 119706, Dilution: 1:100<br>anti-TNFα-BV510 (BioLegend), cat no: 506339, Dilution: 1:100<br>anti-TNFα-BV711 (BioLegend), cat no: 506349, Dilution: 1:100<br>anti-TNFα-PE (BioLegend), cat no: 506306, Dilution: 1:100<br>anti-TRPV1 (Alomone Labs), cat no: ACC-030, Dilution: 1:100<br>DAPI (Vector Laboratories), cat no: H-1000, Dilution: 1:2000 |
| --- | --- |

Viability Dye-eFluor780 (eBioscience), cat no: 65-0865-14, Dilution: 1:1000
ZombieAqua (BioLegend), cat no: 423102, Dilution: 1:100

Validation

Anti-AnnexinV-APC and 7-AAD was validated for flow cytometry and for use in mouse cell lines by supplier, and previously used in ≥ 16 publications (Broggi A, et al. 2017. Nat Immunol. 18:1084)

Anti-CD11b-APC/Cy7 was validated for flow cytometry and for use in mouse cell lines by supplier, and previously used in ≥ 128 publications (Kleppe M et al. 2018. Cancer cell. 33(1):29-43)

Anti-CD16/32 was validated for flow cytometry and for use in mouse cell lines by supplier, and previously used in ≥ 12 publications (Oguri Y, et al. 2020. Cell. 182(3):563-577.e20)

Anti-CD28 was validated for in vitro T cell stimulation/activation by supplier, and previously used in ≥ 9 publications (Vegran F et al., 2014. Nat Immunology, 15(8):758-66)

Anti-CD3 was validated for in vitro T cell stimulation/activation by supplier, and previously used in ≥ 19 publications (Wendland K et al., 2018. J Immunol, 15;201(2):524-532)

Anti-CD3 was validated for in vivo T cell depletion in the mouse by supplier, and previously used in ≥ 19 publications (Peng B. et al., 2009. Blood, 12;114(20):4373-82)

Anti-CD45.1-BV421 was validated for flow cytometry and for use in mouse cell lines by supplier, and previously used in ≥ 22 publications (Phan TG, et al. 2007. Nature Immunol. 8:992)

Anti-CD45.2-BV650 was validated for flow cytometry and for use in mouse cell lines by supplier, and previously used in ≥ 11 publications (Kohlmeier JE, et al. 2008. Immunity. 29:101)

Anti-CD45-Alexa Fluor 700 was validated for flow cytometry and for use in mouse cell lines by supplier, and previously used in ≥ 96 publications (Radtke AJ, et al. 2022. Nat Protoc. 17:378-401)

Anti-CD45-BV421 was validated for flow cytometry and for use in mouse cell lines by supplier, and previously used in ≥ 50 publications (Haynes NM, et al. 2007. J. Immunol. 179:5099)

Anti-CD4-FITC was validated for flow cytometry and for use in mouse cell lines by supplier, and previously used in ≥ 100 publications (Zheng B, et al. 1996. J. Exp. Med. 184:1083)

Anti-CD4-PerCP/Cyanine5.5 was validated for flow cytometry and for use in mouse cell lines by supplier, and previously used in ≥ 48 publications (León-Ponte M, et al. 2007. Blood 109:3139)

Anti-CD8 was validated for cell depletion by supplier, and previously used in ≥ 19 publications (Vegran, F., et al. (2014) Nat Immunol 15(8): 758-766)

Anti-CD8-AF700 was validated for flow cytometry and for use in mouse cell lines by supplier, and previously used in ≥ 54 publications (Shih FF, et al. 2006. J. Immunol. 176:3438)

Anti-CD8-BV421 was validated for flow cytometry and for use in mouse cell lines by supplier, and previously used in ≥ 43 publications (Bouwer HGA, et al. 2006. P. Natl. Acad. Sci. USA 103:5102)

Anti-CD8-Pacific Blue was validated for flow cytometry and for use in mouse cell lines by supplier, and previously used in ≥ 38 publications (Ko SY, et al. 2005. J. Immunol. 175:3309)

Anti-CD8-PerCP/Cyanine5.5 was validated for flow cytometry and for use in mouse cell lines by supplier, and previously used in ≥ 110 publications (Bankoti J, et al. 2010. Toxicol. Sci. 115:422)

Anti-GFP was validated for flow cytometry and for use in mouse cell lines by supplier and previously used in ≥ 49 publications (Zimmerman A et al., 2019 Neuron. 102(2):420-434.e8)

Anti-H-2Kb/OVA257-264 was validated for flow cytometry by the NIH tetramer core and previously used in Crittenden et al., 2018, Sci Rep. 3;8(1):7012

Anti-IFN-γ-APC was validated for flow cytometry and for use in mouse cell lines by supplier and previously used in ≥ 142 publications (Ferrick D, et al. 1995. Nature 373:255)

Anti-IFN-γ-FITC was validated for flow cytometry and for use in mouse cell lines by supplier and previously used in ≥ 74 publications (Ko SY, et al. 2005. J. Immunol. 175:3309)

Anti-IgG(H+L)-AF488 was validated for immunofluorescence by supplier and previously used in ≥ 305 publications (Miao et al., 2022. J Exp Med. 5;219(9):e20220214)

Anti-IL-2-BV510 was validated for flow cytometry and for use in mouse cell lines by supplier and previously used in ≥ 2 publications (Dikiy S, et al. 2021. Immunity. 54(5):931-946.e11)

Anti-IL-2-Pacific Blue was validated for flow cytometry and for use in mouse cell lines by supplier and previously used in Mohammed RN, et al. 2019. Sci Rep. 4.185416667

Anti-IL2-Pecy7 was validated for flow cytometry and for use in mouse cell lines by supplier and previously used in ≥ 5 publications (Xu W, et al. 2021. Immunity. 54(3):526-541.e7)

Anti-IL4 was validated for neutralisation by supplier and previously used in ≥ 19 publications (Tang W, et al. Immunity. 2014 Oct 16;41(4):555-66)

Anti-Lag3-PE was validated for flow cytometry and for use in mouse cell lines by supplier and previously used in ≥ 14 publications (Dong MB, et al. 2020. Cell. 178(5):1189-1204.e23)

Anti-Lag3-PerCP/Cyanine5.5 was validated for flow cytometry and for use in mouse cell lines by supplier and previously used in ≥ 10 publications (Vardhana SA, et al. 2020. Nat Immunol. 1.584722222)

Anti-mCherry was validated for immunofluorescence by supplier and previously used in ≥ 12 publications (Abraira et al., Cell. 2017 168(1-2):295-310.e19)

Anti-PD1-PE-Cy7 was validated for flow cytometry and for use in mouse cell lines by supplier and previously used in ≥ 29 publications (Barili V, et al. 2020. Nat Commun. 0.877777778)

Anti-PDL1 was validated for in vivo PD-L1 blockade by supplier and previously used in ≥ 18 publications (Twyman-Saint Victor C, et al., Nature 2015. 16;520(7547):373-7)

Anti-Tim3-APC was validated for flow cytometry and for use in mouse cell lines by supplier and previously used in ≥ 6 publications (Mooney L, et al. 2002. Nature 415:536)

Anti-TNFα-BV510 was validated for flow cytometry and for use in mouse cell lines by supplier and previously used in ≥ 8 publications (Li C, et al. 2020. Immunity. 52(1):201-202)

Anti-TNFα-BV711 was validated for flow cytometry and for use in mouse cell lines by supplier and previously used in ≥ 6 publications (Routhu NK, et al. 2021. Immunity. 54(3):542-556.e9)

Anti-TNFα-PE was validated for flow cytometry and for use in mouse cell lines by supplier and previously used in ≥ 55 publications (Logan K Smith et al. 2018. Immunity. 48(2):299-312)

Anti-TRPV1 was validated for immunohistochemistry and for use in human sample staining by supplier and previously used in ≥ 167 publications (Nam, J.H. et al. (2015) Brain 138, 3610)

DAPI was validated for immunofluorescence and for use in mouse sample staining by supplier and previously used in ≥ 9300 publications (Bae E et al., 2022. Nature Comm. 25;13(1):4268)

Viability Dye-eFluor780 was validated for flow cytometry and for use in mouse cell lines by supplier and previously used in Mathur et al., JCI Insight. 2021. 22;6(24):e148510.

ZombieAqua was validated for flow cytometry and for use in mouse cell lines by supplier and previously used in ≥ 174 publications (Barry KC, et al. 2018. Nat Med. 24:1178)

# Eukaryotic cell lines

Policy information about cell lines

| Cell line source(s) | B16F0 (ATCC, #CRL-6322)<br>B16F10 (ATCC, #CRL-6475)<br>B16F10-OVA  (Matthew F. Krummel, UCSF)<br>B16F10-OVA-mCherry (Matthew F. Krummel, UCSF)<br>B16F10-eGFP (Imanis, #CL053)<br>YUMM1.7 (ATCC, #CRL-3362)<br>YUMMER1.7 (Marcus Bosenberg, Yale U)<br>Non-tumorigenic keratinocytes (CellnTEC, #MPEK-BL6100) |
|---|---|
| Authentication | Non-commercial cell lines (B16F10-OVA, B16F10-OVA-mCherry and B16F10-eGFP) were authenticated using antibody (against OVA, eGFP, mCherry) and/or imaging as well as morphology and/or growth property. Commercial cell line (ATCC, Imanis, CellnTec) provides a certificate of analysis in which they validate the cell lines with specific test and procedures such as growth property, morphology, mycoplasma detection, species determination, and sterility test. |
| Mycoplasma contamination | All the cell lines tested negative for mycoplasma |
| Commonly misidentified lines<br>(See ICLAC register) | No commonly misidentified lines were used in this study. |

# Animals and other organisms

Policy information about studies involving animals; ARRIVE guidelines recommended for reporting animal research

| Laboratory animals | Both males and females mice (mus musculus) were used equally, and they were used at 6 weeks of age up to 20 weeks of age. Mice were housed in standard environmental conditions (12h light/dark cycle; 23oC; food and water ad libitum) at facilities accredited by the Canadian Council of Animal Care (UdeM) or Association for Assessment and Accreditation of Laboratory Animal Care (BCH). |
|---|---|

C57BL6J (Jax, #000664); CD45.1+ C57BL6J (Jax, #002014), RAMP1-/- (Jax, #031560), Rag1-/- (Jax, #002216), OT1 (Jax, #003831), TRPV1cre (Jax, #017769), ChR2fl/fl (Jax, #012567), td-tomatofl/fl (Jax, #007908), DTAfl/fl (Jax, #009669), QuASR2fl/fl (Jax, #028678) mice were purchased from Jackson Laboratory. NaV1.8cre mice were generously supplied by Professor Rohini Kuner (Heidelberg University) and Professor John Wood (UCL). Excluding CD45.1+ mice, all other lines were backcrossed >6 generations on C57BL6/J background.

We used the cre/lox toolbox to engineer the various mice lines used (TRPV1cre::DTAfl/wt, TRPV1cre::QuASR2fl/wt, TRPV1cre::Tdtomatofl/wt, NaV1.8cre::DTAfl/wt, NaV1.8cre::ChR2fl/wt and littermate control) by crossing heterozygote Cre mice with homozygous loxP mice. Mice of both sexes were used in the various cross. All Cre driver lines used were viable and fertile, and abnormal phenotypes were not detected. Offspring were tail-clipped; tissue was used to assess the presence of transgene by standard PCR, as described by Jackson Laboratory or the donating investigator.

| | |
|---|---|
| Wild animals | The study did not involve any wild animals |
| Field-collected samples | The study did not involve any field-collected samples |
| Ethics oversight | The Institutional Animal Care and Use Committees of Boston Children's Hospital and the Université de Montréal (CDEA: #21046; #21047) approved all animal procedures. |

Note that full information on the approval of the study protocol must also be provided in the manuscript.

# Human research participants

Policy information about studies involving human research participants

| | |
|---|---|
| Population characteristics | In compliance with all the relevant ethical regulation and as approved by Sanford Health IRB protocol #640, ten fully de-identified FFPE melanoma blocks were randomly selected for secondary use research specimens. Described below is the clinical characteristics of these specimens:<br><br>DERM103 patient sample was from a malignant melanoma of the shoulder with evidence of dense lymphohiostiocytic inflammatory infiltrate; mitotic index was low, staging was pT1a with negative lymph nodes<br><br>DERM105 patient sample was from a malignant melanoma of the posterior shoulder with slight to moderate inflammatory infiltrate; mitotic index was low, staging was pT2a with negative lymph nodes<br><br>DERM106 patient sample was a malignant melanoma of the thigh with negative lymph nodes. Breslow's thickness 1.2mm and negative lymph nodes. Mitotic index was low, staging pT2a with negative lymph nodes<br><br>DERM107 patient sample was a malignant melanoma from the upper arm, Breslow's thickness 1.6mm, Clark level IV, lymph nodes were negative, margins were clear, mitotic index was high, staging pT2a with negative lymph nodes<br><br>DERM110 patient sample is a malignant melanoma of the arm, Breslow's thickness 0.81 mm, Clark level IV, negative lymph nodes, staging pT1b, pN0, mitotic index high<br><br>DERM112 patient sample is a metastatic malignant melanoma which metastasized to the neck and liver, mitotic index high, staging pT4b with negative lymph nodes.<br><br>DERM113 patient sample is a metastatic malignant melanoma which metastasized to the vulva and liver, mitotic index high, staging pT4b with negative lymph nodes.<br><br>DERM114 patient sample is a metastatic malignant melanoma which metastasized to the shoulder, lung and liver. Mitotic index was high with positive lymph nodes. Staging pT4a.<br><br>DERM115 patient sample is a metastatic malignant melanoma which metastasized to the back and brain. Mitotic index was low with positive lymph nodes, staging pT2a.<br><br>HN480 patient sample is a malignant melanoma of the right temple, Breslow's thickness 7.0 mm, Clark level IV, negative lymph nodes, staging pT4a, mitotic index of 10 |
| Recruitment | No patients were recruited for this study. |
| Ethics oversight | The ten melanoma samples used in this study were collected by Sanford Health and classified by a board-certified pathologist. Their secondary use as research specimen (fully de-identified FFPE blocks) was approved under Sanford Health IRB protocol #640 (Titled: understanding and improving cancer treatment of solid tumors). As part of this IRB-approved retrospective tissue analysis, and in accordance with US Department of Health and human services (HHS) secretary's advisory committee on human research protections, no patient consent was necessary as these secondary use specimens were free of linkers/identifiers and posed no more than minimal risk to human subjects. |

Note that full information on the approval of the study protocol must also be provided in the manuscript.

# Flow Cytometry

## Plots

Confirm that:

☒ The axis labels state the marker and fluorochrome used (e.g. CD4-FITC).

☒ The axis scales are clearly visible. Include numbers along axes only for bottom left plot of group (a 'group' is an analysis of identical markers).

☒ All plots are contour plots with outliers or pseudocolor plots.

☒ A numerical value for number of cells or percentage (with statistics) is provided.

## Methodology

Sample preparation

Immunophenotyping tumor and tumor-draining lymph node. Mice were euthanized when the tumor reached a volume of 800-1500 mm3. Tumors and their draining lymph nodes (tdLN) were harvested. Tumors were enzymatically digested in DMEM + 5% FBS (Seradigm, #3100) + 2 mg/mL collagenase D (Sigma, #11088866001) + 1 mg/mL Collagenase IV (Sigma, #C5138-1G) + 40 ug/mL DNAse I (Sigma, #10104159001) under constant shaking (40 min, 37oC). The cell suspension was centrifuged at 400 g for 5 min. The pellet was resuspended in 70% Percoll gradient (GE Healthcare), overlaid with 40% Percoll, and centrifuged at 500g for 20 min at room temperature with acceleration and deceleration at 1. The cells were aspirated from the Percoll interface and passed through a 70-µm cell strainer.

Tumor-draining lymph nodes were dissected in PBS + 5% FBS, mechanically dissociated using a plunger, strained (70µm), and washed with PBS.

Single cells were resuspended in FACS buffer (PBS, 2% FCS, EDTA), Fc blocked (0.5 mg/mL, 15 min; BD Biosciences, #553141) and stained (15 min, RT) with ZombieAqua (BioLegend, #423102) or (15 min, RT) a Viability Dye eFluor 780 (eBioscience, #65-0865-14). The cells were then stained (30min, 4oC) with either of anti-CD45-BV421 (1:100, BioLegend, #103134), anti-CD45.1-BV421 (1:100, BioLegend, #110732), anti-CD45.2-BV650 (1:100, BioLegend, #109836), anti-CD45-Alexa Fluor 700 (1:100, BioLegend, #103128), anti-CD11b-APC/Cy7 (1:100, BioLegend, #101226), anti-CD8-AF700 (1:100, BioLegend, #100730), anti-CD8-BV421 (1:100, BioLegend, #100753), anti-CD8-PerCP/Cyanine5.5 (1:100, BioLegend, #100734), anti-CD8-Pacific Blue (1:100, BioLegend, #100725), anti-CD4-PerCP/Cyanine5.5 (1:100, BioLegend, #100540), anti-CD4-FITC (1:100, BioLegend, #100406), anti-PD-1-PE-Cy7 (1:100, BioLegend, #109110), anti-Lag3-PE (1:100, BioLegend, #125208), anti-Lag3-PerCP/Cyanine5.5 (1:100, BioLegend, #125212), anti-Tim-3-APC (1:100, BioLegend, #119706), washed and analyzed using a LSRFortessa or FACSCanto II (Becton Dickinson). Antigen specific CD8+ T cells were stained with H-2Kb/OVA257-264 (NIH tetramer core facility) for 15 minutes at 37̊C and were than stained with surface markers.

Intracellular cytokine staining. Cells were stimulated (3h) with phorbol-12-myristate 13-acetate (PMA; 50 ng/mL, Sigma-Aldrich, #P1585), Ionomycin (1 µg/mL, Sigma-Aldrich, #I3909) and Golgi Stop (1:100, BD Biosciences, #554724). The cells were then fixed/permeabilized (1:100, BD Biosciences, #554714) and stained with anti-IFN-γ-APC (1:100, BioLegend, #505810), anti-IFN-γ-FITC (1:100, BioLegend, #505806), anti-TNFα-BV510 (1:100, BioLegend, #506339), anti-TNFα-BV5711 (1:100, BioLegend, #506349), anti-TNFα-PE (1:100, BioLegend, #506306), anti-IL2-Pecy7 (1:100, BioLegend, #503832), anti-IL-2-Pacific Blue (1:100, BioLegend, # 503820), anti-IL-2-BV510 (1:100, BioLegend, #503833), and analyzed using a LSR Fortessa or FACSCanto II (Becton Dickinson).

B16F10 survival. 1x105 B16F10 cells were cultured in 6-well-plate and challenged with BoNT/A (0-50 pg/µL) for 24h, QX-314 (0-1%) for 72h, BIBN4096 (1-4 µM) for 24h or their vehicle. B16F10 cell survival was assessed using anti-annexin V staining and measured by flow cytometry using a LSRFortessa or FACSCanto II (Becton Dickinson).

Drugs impact on CD8+ T-cells function. Splenocytes-isolated CD8+ T-cells from naive C57BL6 mice were cultured under Tc1-stimulating conditions (ex vivo activated by CD3/CD28, IL-12, and anti-IL4) in 24-well plate for 48h. The cells were then exposed to QX-314 (50-150 µM), BoNT/A (10-50 pg/µL) or BIBN4096 (1-4 µM) for 24h. Apoptosis, exhaustion and activation were measured by flow cytometry using a LSRFortessa or FACSCanto II (Becton Dickinson).

In vitro cytotoxic CD8+ T-cell stimulation with CGRP. CD8+ T-cells were isolated and stimulated under Tc1 condition in 96 wells plate. After 48h, cells were treated with either CGRP (0.1 µM) or PBS in the presence of peptidase inhibitor (1 µM) for another 96h. Expression of PD-1, Lag-3, and Tim-3, as well as IFN-γ, TNF-α, and IL-2, was immunophenotyped by flow cytometry using a LSRFortessa or FACSCanto II (Becton Dickinson). Cytokines expression levels were analyzed after in vitro stimulation (PMA/ionomycin; see Intracellular cytokine staining).

In vitro cytotoxic CD8+ T-cell stimulation with neuron-conditioned media. Naive or ablated DRG neurons were cultured (72h) in Neurobasal-A medium supplemented with 0.05 ng/µL NGF (Life Technologies, #13257-019) and 0.002 ng/µL GDNF (PeproTech, #450-51-10). After 48h, the neurobasal medium was removed, neurons were washed with PBS, and 200 µL/well of T-cell media supplemented with 1 µL/mL peptidase inhibitor (Sigma, #P1860) and, in certain cases, capsaicin (1µM) or KCl (50mM) was added to DRG neurons. The conditioned media or vehicle were collected after 30min and added to Tc1 CD8+ T-cells for another 96h. The CD8+ T-cells expression of exhaustion markers (PD-1, Lag-3, Tim-3) and cytokine (IFN-γ, TNF-α, IL-2) were analyzed by flow cytometry using a LSRFortessa or FACSCanto II (Becton Dickinson). Cytokines expressions were analyzed after in vitro stimulation (PMA/ionomycin; see Intracellular cytokine staining).

CD8+ T-cell and DRG neurons co-culture. Naive DRG neurons (2×104) were seeded in a 96-well-plate with T-cell media (supplemented with 0.05 ng/µL NGF (Life Technologies, #13257-019), 0.002 ng/µL GDNF (PeproTech, #450-51-10). One day after, Tc1 CD8+ cells (1×105) were added to the neurons in the presence of IL-2 (BioLegend, #575408). In some instances, co-cultures were stimulated with either capsaicin (1 µM) or KCl (50mM). After 96h, the cells were collected by centrifugation (5

min at 1300 rpm), stained, and immunophenotyped by flow cytometry using a LSRFortessa or FACSCanto II (Becton Dickinson). Cytokines expression were analyzed after in vitro stimulation (PMA/ionomycin; see Intracellular cytokine staining).

OT1 CD8+ T-cells induced B16F10 elimination. 2×104 naive TRPV1Cre::QuASR2-eGFPfl/wt DRG neurons were co-cultured with 1×105 B16F10-mCherry-OVA overnight in T-cell media (supplemented with 0.05 ng/μL NGF (Life Technologies, #13257-019), 0.002 ng/μL GDNF (PeproTech, #450-51-10). One day after, 4 ×105 stimulated OVA-specific CD8+ T-cells under Tc1 condition were added to the co-culture. After 48h, the cells were detached by trypsin (Gibco, #2062476) and collected by centrifugation (5 min at 1300 rpm), stained using anti-Annexin V, 7-AAD (BioLegend, #640930), and anti-CD8 for 20 minutes at 4oC, and were immunophenotyped by flow cytometry using a FACSCanto II (Becton Dickinson). Cytokines expression were analyzed after in vitro stimulation (PMA/ionomycin; see Intracellular cytokine staining).

Neuron's conditioned media impact on OT1 CD8+ T-cells induced B16F10 elimination, 4×105 stimulated OVA-specific CD8+ T-cells were added to 1×105 B16F10-mCherry-OVA and treated with fresh condition media (1:2 dilution). After 48h, cells were stained using anti-Annexin V, 7-AAD (BioLegend, #640930), and anti-CD8 for 20 minutes at 4oC, and were immunophenotyped by flow cytometry using a LSRfortessa or FACSCanto II (Becton Dickinson). For CGRP, 4×105 stimulated OVA-specific CD8+ T-cells were added to 1×105 B16F10- mCherry-OVA and treated with CGRP (100nM). After 24h the cells were stained using anti-Annexin V, 7-AAD (BioLegend, #640930), and anti-CD8 for 20 minutes at 4oC, and were immunophenotyped by flow cytometry using a LSRFortessa or FACSCanto II (Becton Dickinson). Cytokines expression were analyzed after in vitro stimulation (PMA/ionomycin; see Intracellular cytokine staining).

Adoptive transfer of RAMP1wt or RAMP1-/- CD8 T-cells. Total CD8+ T-cells were isolated from the spleen of wild-type (CD45.1+) or RAMP1-/- (CD45.2+) mice, expanded and stimulated in vitro using a mouse T-cell Activation/Expansion Kit (Miltenyi cat #130-093-627). CD8+ cells from RAMP1-/- and RAMP1wt were injected separately or 1:1 mix through tail vein of RAG1-/- mice. One week after, the mice were inoculated with B16F10-mCherry-OVA cancer cells (5×105 cells; i.d.). On day 10, tumors were harvested and RAMP1-/- (CD45.2+) and RAMP1wt (CD45.1+) CD8+ T-cells were immunophenotyped using a FACSCanto II (Becton Dickinson) or FACS-purified using a FACSAria IIu cell sorter (Becton Dickinson).

Adoptive T-cell transfer in RTX-exposed mice. CD8+ T-cells were isolated from OT-1 mice spleens and magnet-sorted (StemCell; #19858). Naïve CD8+ T-cells (CD8+CD44lowCD62Lhi) cells were then FACS-purified using a FACSAria IIu cell sorter (Becton Dickinson) and injected (1x106 cells; i.v., tail vein) to vehicle- or RTX-exposed RAG1-/- mice.

RNA sequencing of triple co-culture and data processing. 1×104 naive TRPV1Cre::QuASR2fl/wt DRG neurons were co-cultured with 1×105 B16F10-mCherry-OVA overnight in T-cell media (supplemented with 0.05 ng/μL NGF (Life Technologies, #13257-019), 0.002 ng/μL GDNF (PeproTech, #450-51-10). One day after, 4 ×105 stimulated OVA-specific CD8+ T-cells under Tc1 condition were added to the co-culture. After 48h, the cells were detached and TRPV1 neurons (CD45- eGFP+ mCherry-), B16F10-mCherry-OVA (CD45- eGFP- mCherry+), and OVA-specific CD8+ T-cells (eGFP- mCherry-CD45+ CD3+ CD8+) were FACS-purified using a FACSAria IIu cell sorter (Becton Dickinson) prior to sequencing.

RNA sequencing of cancer and neurons co-culture and data processing. 1×104 naive TRPV1Cre::QuASR2-eGFPfl/wt DRG neurons were co-cultured with 5×104 B16F10-mCherry-OVA overnight in complete Dulbecco's Modified Eagle's Medium high glucose (DMEM, Corning, #10-013-CV) supplemented with 10% fetal bovine serum FBS, Seradigm, #3100), 1% penicillin/ streptomycin (Corning, #MT-3001-Cl), 0.05 ng/μL NGF (Life Technologies, #13257-019), 0.002 ng/μL GDNF (PeproTech, #450-51-10). After 48h, the cells were detached and TRPV1 neurons (eGFP+ mCherry-), and B16F10-mCherry-OVA (eGFP-mCherry+) were FACS-purified using a FACSAria IIu cell sorter (Becton Dickinson) prior to sequencing.

| Instrument | Cells were immunophenotyped using a LSRFortessa (Becton Dickinson) or FACSCanto II (Becton Dickinson).<br>Cells were purified using a FACSAria IIu cell sorter (Becton Dickinson) |
| --- | --- |
| Software | Data were analyzed using FlowJo v10.0.0 software (Tree Star) |
| Cell population abundance | For sorting experiments (i.e. CD8, neurons or B16F10 cells), flow cytometry was used and >96% purity was achieved |
| Gating strategy | Relevant gating strategy are provided as extended data figure 6A, 9F, 10W<br><br>To gate samples for FACS analysis, cell were initially gated by FSC-A vs SCS for the exclusion of debris and identification of relevant population (lymphocytes) by size and granularity. For single cells, samples were further gated by FSC-A vs FSC-H. Live cells were finally gated and identified by using fixable dye or fluorescent reporters. |

☒ Tick this box to confirm that a figure exemplifying the gating strategy is provided in the Supplementary Information.

