## [Peer Review File · Nature]

Manuscript Title: Nociceptor neurons affect cancer immunosurveillance **Reviewer Comments & Author**

Rebuttals

Reviewer Reports on the Initial Version:

Referees' comments:

Referee #1 (Remarks to the Author):

Sensory neurons are known to regulate tumor growth via different mechanisms. In this study, the authors demonstrated that nociceptors promote melanoma growth via CGRP release. In particular, the data showed that nociceptor activation and CGRP release cause exhaustion of cytotoxic CD8+ T cells (PD1+/Lag3+/Tim3+/IFN γ -), limiting their capacity to suppress melanoma growth. The authors used multiple loss-of-function approaches (genetic NaV1.8 or TRPV1 lineage ablation, nerve blockade with Botox and QX-314) and gain-of-function approach (optogenetic activation of nociceptors) to support a critical role of nociceptors in enhancing melanoma growth and T cell tolerance. They also validated the mechanism using adoptive transfer of CD8+ T-cells with or without RAMP1 expression. Finally, the authors showed in melanoma patients, survival rates are associated with different expression levels of RAMP1 and neural genes in skin biopsies.

Overall, the authors have spent a tremendous experimental effort by including 49 supplementary figures in the study, and the findings are significant with translational relevance. Neuro-immune interactions through nociceptor CGRP release have been demonstrated by many groups, but CRGP regulation of CD8+ T cells and immune tolerance through RAMP1 signaling is novel. The manuscript could be improved by including additional control experiments and addressing the following issues.

1. The authors highlighted hyper-innervation of TRPV1+ neurons in patients' melanoma. However, the morphology of nerve fibers is not so convincing (Supplementary Figure 1). It is unclear if these dots are nerve terminals or cellular staining. TRPV1 may also be expressed in non-neuronal cells. Even these cells show negative mRNA expression (Supplementary Figure 5, 44, 45, 46, 47, 48), they can still be positive for TRPV1 protein staining. Protein expression and mRNA expression may not be correlated; and protein expression is certainly more relevant to the function of TRPV1. Do these cells respond to capsaicin?
2. Please consider additional characterization of RAMP1 expression in T cells. Page 5: In human melanomas, only 1% of tumor-infiltrating CD8+ T-cells expressed RAMP1 (ref 44). This ratio is quite low. It is important to demonstrate RAMP1 protein expression in CD8+ T cells. This could be tested in WT and KO mice.
3. Please also measure CGRP secretion in melanoma cells in addition to Calca expression, because the mRNA expression may not be correlated with the peptide expression and release.
4. Supplementary Fig. 1F: It is surprising not to see TRPV1 staining in the intact skin.

5. Fig. 1A: Density of nerve innervation should be quantified as number of epidermal nerve fibers. Fig. 1A and 1B: Control images are missing. Low magnification image will be helpful. The orientation of skin should be indicated to show the projection and branches of nerves.

6. Sample sizes and statistic analyses: Figure 1 legend: It is unclear if the sample size indicates number of animals, number of cultures, or number of cells. Fig. 1F and 1G: Sample sizes for statistical analyses of calcium imaging should be based on number of cultures not number of cells. Fig. 3F: $n=4-6$ /group. Do you mean 4-6 mice per group? This appears to be a small sample size for cancer research. Fig. 4E: $n=11-89$ /groups. Did you use 89 animals per group, why?

7. Cell ablation approach in general will boost immunity. Ablation of non-peptidergic neurons could be a great control, if possible.

8. The clinical significance could be limited by only testing pre-treatment or early-phase treatment in this study. Post-treatment with nerve blockade or ablation has higher clinical relevance.

Other comments:

Fig. 2E: The effect of CGRP is mild as compared to capsaicin. This may argue for CGRP-independent mechanism.

Fig. 4M: Are these changes for RAMP1 (2.0-fold), PD1 (1.7-fold), Lag3 (1.6-fold), Ctl4 (1.6-fold), and Tim3 (1.7-fold) statistically significant?

Supplementary Figure 42A: The tumor size is enormous; it is several times larger than a hind paw. It will be difficult to assess pain and paw withdrawal under this extreme condition. I wonder if you measure a time course of pain in. Metastasis to lung and brain tissues should also be robust in these animals, which is a major cause of death.

Discussion: "Silencing tumor-innervating sensory neurons represents therefore a strategy for attenuating the immunomodulatory action of the nervous system, and promoting robust anti-tumor activity of the immune system (Supp. Fig. 49)". This is a strong statement, but clinical relevance will be enhanced by post-treatment in the melanoma model.

Capsaicin was used to in vitro studies. Why not for in vivo study to activate nociceptors? Should melanoma patients eat less spicy food?

Referee #2 (Remarks to the Author):

The manuscript entitled "Nociceptor neuron impair cancer immunosurveillance" presented by Balood, et al. describes the relationship and crosstalk between the nociceptor neuron and tumor cells in melanoma models. The authors are demonstrating this by performing a well-designed series of experiments that shows the innervation of nociceptors into the tumor as well as the exhaustion of cytotoxic CD8 T cells by the secretion of SLPI and CGRP.

All the findings are presented in a logical way to generate a well-organized description of the investigation. This manuscript provides important and novelty findings in the already complex tumor

microenvironment research field by proposing the immunomodulation of the nervous system to improve the anti-tumor activity of the immune system.

Major concern. The discussion of the findings can be improved. The results are not well discussed in terms of similar reports.

Minor concern. Second page, last paragraph: must say 10-10000 pg/mL not ng/mL.
Some supplementary figures can be consolidated, for example, supplementary figures 6-12.

Referee #3 (Remarks to the Author):

Balood argue that many tumors establish a significant degree of de novo connections with the nervous system and that tumor denervation can affect tumor progression. Based on this, they began to systematically investigate how the infiltration of tumors with nociceptors and how their associated signaling pathways impact tumor progression.

Overall, they present an impressive study. They found that ablation of nociception significantly slows tumor progression, but most surprisingly, nociception and its associated neurotransmitters were identified to have a very profound effect on the phenomenon of T cell exhaustion. Here, compelling evidence is presented that co-culturing activated T cells with capsaicin-stimulated neurons significantly alters the T cell phenotype, but this effect only occurs when nociceptor neurons are present in the co-cultures. While, as an immunologist, I am less familiar with the neurological aspects of this project, I can clearly say that the authors' immunological observations are quite substantial. It is important to keep in mind that the induction of an exhausted phenotype in ex vivo cultures is still a largely unachieved goal. What has always been puzzling is that T cell exhaustion is rapidly induced in vivo but not in vitro cultures. The findings made by the authors could mean that nociception and interactions with neuropeptides could be the missing link for inducing T cell exhaustion.

Despite these very positive assessments, there are some significant structural, as well as scientific, issues associated with the manuscript.

Major points:

1. Presently, the manuscript is clearly overloaded with data. I appreciate the authors' decision to present the findings in as much detail as possible, but 49 supplemental figures is a very substantial number and I assume over the limits of nature.

2. at the same time, the massive amount of data precludes the main findings from being adequately explained and reflected by the authors. In fact, I think up to 50% of the data could be removed from the manuscript or labeled as not shown without significantly affecting the core message. So the

challenge is to streamline the manuscript.

3. In this context, it is sometimes difficult to reconcile the authors' decisions about which data to show in the main figures and which to show in the supplementary section. Fig. 1 does not provide any significant new insights and is rather technical in nature. At the same time, the impressive phenotype of T cells in tumors with and without nociception is hidden in Supp.Fig. 14. At the same time, Figs. 3 and 4 could be merged and filtered.

4. It would be very informative if the authors could establish further correlations between the level of nociception that is found in tumors and the level of T cell exhaustion.

5. The degree of T cell exhaustion is somewhat under-explored. I mean, the phenotype is very interesting, but additional signatures such as transcriptional profiles, i.e., Tox, Eomes, Tbet, ID2/ID3, Nr4a profiles would be helpful. Broader gene expression characterization of nociceptor exposed and unexposed T cells would be informative.

6. It would be interesting to see how the absence of nociception synergizes with checkpoint inhibition.

7. Kinetic characterization of the effect of nociception on T cell differentiation in vivo in tumors (i.e. when is it required) would be very informative

Minor parts:

- Abstract is loaded with specific and very condensed information. More conceptual focus and less detailed information would be helpful.

- Since this manuscript bridges the two fields of neurology and immunology, it would be very helpful to clarify some of the abbreviations. In several places, nomenclature and mice are used that are not defined, i.e., terms such as Nav1.8^{cre::Td-tomato}/wt mice and iDISCO are used without explanation.

- Tc1 cultured T cells is not standard nomenclature in mice and should be replaced by ex vivo CD3/CD28 activated T cells.

- The use of truly naive (CD44^{low}) OT1 or similar TCR transgenic cells in the tumor experiments would be helpful.

- Sometimes authors use an unconventional FACS axis labelling. The axis often combines staining and gating information, which should be strictly separated.

- Figure 1A: The authors use the plots to show the degree of infiltration of nociceptor-positive neurons. Since only one condition is shown, it remains a judgment call by the authors that this represents a significant level of nociception. I am saying this as others have pointed out that

melanomas have lower nociception than normal tissues. ([https://www.jidonline.org/article/S0022-202X\(18\)31748-2/fulltext](https://www.jidonline.org/article/S0022-202X(18)31748-2/fulltext)).

- Supp.Fig. 1G and H: is insufficiently explained and it is unclear what is being shown.
- Supp.Fig. 5B: the meaning of the display item is unclear.
- Supp.Fig.6 and following FACS figures: they are too small and not readable after printing.
- Supp.Fig.6A: why are different gates used in Life Dead staining in the two plots shown,
- Supp.Fig.7 is difficult to interpret. I assume the middle Colum (KCI-Stim, Neuron CM) is the experimental group and the right one is the control group, but the labeling is confusing and inconsistent between the figure, legend and text.
- Supp.Figs. 7-9: Of concern, the phenotypes shown here are less clear than in Supp.Fig. 6. For example, PD-1 expression looks similar between the middle experimental group and the left vehicle group in Supp.Fig. 7, and clear differences between TNFa and IFNg are difficult to discern.
- The authors used antiCD3 depletion, which is uncommon in mice, it would be helpful to repeat these experiments with antiCD8 depletion

Author Rebuttals to Initial Comments:

Referee #1: sensory inputs pain, immune system links

Sensory neurons are known to regulate tumor growth via different mechanisms. In this study, the authors demonstrated that nociceptors promote melanoma growth via CGRP release. In particular, the data showed that nociceptor activation and CGRP release cause exhaustion of cytotoxic CD8⁺ T cells (PD1⁺/Lag3⁺/Tim3⁺/IFN γ ⁻), limiting their capacity to suppress melanoma growth. The authors used multiple loss-of-function approaches (genetic NaV1.8 or TRPV1 lineage ablation, nerve blockade with Botox and QX-314) and gain-of-function approach (optogenetic activation of nociceptors) to support a critical role of nociceptors in enhancing melanoma growth and T cell tolerance. They also validated the mechanism using adoptive transfer of CD8⁺ T-cells with or without RAMP1 expression. Finally, the authors showed in melanoma patients, survival rates are associated with different expression levels of RAMP1 and neural genes in skin biopsies.

Overall, the authors have spent a tremendous experimental effort by including 49 supplementary figures in the study, and the findings are significant with translational relevance. Neuro-immune interactions through nociceptor CGRP release have been demonstrated by many groups, but CGRP regulation of CD8⁺ T cells and immune tolerance through RAMP1 signaling is novel. The manuscript could be improved by including additional control experiments and addressing the following issues.

We would like to thank the reviewer for the positive review.

1.1. The authors highlighted hyper-innervation of TRPV1⁺ neurons in patients' melanoma. However, the morphology of nerve fibers is not so convincing (Supplementary Figure 1). It is unclear if these dots are nerve terminals or cellular staining.

Patients' malignant melanoma cells (defined as CD90⁺CD45⁻) do not express the transcripts encoding for TRPV1 or Tubb3, as shown by the in-silico analysis of ten different patients' biopsies (Supp. Figure 1A). As such, the signal we measured by histology (Supp. Figure 2) is likely ascribed to tumor neo-innervation. In support of this hypothesis, we used IF and confirmed that the TRPV1 signal measured by IHC (Supp. Figure 2) is co-expressed with the neuronal marker Tubb3 (Rebuttal Figure 1; as shown by yellow color overlay on the left panels). In addition, four metastatic melanoma patients' primary tumor biopsies were stained with S100 (malignant melanoma cell marker; PMID: 9187834), Tubb3 (neuronal marker; PMID: 1527798) and TRPV1 (nociceptor neuron marker; PMID: 9349813). Although the presence of TRPV1⁺ nerve twigs is sparse, we can appreciate the presence of TRPV1/Tubb3⁺ cells surrounding the malignant cells (S100⁺; Rebuttal Figure 1; right panels).

1.2. Supplementary Fig. 1F: It is surprising not to see TRPV1 staining in the intact skin.

A high-power image of the patient's biopsy confirms the presence of TRPV1 and Tubb3 positive nerve terminals in intact skin (**Rebuttal Figure 2**; blue arrows in bottom panel).

1.3. TRPV1 may also be expressed in non-neuronal cells. Even these cells show negative mRNA expression (Supplementary Figure 5, 44, 45, 46, 47, 48), they can still be positive for TRPV1 protein staining. Protein expression and mRNA expression may not be correlated; and protein expression is certainly more relevant to the function of TRPV1. Do these cells respond to capsaicin?

Single cell RNA sequencing, bulk RNA sequencing and microarray data confirmed that mouse and human immune cells or malignant cells do not express TRPV1 (**Supp. Figures 1, 3**). To further address this, we show that:

- Capsaicin does not impact B16F10 cell survival (**Rebuttal Figure 3**)
- Cultured CD8 T-cells failed to respond to capsaicin (**Supp. Figure 7**)
- We used single-cell RNA sequencing and CITEseq to profile B16F10 cell-draining lymph nodes. In the absence of TRPV1⁺ neurons, we found drastic differences in the number and transcriptomes of TdLN lymphocytes and myeloid cells. In intact mice, the ablation of TRPV1 had absolutely no impact on the transcriptome and number of lymph node lymphocytes and myeloid cells (**Rebuttal Figure 4**). These data confirm that TRPV1 is not expressed in immunocytes or in the cells composing the draining lymph node.
- CD8⁺ T-cells or B16F10 cells failed to respond to QX-314, whose activity depends on the presence of activated TRP channels (**Supp. Figure 6**).

In support of TRPV1 specificity for sensory neurons:

1. Mathur et al., JCI Insight. PMID: 34727095
 - Supplementary Figure 1E: In-silico analysis of single-cell RNA-sequencing of mouse lung immune cells revealed high transcript levels of CD3e, but no expression of TRPV1 or Na_v1.8.
2. Crosson et al., JACI. 2021. PMID: 33453289
 - Supplementary Figure 19: Representative flow cytometry gating show that DTA-mediated ablation of TRPV1⁺ cells (TRPV1^{cre}DTA^{l/wt}) doesn't not impact the number of peritoneal immune cells when compared to TRPV1-expressing mice (TRPV1^{wt}DTA^{l/wt}).
 - Supplementary Figure 20: Using TRPV1^{cre} as a reporter (TRPV1^{cre}Td-tomato^{l/wt}), we immunophenotyped td-tomato expressing cells in dorsal root ganglion, peritoneal and lung CD45⁺ cells and confirmed that TRPV1 expression was limited to DRG nociceptor neurons.

3. Cohen et al. Cell. 2019; PMID: 31353219
 - In Supplementary Figure 1, the authors used the TRPV1^{cre} mouse line and found an absence of *cre* expression in skin immunocytes
4. Baral et al., Nature Medicine. 2018; PMID: 29505031
 - In Supplementary Figure 18, the authors showed ImmGen expression data and also concluded in the absence of TRPV1 in immune cells.
 - In Supplementary Figure 20, the authors used qPCR to confirm that immune cells do not express TRPV1.

Published literature using TRPV1^{cre} mice to target nociceptors:

1.	Baral et al.,	Nature Medicine	2018	PMID: 29505031
2.	Cohen et al.,	Cell	2019	PMID: 31353219
3.	Lai et al.,	Cell	2019	PMID: 31813624
4.	Talbot et al.,	JACI	2020	PMID: 31954778
5.	Foster et al.,	Frontiers in Immunology	2017	PMID: 29163530
6.	Trankner et al.,	PNAS	2014	PMID: 25049382
7.	Hoon et al.,	Cell Reports	2019	PMID: 30917312
8.	Michoud et al.,	Nature Biotechnology	2020	PMID: 32958958
9.	Cobos et al.,	Cell Reports	2018	PMID: 29386116
10.	Mathur et al.,	JCI Insights	2021	PMID: 34727095

[Redacted figure]

1.4. Fig. 1A: Density of nerve innervation should be quantified as number of epidermal nerve fibers. Fig. 1A and 1B: Control images are missing. Low magnification image will be helpful. The orientation of skin should be indicated to show the projection and branches of nerves.

The investigator who originally performed the iDISCO experiment (Fig. 1A) has now left Cygnal Therapeutics. Nevertheless, we used whole mount skin to quantify Nav1.8^+ neuron density in naïve $\text{Nav1.8}^{\text{Cre}}::\text{Td-tomato}^{\text{I/wt}}$ mice. As we found to be the case in B16F10-inoculated $\text{TRPV1}^{\text{Cre}}::\text{Td-tomato}^{\text{I/wt}}$ mice (Rebuttal Figures 5A-B), tumor inoculated $\text{Nav1.8}^{\text{Cre}}::\text{Td-tomato}^{\text{I/wt}}$ mouse skin shows marginally higher nerve terminal density (~11% of Td-tomato^+ cells/ mm^2 ; Rebuttal Figure 5C) than that of naïve mice (~9% of Td-tomato^+ cells/ mm^2 ; Rebuttal Figure 5D).

Of note, we directly assess the impact of cancer cells on nociceptor neuron neurite outgrowth in culture – which allowed us to precisely control the number of cells plated, the timing and the quantification of the neurites (Fig. 1C-D). Rather than the possibility of tumor-inoculated skin to show higher nerve density, we modified the description of these data to emphasize that sensory neurons densely infiltrate the tumor core. Such findings are the biological equivalent to tumor neo-angiogenesis.

The experiment in Figure 1B was carried out *in vitro*; and quantification is showed in Figure 1C-D. TRPV1^+ neurite outgrowth was increased when neurons were co-cultured with B16F10 cells, in contrast to when they were cultured with non-tumorigenic keratinocytes (Fig. 1C). Of note, TRPV1 arborisation was decreased in the nociceptor-B16F10 cell co-culture (Fig. 1D).

Representative pictures ($n=3/\text{group}$) are shown in Rebuttal Figure 5E-F. To abide by Nature's supplementary figure limit, we opted not to include these panels in the revised manuscript.

Rebuttal Figure 5

1.5. Please consider additional characterization of RAMP1 expression in T cells. Page 5: In human melanomas, only 1% of tumor-infiltrating CD8^+ T-cells expressed RAMP1 (ref 44). This ratio is quite low. It is important to demonstrate RAMP1 protein expression in CD8^+ T cells. This could be tested in WT and KO mice.

Single-cell RNA sequencing, bulk RNA sequencing and microarray data confirm that mouse and human CD8^+ T-cells express RAMP1 (Supp. Fig. 1, Supp. Fig. 3). Along with this sequencing data, we have also confirmed that RAMP1 protein is enriched in CD4^+ and CD8^+ T-cells [redacted]. At the tissue level, we found that RAMP1 is more expressed in wildtype mouse spleen than in the brain (Rebuttal Figure 6B). These findings are in accordance with the Human Protein Atlas data which show enriched RAMP1 transcript expression in human skin and spleen T-cells (Rebuttal Figure 6C-D).

In relation to reference 44, the patient's single cell RNA sequencing we analyzed *in-silico* had a gene coverage of ~10%. This means that only the top-10% most expressed genes were detected and this may explain the limited number of RAMP1^+ CD8^+ T-cells detected in the tumors.

- In our own unpublished single-cell RNA sequencing from tumor-draining lymph nodes, we found that in nociceptor intact mice, ~28% of B16F10-infiltrating CD8^+ T-cells express RAMP1. Interestingly, in nociceptor-ablated mice, slightly more B16F10-infiltrating CD8^+ T-cells were present (3490 vs 3400), but only ~1% express RAMP1 (Rebuttal Figure 6E).

[Text redaction (above)]

- The RAMP1 agonist CGRP increases CD8⁺ T-cell exhaustion and decreases their cytotoxic potential. This effect was absent in CD8⁺ T-cells harvested from Ramp1^{-/-} mice (**Fig. 4A**; **Supp. Fig. 7E-F**).
- When we rescued CGRP in nociceptor ablated mice, we found that CD8⁺ T cells were more exhausted and less cytotoxic (**Fig. 4D-E**).
- In RAG1^{-/-} mice, we found that Ramp1^{-/-} CD8 T-cells are more exhausted and less cytotoxic than Ramp1^{wt} CD8 T-cells. We confirmed these findings when co-transplanted in the same mice (**Fig. 4H-J**). We also made similar observation when we FACS-purified these cells from tumor-bearing mouse and analyzed their transcriptome using RNA-sequencing (**Fig. 4K**).

Of note, Assas and colleagues (PMID: 24592205) reviewed *Ramp1* expression on CD8⁺ T-cells and mentioned:

"CGRP was described as a neuropeptide with "chemotactic" properties on human CD4 and CD8 T lymphocytes in the skin, inducing T cell trafficking (Foster et al., 1992). Ever since, T cells more than any other cell, have been the main targets for studies to understand the effect of CGRP on immune cell migration, adhesion, and motility"

"CGRP plays a key role in T cell adhesion to fibronectin (Levite, 1998; Levite et al., 1998) and beta integrin mediated T cell migration. Somatostatin, CGRP and neuropeptide Y all induced of freshly purified T cells to fibronectin coated plates (Springer et al., 2003) "

"In the gut CGRP from c fibers stimulates T cell migration (Talme et al., 2008). In vitro studies showed that in contrast to other neuropeptides, CGRP can stimulate the migration of CD3 T cells into collagen matrix (Talme et al., 2008), an effect inhibited with CGRP receptor antagonist"

"CGRP binding to T cells, through its receptor, opens the voltage-gated K(+) channels, releasing K(+) from the intracellular matrix and activating β1 integrin (Levite et al., 2000). This facilitates T cell integrin-induced function highlighting a critical role for CGRP in alternative pathways for T cell adhesion, migration, and motility"

• Teresi.	Immunol Lett.	1996	PMID: 8793567
• Assas.	Front Neurosci.	2014	PMID: 24592205
• Foster	Ann N Y Acad Sci	1992	PMID: 1637095
• Levite	PNAS	1998	PMID: 9770522
• Levite	Ann N Y Acad Sci	2000	PMID: 11268358
• Levite	Curr Opin Pharmacol	2008	PMID: 18579442
• Levite	J Immunol	1998	PMID: 9551939
• Springer	Pulm Pharmacol Thr	2003	PMID: 12749828
• Talme	J Neuroimmunol	2008	PMID: 18423624

1.6. Please also measure CGRP secretion in melanoma cells in addition to Calca expression, because the mRNA expression may not be correlated with the peptide expression and release.

B16F10 cells do not release CGRP (**Fig. 1G**) and do not express *Calca* (**Supp. Fig. 3A**).

Of note:

- Tumor-exposed neurons overexpress *Calca* (gene encoding CGRP; **Fig. 1H-J**).
- Cultured DRG neurons release CGRP when exposed to B16F10 cells (**Fig. 1G**) or B16F10 cell-released SLPI (**Fig. 2J**).
- When stimulated with capsaicin, tumor-surrounding skin explant harvested from B16F10-inoculated mice showed increased CGRP release (**Supp. Fig. 6G**).

1.7. Sample sizes and statistical analyses: Figure 1 legend: It is unclear if the sample size indicates number of animals, number of cultures, or number of cells. Fig. 1F and 1G: Sample sizes for statistical analyses of calcium imaging should be based on number of cultures not number of cells. Fig. 3F: n=4-6/group. Do you mean 4-6 mice per group? This

Rebuttal Figure 6

appears to be a small sample size for cancer research. Fig. 4E: n=11-89/groups. Did you use 89 animals per group, why?

We have amended the text and statistics.

For the α CD3 experiment (**Supp. Fig. 4F**), we showed one representative experiment out of the three we performed, which had an n=4-6 group. Of note, we have now used α CD8 to deplete CD8 T-cells and confirm T-cells' key role in decreased tumor growth observed in sensory neuron ablated mice (**Fig. 3I**).

For the survival experiment (**Supp. Fig. 6J**), we initially plotted one survival graph comparing B16F10-injected mice treated with (i) vehicle, (ii) QX-314, (iii) BoNT/A or (iv) BIBN4096. Because we used the same protocol in all the B16F10-exposed mice treated with vehicle, we pooled all the vehicle-exposed mice from six experiments (n=89 mice). Specifically, in the experiments carried out on B16F10-injected mice treated with QX-314 or vehicle, we found that silencing tumor-innervating nociceptors increased mouse survival by ~2.0-fold (n=22 mice; hazard ratio: 0.45; **Rebuttal figure 7**).

Rebuttal Figure 7

1.8. Cell ablation approach in general will boost immunity. Ablation of non-peptidergic neurons could be a great control, if possible.

At four weeks of age, $Mrgd^{CreER}DTA^{l/wt}$ (MrgD neuron-ablated) and $Mrgd^{W}DTA^{l/wt}$ (neuron intact) mice were exposed to tamoxifen for 5 consecutive days. Two weeks later, the mice were inoculated with B16F10-OVA and the impact of MrgD neuron ablation was assessed on tumor growth and on intra-tumoral CD8⁺ T-cells exhaustion. We found that the systemic ablation of Mrgd⁺ neurons had no impact on B16F10 tumor growth or on intra-tumoral CD8 T-cell exhaustion (**Rebuttal figure 8**).

Rebuttal Figure 8

In addition to pharmacological tools (rCGRP, BIBN4096, QX-314, BoNT/A, RTX), we found that ablation of TRPV1 (**Supp. Fig. 4A**) decrease tumor growth. Using optogenetics to stimulate light-sensitive nociceptor neurons ($Nav1.8^{Cre}::ChR2^{l/wt}$), we found that the local activation of $Nav1.8^{+}$ tumor-innervating neurons increases tumor growth (**Supp. Fig. 4G**). In support of these findings, we also showed that the chemoablation of TRPV1⁺ neurons using RTX results in smaller tumor growth in these animals (**Fig. 3J**). Similarly, naïve OT-1 CD8 T-cells enhance tumor shrinkage (**Fig. 3K**) and show limited exhaustion (**Fig. 3L**) when transplanted in $RAG1^{-/-}$ mice whose nociceptor neurons were chemically depleted with RTX. Altogether, these data indicate that the slower tumor growth found in $TRPV1^{Cre}::DTA^{l/wt}$ or in RTX-exposed mice depends on the relief from peptidergic nociceptor-induced CD8 exhaustion.

1.9. The clinical significance could be limited by only testing pre-treatment or early-phase treatment in this study. Post-treatment with nerve blockade or ablation has higher clinical relevance.

To address this, we compared the effects of BIBN4096 (n=15/group; 3 independent experiments), QX-314 (n=15/group; 3 independent experiments), and BoNT/A (n=15/group; 3 independent experiments) when used either as a prophylactic (started the same day as tumor inoculation, or 1-3 days prior to) or as a therapeutic (treatment started when tumor volume reached ~200mm³).

BoNT/A preserved CD8 T-cells cytotoxic potential and reduced tumor growth when used as a prophylactic treatment. Although BoNT/A had no effect when used as a therapeutic, it increased aPDL1-mediated tumor regression (**Supp. figure 5**). In contrast, QX-314 preserved CD8 T-cells cytotoxic potential, prevented their exhaustion, and reduced tumor growth when used both as a prophylactic and as a therapeutic treatment. QX-314 increase B16F10-inoculated mouse survival and increased the clinical response to aPDL1 (**Supp. figure 6**). BIBN4096 showed similar anti-tumor efficacy when used as a therapeutic or as prophylactic (**Supp. figure 8**).

Other comments:

1.10. Fig. 2E: The effect of CGRP is mild as compared to capsaicin. This may argue for CGRP-independent mechanism.

We agree with this interpretation and after investigating other possible mechanisms, we found that tumor-innervating neurons-released galanin and increased CD8 T-cell exhaustion.

Along with a raised in *Calca* expression, we found that tumor-innervating neurons upregulate Galanin (Fig. 1H-J) and that CD8 T-cells express basal levels of galanin receptors (ImmGen).

We tested the pro-exhaustion effects of galanin on cultured cytotoxic CD8 T-cells. Along with CGRP (Supp. figure 7), galanin also drives the expression of immune checkpoint receptors on cultured cytotoxic CD8 T-cells (Rebuttal Figure 9A-C).

In patient biopsies, upregulation of galanin – as determined by bulk RNA-sequencing – correlates with reduced survival (Supp. figure 9J; Rebuttal Figure 9D). Of note, we found no effects of Substance P on any of the tested parameters (Rebuttal Figure 9E-G).

1.11. Fig. 4M: Are these changes for RAMP1 (2.0-fold), PD1 (1.7-fold), Lag3 (1.6-fold), Ctla4 (1.6-fold), and Tim3 (1.7-fold) statistically significant?

The differences are statistically significant. We have amended the graph to include the statistics (Supp. figure 9N-P)

1.12. Supplementary Figure 42A: The tumor size is enormous; it is several times larger than a hind paw. It will be difficult to assess pain and paw withdrawal under this extreme condition. I wonder if you measure a time course of pain in. Metastasis to lung and brain tissues should also be robust in these animals, which is a major cause of death.

To assess this, we implanted B16F10-mCherry-OVA cells (i.d., 5×10^5 cells) into the hindpaw of 8-week-old male and female mice (n=75). We found that enhanced tumor volume led to increased thermal pain hypersensitivity and increased intra-tumoral CD8 T-cell exhaustion (rebuttal figure 10A-B). The heightened thermal hypersensitivity positively correlated ($R^2=0.52$, $p<0.0001$) with exacerbated CD8 T-cell exhaustion (Fig. 3A, rebuttal figure 10C; as measured on day 15 post implantation).

1.13. Discussion: “Silencing tumor-innervating sensory neurons represents therefore a strategy for attenuating the immunomodulatory action of the nervous system and promoting robust anti-tumor activity of the immune system (Supp. Fig. 49)”. This is a strong statement, but clinical relevance will be enhanced by post-treatment in the melanoma model.

We agree with the reviewer and have nuanced the text.

As detailed in point 1.9, we have now demonstrated the efficacy of post-treatment with QX-314 (Supp. figure 6), and BIBN4096 (Supp. figure 8). As opposed to BoNT/A (Supp. figure 5), we found that therapeutic (beginning when tumor volume was $\geq 200\text{mm}^3$) silencing tumor-innervating neurons with QX-314 or the CGRP-RAMP1 axis using BIBN decreases tumor size and prevents CD8 T-cell exhaustion (Supp. figure 6; Supp. figure 8).

Rebuttal Figure 9

Rebuttal Figure 10

1.14. Capsaicin was used to in vitro studies. Why not for in vivo study to activate nociceptors? Should melanoma patients eat less spicy food?

A 2015 study published in BMJ (PMID: 26242395) performed on ~500 000 healthy individuals showed that the consumption of spicy food (6 - 7 times/week) reduced the risk of cancer death (OR: 0.73 (0.64 - 0.84)). In a subgroup analysis (consumption of spicy foods ≥ 6 days a week), the reduction in cancer death was higher in participants who reported eating fresh chili peppers (OR: 0.89 (0.81 - 0.97)) than those consuming non-fresh peppers (OR: 0.99 (0.87 - 1.12)).

In addition to activating TRPV1 expressing sensory neurons, capsaicin can also desensitize these sensory neurons through repeated exposure or lead to their chemoablation at high concentrations.

To avoid these complications, we used optogenetic to activate $\text{Na}_v1.8^+$ tumor-innervating nociceptor neurons and found that light activation drives tumor growth when started (i) once tumors were visible ($\sim 20\text{mm}^3$) or (ii) when tumors were established ($\sim 200\text{mm}^3$; **Supp. figure 4G**; **rebuttal figure 11A-B**). This is accompanied by a reduced capacity of intratumoral CD8 T-cells to proliferate (**rebuttal figure 11C**).

Rebuttal Figure 11

Referee #2: melanoma

The manuscript entitled “Nociceptor neuron impair cancer immunosurveillance” presented by Balood, et al. describes the relationship and crosstalk between the nociceptor neuron and tumor cells in melanoma models. The authors are demonstrating this by performing a well-designed series of experiments that shows the innervation of nociceptors into the tumor as well as the exhaustion of cytotoxic CD8 T cells by the secretion of SLPI and CGRP.

All the findings are presented in a logical way to generate a well-organized description of the investigation. This manuscript provides important and novelty findings in the already complex tumor microenvironment research field by proposing the immunomodulation of the nervous system to improve the anti-tumor activity of the immune system.

We would like to thank the reviewer for the positive review.

2.1. Major concern. The discussion of the findings can be improved. The results are not well discussed in terms of similar reports.

We have streamlined the discussion with a specific focus on the CGRP-RAMP1 axis.

2.2. Minor concern. Second page, last paragraph: must say 10-10000 pg/mL not ng/mL.

We have amended the text.

2.3. Some supplementary figures can be consolidated, for example, supplementary figures 6-12.

We have streamlined the supplementary figures into 10.

Referee #3: T cells, exhaustion

Balood argue that many tumors establish a significant degree of de novo connections with the nervous system and that tumor denervation can affect tumor progression. Based on this, they began to systematically investigate how the infiltration of tumors with nociceptors and how their associated signaling pathways impact tumor progression.

Overall, they present an impressive study. They found that ablation of nociception significantly slows tumor progression, but most surprisingly, nociception and its associated neurotransmitters were identified to have a very profound effect on the phenomenon of T cell exhaustion. Here, compelling evidence is presented that co-culturing activated T cells with capsaicin-stimulated neurons significantly alters the T cell phenotype, but this effect only occurs when nociceptor neurons are present in the co-cultures. While, as an immunologist, I am less familiar with the neurological aspects of this project, I can clearly say that the authors' immunological observations are quite substantial. It is important to keep in mind that the induction of an exhausted phenotype in ex vivo cultures is still a largely unachieved goal. What has always been puzzling is that T cell exhaustion is rapidly induced in vivo but not in vitro cultures. The findings made by the authors could mean that nociception and interactions with neuropeptides could be the missing link for inducing T cell exhaustion. Despite these very positive assessments, there are some significant structural, as well as scientific, issues associated with the manuscript.

We would like to thank the reviewer for the positive review.

Major points:

3.1. Presently, the manuscript is clearly overloaded with data. I appreciate the authors' decision to present the findings in as much detail as possible, but 49 supplemental figures is a very substantial number and I assume over the limits of nature. At the same time, the massive amount of data precludes the main findings from being adequately explained and reflected by the authors. In fact, I think up to 50% of the data could be removed from the manuscript or labeled as not shown without significantly affecting the core message. So the challenge is to streamline the manuscript. In this context, it is sometimes difficult to reconcile the authors' decisions about which data to show in the main figures and which to show in the supplementary section. Fig. 1 does not provide any significant new insights and is rather technical in nature. At the same time, the impressive phenotype of T cells in tumors with and without nociception is hidden in Supp.Fig. 14. At the same time, Figs. 3 and 4 could be merged and filtered.

We streamlined the article, placing a specific focus on the CGRP-RAMP1 axis.

3.2. It would be very informative if the authors could establish further correlations between the level of nociception that is found in tumors and the level of T cell exhaustion.

To assess this, we implanted B16F10-mCherry-OVA cells (i.d., 5×10^5 cells) into the hindpaw of 8-week-old male and female mice (n=75). We found that enhanced tumor volume led to increased thermal pain hypersensitivity as well as increased intra-tumoral CD8 T-cell exhaustion (**rebuttal figure 12A-B**). The heightened thermal hypersensitivity positively correlated ($R^2=0.52$, $p<0.0001$) with exacerbated CD8 T-cells exhaustion (**Fig. 3A**; **rebuttal figure 12C**; as measured 15 days post implantation).

Rebuttal Figure 12

3.3. Kinetic characterization of the effect of nociception on T cell differentiation in vivo in tumors (i.e. when it requires) would be very informative.

Rebuttal Figure 13

We found that nociceptor neuron ablation increased the proportion of intra-tumoral effector CD8 T-cells (defined as PD1⁺CXCR3⁺ (**Rebuttal Figure 13A**), PD1⁺Tbet⁺ (**Rebuttal Figure 13B**), PD1⁺Ki67⁺ (**Rebuttal Figure 13C**)) as well as that of stem-like memory CD8 T-cells (defined as PD1⁺TCF1⁺CXCR5⁺Tim3⁻, **Rebuttal Figure 13D**).

Due to space constraints, these data are currently not included in the article.

3.4. The degree of T cell exhaustion is somewhat under-explored. I mean, the phenotype is very interesting, but additional signatures such as transcriptional profiles, i.e., Tox, Eomes, Tbet, ID2/ID3, Nr4a profiles would be helpful. Broader gene expression characterization of nociceptor exposed, and unexposed T cells would be informative.

We have addressed this in the experiments in RAG1^{-/-} mice co-transplanted with RAMP1^{wt} and RAMP1^{-/-} CD8 T-cells (Fig 4J-K). In co-transplanted RAG1^{-/-} mice, we found that within the same tumor, the relative proportion of intra-tumor PD1⁺Lag3⁺Tim3⁺ CD8⁺ T-cells was lower in RAMP1^{-/-} CD8⁺ T-cells than in RAMP1^{wt} CD8⁺ T-cells (Fig 4J). Next, we RNA-sequenced FACS-purified RAMP1^{wt} and RAMP1^{-/-} CD8⁺ T-cells from these mice and confirmed that intra-tumoral RAMP1^{-/-} CD8⁺ T-cells express fewer exhaustion markers (PD1, Lag3, Tim3) compared to their RAMP1^{wt} counterparts (Fig 4K). Specifically, we found that levels of the pro-exhaustion transcription factors TOX and EOMES are elevated in intra-tumoral RAMP1^{wt} CD8 T-cells (Fig 4K). In contrast, the level of anti-exhaustion marker T-bet was reduced in the RAMP1^{wt} CD8⁺ T-cells (Fig 4K). Interestingly, the sequencing data confirms our immunophenotyping data by showing an overexpression of PD-1, Lag3 and Tim3 in intra-tumoral RAMP1^{wt} CD8⁺ T cells. These changes were not observed amongst LN resident CD8⁺ T-cells (Fig 4K). Therefore, RAMP1^{-/-} CD8⁺ T-cells are protected from undergoing nociceptor-induced exhaustion, thereby safeguarding their anti-tumor responses.

3.5. It would be interesting to see how the absence of nociception synergizes with checkpoint inhibition.

The absence of nociceptor neurons increased α PDL1-mediated tumor regression. This effect was observed when α PDL1 were given to mice whose tumors were inoculated on the same day as well as in mice whose tumors were the same size at the start of the treatment (achieved by inoculating nociceptor neuron-ablated mice ~2-3 days before littermate control animals; Fig 3G). Tumor-innervating nociceptor neuron silencing completely phenocopied our genetic data, increasing α PDL1-mediated tumor regression in mice co-exposed to BoNT/A (Supp. Fig 5P) or QX-314 (Supp. Fig 6S-T).

Minor parts:

3.6. The authors used antiCD3 depletion, which is uncommon in mice, it would be helpful to repeat these experiments with antiCD8 depletion.

We phenocopied the α CD3 (Supp. Fig. 4F) systemic depletion experiments using α CD8 (Fig. 3I). We found that the defect in tumor growth observed in sensory neuron ablated mice was absent when circulating CD8 T-cells were depleted.

3.7. The use of truly naive (CD44^{low}) OT1 or similar TCR transgenic cells in the tumor experiments would be helpful.

We tested this possibility and found that transplanted naïve OT-1 CD8 T-cells enhanced tumor shrinkage (Fig. 3K) and showed limited exhaustion (Fig. 3L) when transplanted in RAG1^{-/-} mice whose nociceptor neurons were chemically depleted with RTX as compared to when they were administered to naïve RAG1^{-/-} mice. To achieve this, we chemically depleted TRPV1⁺ nociceptor neurons (Resiniferatoxin; 30, 70, 100 μ g/kg; s.c.) from RAG1^{-/-} mice (devoid of B and T-cells), inoculated B16F10-mCherry-OVA (i.d., 5×10^5 cells) six weeks post RTX and adoptively transferred these mice with naïve OVA-specific CD8 T-cells (OT-1 mice; i.v. 1×10^6 cells) when the tumor reached $\sim 500 \text{mm}^3$ (Fig. 3K-L). These data indicate that the slower tumor growth found in TRPV1^{cre::DTA}^{fl/wt} or in RTX-exposed mice depends on nociceptor-induced CD8 exhaustion.

3.8. Abstract is loaded with specific and very condensed information. More conceptual focus and less detailed information would be helpful.

We have amended the abstract.

3.9. Since this manuscript bridges the two fields of neurology and immunology, it would be very helpful to clarify some of the abbreviations. In several places, nomenclature and mice are used that are not defined, i.e., terms such as Nav1.8cre::Td-tomatofl/wt mice and iDISCO are used without explanation.

We have amended the text to better introduce these concepts.

3.10. Tc1 cultured T cells is not standard nomenclature in mice and should be replaced by ex vivo CD3/CD28 activated T cells.

We have amended the text and now define the stimulated cells as ex vivo CD3/CD28 activated T-cells.

3.11. Sometimes authors use an unconventional FACS axis labelling. The axis often combines staining and gating information, which should be strictly separated.

We have relabeled our flow cytometry gating.

3.12. Figure 1A: The authors use the plots to show the degree of infiltration of nociceptor-positive neurons. Since only one condition is shown, it remains a judgment call by the authors that this represents a significant level of nociception.

The investigator who originally did the iDISCO experiment (**Fig. 1A**) has now left Cygnal Therapeutics. Nevertheless, we used whole-mount skin to quantify Nav1.8⁺ neuron density in naïve Nav1.8^{cre}::Td-tomato^{l/wt} mice. As we found to be the case in B16F10-inoculated TRPV1^{cre}::Td-tomato^{l/wt} mice (**Rebuttal Figure 14A-B**), tumor-inoculated Nav1.8^{cre}::Td-tomato^{l/wt} mouse skin shows marginally higher nerve terminal density (~11% of Td-tomato⁺ nerve/mm²; **Rebuttal Figure 14C**) than that of naïve mice (~9% of Td-tomato⁺ nerve/mm²; **Rebuttal Figure 14D**).

Of note, we directly assess the impact of cancer cells on nociceptor neuron neurite outgrowth in culture – which allowed us to precisely control the number of cells plated, the timing and the quantification of the neurites (**Fig. 1C-D**). Rather than the possibility of tumor-inoculated skin to show higher nerve density, we modified the description of these data to emphasize that sensory neurons densely infiltrate the tumor core. Such findings are the biological equivalent to tumor neo-angiogenesis.

Rebuttal Figure 14

3.13. Supp. Fig. 1G and H: is insufficiently explained and it is unclear what is being shown.

We have amended the text to further introduce these concepts.

3.14. Supp. Fig. 5B: the meaning of the display item is unclear.

We have amended the text to clarify this figure.

3.15. Supp.Fig.6 and following FACS figures: they are too small and not readable after printing.

To abide by Nature's supplementary figure limit, we do not show these panels in the revised manuscript.

3.16. Supp.Fig.6A: why are different gates used in Life Dead staining in the two plots shown.

We apologise for this error while putting together the figure. The same Life/Dead plots are used across all the samples.

To abide by Nature's supplementary figure limit, we do not show these panels in the revised manuscript.

3.17. Supp.Fig.7 is difficult to interpret. I assume the middle Colum (KCI-Stim, Neuron CM) is the experimental group and the right one is the control group, but the labeling is confusing and inconsistent between the figure, legend and text.

To abide by Nature's supplementary figure limit, we do not show these panels in the revised manuscript.

3.18. Supp. Figs. 7-9: Of concern, the phenotypes shown here are less clear than in Supp.Fig. 6. For example, PD-1 expression looks similar between the middle experimental group and the left vehicle group in Supp.Fig. 7, and clear differences between TNF α and IFN γ are difficult to discern.

IFN γ , TNF α , IL2 and Tim3 typically show the strongest differential expression between neuron-intact and ablated/silenced groups. This is followed by Lag3 and PD-1. Although the difference in PD-1 expression is more limited, the simultaneous loss of cytotoxic molecules (i.e., IFN γ , TNF α) and proliferative capacity (i.e., IL2), coupled with increased co-expression of exhaustion markers (PD1, Lag3, Tim3) and a reduced capacity to kill melanoma cells indicates that these cells are exhausted.

To abide by Nature's supplementary figure limit, we do not show these panels in the revised manuscript

3.19. Others (Guo 2018, PMID: 29580868) have pointed out that melanomas have lower nociceptor than normal tissues.

Guo and colleagues (2018, PMID: 29580868) quantified TRPV1 expression in a malignant melanoma tissue microarray (18 melanocytic nevus tissues, 62 primary melanoma tissues and 20 metastatic melanoma tissues; <https://www.biomax.us/tissue-arrays/Melanoma/ME1004e>).

To describe their quantification method, the authors refer to Guao et al., (2017) – a paper in which has no quantification of protein expression. This limit greatly our interpretation of their findings. Nevertheless, they do mention that: “immunoreactivity score was the product of (1) the number of cells with positive staining ($\leq 5\%$: 0; 6–25%: 1; 26–50%: 2; 51–75%: 3 and $>75\%$: 4) and (2) the staining intensity (colorless: 0; pallide-flavens: 1; yellow: 2; brown: 3) “.

As such, melanocytic nevus samples (Figure 1E; PMID: 29580868) show a mean score of 9, and a maximum of 12.

- This means that on average: 51-75% (score of 3) of skin cells are TRPV1^{hi} (brown; score of 3).
- Some samples (score 12) would show that $>75\%$ of skin cells (score of 4) are TRPV1^{hi} (score of 3; brown).

This is biologically impossible. Thus, strictly in neurons, less than ~40% of sensory neurons express TRPV1.

Altogether, we are confident that our quantification (comparing adjacent intact tissue and malignant skin) is appropriate.

Reviewer Reports on the First Revision:

Referees' comments:

Referee #1 (Remarks to the Author):

The authors are very responsive and have conducted numerous new experiments and analyses. 1) They showed double staining of TRPV1 and TUBB3 in patient's skin / melanoma samples; 2) They demonstrated several lines of evidence that T cells have significant expression of Ramp1; 3) In addition to CGRP, the authors suggested additional mediators (e.g., galanin) could be released from nociceptors; 4) They conducted additional control experiment for the ablation of non-peptidergic neurons, in further support of a critical role of peptidergic neurons. 5) They used optogenetic to activate NaV1.8+ tumor-innervating nociceptor neurons and found that light activation drives tumor growth; 6) they also justified the sample sizes and added additional statistical analyses. Furthermore, they have now demonstrated the efficacy of post-treatment with QX-314 and BIBN4096, increasing the translational potential of this study. The revised manuscript has been greatly improved. This study is a significant contribution to the field of immunotherapy.

Minor comments:

Fig. 2H: Part of this figure is missing due to pdf conversion.

Fig. 2: D-J: "SLPI-responsive neurons are mostly small- to medium-sized neurons (G-H; mean area = $151\mu\text{m}^2$)". I am not sure. Neurons at this size ($151\mu\text{m}^2$) should be small-sized ones.

Even all the RNAseq data bases show no expression of TRPV1 in immune cells, some of these cell types may still respond to capsaicin, an extremely potent chemical. No expression only means below the detection threshold. People may have difficulty to understand that mRNA expression, protein expression, and function may be not be well correlated.

3.19. Others (Guo 2018, PMID: 29580868) have pointed out that melanomas have lower nociceptor than normal tissues. The authors have tried to address this controversy. It is also possible this may be disease stage dependent. Melanoma patients do not suffer from severe pain, at least in the early stage of disease. Otherwise, patients will seek life-saving early treatments.

Referee #3 (Remarks to the Author):

The authors have significantly revised the manuscript, and I am pleased that the number of display items has been reduced. However, I still see significant deficiencies. These relate in part to the scientific conclusion and to a greater extent to the organization and writing of the manuscript.

In general, I greatly appreciate the results obtained by the authors. They provide strong evidence that "nociceptor neurons" influence tumor growth, and they link this to differences in the degree of

T-cell exhaustion.

However, it remains unclear whether the difference in T cell exhaustion is really due to a direct effect of the "nociceptor neurons" on T cells or indirectly due to differences in tumor growth. The authors show that tumors in which the "nociceptor neurons" have been decimated are smaller, and such smaller tumors might simply cause less T-cell exhaustion. Of course, one could argue that the authors also showed that removal of T cells abolished the reduced tumor growth observed after depletion of "nociceptor neurons," but that does not answer the question. It could still be that T-cell responses targeting the tumor are more effective at an early stage in "nociceptor neuron"-depleted tumors, and this effect could occur before the onset of T-cell exhaustion, which takes some time in tumors. Thus, there is still the possibility that the difference in the level of T cell exhaustion detected at late stage tumors, is simply the result of an early failure of tumor control. This would lead to a higher tumor burden and could in trans lead to a higher level of T cell exhaustion. I therefore asked the authors to provide a better link between the two observations (tumor control and T cell exhaustion) and to better illustrate the kinetics of this phenomenon. In Figure 3A, the authors show a correlation between pain and CD8 depletion. I assume that "pain" somehow refers to the degree of thermal hypersensitivity. However, the authors would have to show that greater "pain" does not simply mean greater tumor size, which could then correlate with greater T-cell depletion. So the authors didn't really answer my question, nor did they respond to the request to illustrate the kinetics of the occurrence of differences in T-cell exhaustion.

Also, I don't think the mechanisms leading to the altered T cell phenotype are really clear. The authors favor that this is mediated via the "CGRP-RAMP1 axis," but both the effects of this pathway on tumor growth and on T-cell phenotype (SupFig. 8) seem to be less potent than those of "nociceptor neuron" ablation. Thus, considerable uncertainty remains as to whether the mechanisms that enhances T cell exhaustion are fully elucidated.

Although the authors have reduced the number of supplementary figures, I still feel that the authors present an enormous amount of data that are often vaguely explained, at least in the immunological part that I feel confident to assess.

For example, a key element of their experimental strategy is the use of DTA mice. Surprisingly, there is no mention in the main text or in the figure legends that this is a *LoxStop*-silenced diphtheria toxin receptor transgenic mouse, nor do the authors mention that the mice were administered diphtheria toxin.

In supplemental Figure 4A, it remains completely unclear how the analysis was performed: Were the cells stimulated before their cytokine production was measured?

In Figure 4H, the legend states that naive cells were "grown and propagated"? What is meant by this?

Throughout the manuscript, figures and legends are too small. To read them requires a 300% magnification. For example, in Supp. Fig. 4 and 7, why do the authors show an enormous amount of primary FACS data instead of following standards in the field and showing representative FACS plots

and using summary bar or scatter plots to compare responses in different mice?

I mentioned earlier that it remains completely unclear why the authors label the x-axis in FACS dot plots with multiple markers, e.g., in Fig. 7F (CD8, PD1, Lag3, Tim3)? Moreover, gating also contradicts the standards in this field. What is the reason for the particular FSC-SSC gates in Fig. 7I, why do the 7-AAD gates just pass through the population? Similar problems arise with Fig. 8.

See Fig. 3. The authors write in the text that "neither B16F10 cells nor mouse tumor-infiltrating immunocytes express neuronal markers." I thought this reference was to a figure in which tumor-infiltrating cells were analyzed. Instead, expression profiles are shown for cells taken from the IMMGEN database. Also, what are "immunocytes"?

Most of these things can be fixed, but I am surprised at the large number of these deficiencies, which are not all mentioned here, at this advanced stage of the manuscript.

Author Rebuttals to First Revision:

REFEREE #1 (Pain). The authors are very responsive and have conducted numerous new experiments and analyses. 1) They showed double staining of TRPV1 and TUBB3 in patient's skin/melanoma samples; 2) They demonstrated several lines of evidence that T cells have significant expression of Ramp1; 3) In addition to CGRP, the authors suggested that additional mediators (e.g., galanin) could be released from nociceptors; 4) They conducted an additional control experiment for the ablation of non-peptidergic neurons, in further support of a critical role of peptidergic neurons; 5) They used optogenetics to activate NaV1.8+ tumor-innervating nociceptor neurons and found that light activation drives tumor growth; 6) they also justified the sample sizes and added additional statistical analyses. Furthermore, they have now demonstrated the efficacy of post-treatment with QX-314 and BIBN4096, thereby increasing the translational potential of this study. The revised manuscript has been greatly improved. This study is a significant contribution to the field of immunotherapy.

We would like to thank the reviewer for their positive review.

MINOR

1. Fig. 2H: Part of this figure is missing due to pdf conversion.

We have amended the figure.

2. Fig. 2: D-J: "SLPI-responsive neurons are mostly small- to medium-sized neurons (G-H; mean area = $151\mu\text{m}^2$)". I am not sure. Neurons at this size ($151\mu\text{m}^2$) should be small-sized ones.

We amended the text accordingly.

3. Although all of the RNAseq databases show no expression of TRPV1 in immune cells, some of these cell types may still respond to capsaicin, an extremely potent chemical. No expression only means below the detection threshold. People may have difficulty understanding that mRNA expression, protein expression, and function may not be well correlated.

We agree with the reviewer and therefore tested this possibility functionally. We found that capsaicin does not impact cultured T cells' functionality (Supp. Fig. 7A-B) nor B16F10 cells' survival (Rebuttal Fig. 1).

4. Guo 2018 (PMID: 29580868) noted that melanomas have lower nociceptor than normal tissues. The authors have tried to address this controversy. It is also possible this may be disease stage-dependent. Melanoma patients do not suffer from severe pain, at least in the early stage of the disease. Otherwise, patients will seek life-saving early treatments.

We agree that pain intensity is likely dependent on the patient's disease state. To partially address this point, we have now carefully mapped out the onset of thermal pain hypersensitivity, the increased frequency of intra-tumoral PD1⁺Lag3⁺Tim3⁺ CD8⁺ T cells, and tumor growth kinetics. To do so, we first conducted two pilot studies which showed that thermal hyperalgesia begins ~13 days post-tumor inoculation (Rebuttal Figure 2A-B). Based on this, we implanted B16F10 cells in various groups of littermate control (TRPV1^{wt}::DTA^{fl/wt}; n=96) and nociceptor-ablated (TRPV1^{cre}::DTA^{fl/wt}; n=18) mice. We then evaluated the level of thermal hypersensitivity (daily), tumor size, and intra-tumoral CD8⁺ T cell exhaustion at the time of sacrifice (days 1, 4, 7, 8, 12, 13, 14, 19, 22; Rebuttal Figure 4A-D). We processed these data by determining the percentage change of each data point to the maximal value obtained in the pain, exhaustion, size dataset, and present these data as percentages of the maximum (100%; Rebuttal Figure 4E).

When compared to their baseline threshold, and in comparison to sensory neuron-ablated mice, 8-week-old female littermate control mice inoculated with B16F10 cells (i.d., 5×10^5 cells) showed significant thermal hypersensitivity on day 7, an effect that peaked on day 21 (Supp. Fig. 4B). In these mice, the intra-tumoral frequency of PD1⁺Lag3⁺Tim3⁺ CD8⁺ T cells (Supp. Fig. 4C), as well as IFN γ ⁺ CD8⁺ T cells (Supp. Fig. 4D), increased 12 days post-tumor inoculation and peaked on day 19. Finally, B16F10 tumor volume peaked on day 22 (Supp. Fig. 4E). Altogether, these data indicate that thermal hypersensitivity precedes significant intra-tumoral CD8 T cell exhaustion by ~5 days and that pain hypersensitivity develops prior to the tumor being measurable using a digital calliper (Supp. Fig. 4F; Rebuttal Figure 4E). Consequently, we found that the genetic elimination of pain-transmitting neurons shields melanoma-bearing mice from significant TILs exhaustion and thus delays tumor growth (Fig. 3B-F; Supp. Fig. 4A-F).

Additionally, melanoma patients often experience chronic itching — a sensory feature that involves an overlapping somatosensory circuitry with the one responsible for transmitting pain information. Along with pain (thermal hyperalgesia; see Rebuttal Figs. 2 and 4), we also found that ~1/3 of melanoma-bearing mice experienced spontaneous itch (not shown).

As it relates to Guo *et al.* 2018 ² (PMID: 29580868), they quantified TRPV1 expression in a malignant melanoma tissue microarray (18 melanocytic nevus tissues, 62 primary melanoma tissues and 20 metastatic melanoma tissues; <https://www.biomax.us/tissue-arrays/Melanoma/ME1004e>). For quantification methods, the authors refer to Guo *et al.* 2017 ³ (PMID: 28740547) - a paper in which the authors did not actually quantify protein expression. They mentioned that "*immunoreactivity score was the product of (1) the number of cells with positive staining (≤5%: 0; 6–25%: 1; 26–50%: 2; 51–75%: 3 and >75%: 4) and (2) the staining intensity (colourless: 0; pallide-flavens: 1; yellow: 2; brown: 3)*".

Based on this, melanocytic nevus samples (Fig. 1E) showed a mean score of 9 and a maximum of 12.

- This means that, on average, 51–75% (score: 3) of skin cells are according to them TRPV1^{hi} (brown; score: 3).
- Some samples (score: 12) would show that >75% of skin cells (score: 4) are TRPV1^{hi} (brown; score: 3).

This is simply biologically impossible. Strictly in neurons, less than ~40% of sensory neurons express TRPV1. Altogether, we are confident that our quantification (comparing adjacent intact tissue and malignant skin) is appropriate and the Guo *et al* data is incorrect.

REFEREE #3 (*T cells, exhaustion*). The authors have significantly revised the manuscript, and I am pleased that the number of display items has been reduced. However, I still see significant deficiencies. These relate in part to the scientific conclusion and—to a greater extent—the organization and writing of the manuscript.

1. In general, I greatly appreciate the results obtained by the authors. They provide strong evidence that "nociceptor neurons" influence tumor growth and link this to differences in the degree of T cell exhaustion. However, it remains unclear whether the difference in T cell exhaustion is really due to a direct effect of the "nociceptor neurons" on T cells or indirectly due to differences in tumor growth. The authors show that tumors in which the "nociceptor neurons" have been decimated are smaller, and such smaller tumors might simply cause less T cell exhaustion. Of course, one could argue that the authors also showed that removal of T cells abolished the reduced tumor growth observed after depletion of "nociceptor neurons," but that does not answer the question.

It could still be that T cell responses targeting the tumor are more effective at an early stage in "nociceptor neuron"-depleted tumors, and this effect could occur before the onset of T cell exhaustion, which takes some time in tumors. Thus, there is still the possibility that the difference in the level of T cell exhaustion detected in late-stage tumors is simply the result of an early failure of tumor control. This would lead to a higher tumor burden and could in trans lead to a higher level of T cell exhaustion. Therefore, I asked the authors to provide a better link between the two observations (tumor control and T cell exhaustion) and to better illustrate the kinetics of this phenomenon.

In Fig. 3A, the authors show a correlation between pain and CD8 depletion. I assume that "pain" somehow refers to the degree of thermal hypersensitivity. However, the authors would have to show that greater "pain" does not simply mean greater tumor size, which could then correlate with greater T cell depletion. So, the authors didn't really answer my question, nor did they respond to the request to illustrate the kinetics of the occurrence of differences in T cell exhaustion.

We have now carefully mapped out the onset of thermal pain hypersensitivity, increased frequency in intra-tumoral PD1⁺Lag3⁺Tim3⁺CD8⁺ T cells, and tumor growth (i.d., 2×10⁵ cells; left hindpaw) kinetics. To do so, we first conducted two pilot studies which showed that littermate control mice develop pain hypersensitivity ~13 days post-tumor inoculation (volume ≥ 525mm³; **Rebuttal Figure 2A-B**). This effect was absent in nociceptor-ablated mice with large tumors (volume ≥ 525mm³). That being said, no hypersensitivity was observed in either group of mice with small tumors (~50mm³). In comparison with littermate control mice with large tumors (volume ≥ 525mm³; ~13 days post-tumor inoculation), we also found that nociceptor neuron ablated mice were also shielded from increased intra-tumoral frequency of PD1⁺Lag3⁺Tim3⁺ CD8⁺ T cells (**Rebuttal Figure 2A-D**). Albeit in a different context, this finding is in accordance with that of Kuchroo *et al.*¹ (PMID: 32937153). In their work,

Rebuttal Figure 2. Pilot study. At an early stage, defined as a B16F10 tumor volume of ~50mm³, nociceptor neurons ablation does not impact tumor size (A), thermal pain sensitivity (B), intra-tumoral frequency of PD1⁺Lag3⁺Tim3⁺ (C) or IFN γ (D) CD8⁺ T-cells. In mice with an established B16F10 tumor, defined as having a volume of ~525mm³ (13 days post-tumor inoculation), nociceptor neuron ablation is protected from thermal pain hypersensitivity (B) and shows a lower tumor volume (A) as well as lower intra-tumoral frequency of PD1⁺Lag3⁺Tim3⁺ CD8⁺ T-cells (C). Frequency of IFN γ ⁺ CD8⁺ T-cells are increased in nociceptor ablated mice (D).

they also found that differential exhaustion (glucocorticoid-mediated in their case) was absent in mice with small tumors (**Rebuttal Fig. 3**; reproduced from Fig. 3G. *Immunity*; PMID: 32937153)¹.

Based on these pilot studies, we implanted B16F10 cells in several groups of littermate control (TRPV1^{wt}::DTA^{fl/wt}; n=96) and nociceptor-ablated (TRPV1^{cre}::DTA^{fl/wt}; n=18) mice. We then evaluated the level of thermal hypersensitivity (daily), tumor size, and intra-tumoral CD8⁺ T cell exhaustion at the time of euthanasia (days 1, 4, 7, 8, 12, 13, 14, 19, 22; **Rebuttal Figure 4A-D**). We processed these data by determining the percentage change of each data point to the maximal value obtained in the pain, exhaustion, size datasets, and then presented these data as percentages of the maximum (100%; **Rebuttal Fig. 4E**; **Supp. Fig. 4F**).

When compared to their baseline threshold and sensory neuron-ablated mice, 8-week-old female littermate control mice inoculated with B16F10 cells (i.d., 2×10⁵ cells; left hindpaw) showed significant thermal hypersensitivity on day 7, with an effect that peaks on day 21 (**Supp. Fig. 4B**). In these mice, the intra-tumoral frequency of PD1⁺Lag3⁺Tim3⁺ CD8⁺ T cells (**Supp. Fig. 4C**), as well as IFN γ ⁺ CD8⁺ T cells (**Supp. Fig. 4D**), increased 12 days post-tumor inoculation and peaked on day 19. Finally, B16F10 tumor volume peaked on day 22 (**Supp. Fig. 4E**). Altogether, these data indicate that thermal hypersensitivity precedes that of significant intra-tumoral CD8 T cell exhaustion by ~5 days. The data also suggest that pain hypersensitivity developed prior to the tumor being measurable using a digital calliper (**Supp. Fig. 4F**; **Rebuttal Fig. 4A-E**). Consequently, we found that the genetic elimination of pain-transmitting neurons shields melanoma-bearing mice from significant TILs exhaustion and thereby delays tumor growth (**Fig. 3E**; **Supp. Fig. 4A-F**).

It is also worth noting that unlike in immunogenic tumors (**Fig. 3A-E**), the ablation of nociceptor neuron shows a limited impact on tumor growth and intra-tumoral CD8⁺ T cell exhaustion in mice inoculated with non-immunogenic tumors (**Supp. Fig. 4J**). Building on these data, we also showed that naïve OT-1 CD8⁺ T cells given to mice with large established tumors (volume \geq 500mm³) are more potent in eliminating malignant cells in the absence of nociceptor neurons (RTX; **Fig. 3K-L**). These data indicated that i) the mechanism to prime cytotoxic CD8⁺ T cells is not impacted by the absence of neurons and that ii) despite having established tumors, the ability of T cells to eliminate these malignant cells is increased in neuron-ablated mice. The optogenetic activation of tumor-innervating nociceptor neurons in mice with large (volume \geq 200mm³) or visible tumors (volume \geq 20 mm³) promotes CD8⁺ T cell exhaustion and leads to higher tumor growth (**Supp. Fig. 4G**). The increase in tumor growth was also linked to enhancing intra-tumoral CGRP levels, indicating the engagement of pain-transmitting neurons (**Supp. Fig. 4M**).

Rebuttal Figure 3. Copy of Fig 3G from V. Kuchroo and colleagues (*Immunity*. 2020, PMID: 32937153)¹ showing that corticoids do not impact the level of exhausted intra-tumoral CD8⁺ T-cells found in small size tumors (defined as early). In contrast, intermediate size tumors show lower exhaustion of intra-tumoral CD8⁺ T-cells when exposed to corticoids (red group).

Rebuttal Figure 4. When compared to their baseline threshold, littermate control mice (TRPV1^{wt}::DTA^{fl/wt}) inoculated with B16F10 cells showed significant thermal hypersensitivity on day 7, an effect that peaks on day 21 (**A**). In these mice, intra-tumoral frequency of PD1⁺Lag3⁺Tim3⁺ CD8⁺ T-cells is increased 12 days post tumor inoculation, an effect that peaked on day 22 (**B**). Finally, B16F10 tumor volume peaked on day 22 (**D**). When compared with littermate control mice, sensory neuron ablated (TRPV1^{cre}::DTA^{fl/wt}) mice inoculated with B16F10 cells showed no thermal pain hypersensitivity (**A**), reduced intra-tumoral frequency of PD1⁺Lag3⁺Tim3⁺ CD8⁺ T-cells (**B**) and tumor volume (**D**). In littermate control mice, we found that the onset of significant thermal pain hypersensitivity (day 7), precedes the increase in intra-tumoral frequency of PD1⁺Lag3⁺Tim3⁺ CD8⁺ T-cells (day 12) as well as significant tumor growth (day 12; **E**).

[Text redacted (below)]

Another possibility to test this hypothesis would have been to generate a CRISPR knockout of SLPI (i.e., the nociceptor activating agent we identified to be secreted from the cancer cells) from the B16F10 cells and measure whether the SLPI^{ko} B16F10 cells would trigger less pain and intra-tumoral CD8 exhaustion. While this is an attractive option, we found that many other nociceptor-activating agents—including ██████████ (not shown)—are produced by B16F10 cells. We discovered that these agents also readily trigger pain and CGRP release from nociceptor neurons. It would therefore be virtually impossible to genetically eliminate all these different noxious agents together from the B16F10 cells.

2. I don't think the mechanisms leading to the altered T cell phenotype are really clear. The authors favour that this is mediated via the "CGRP-RAMP1 axis," but both the effects of this pathway on tumor growth and T cell phenotype (Supp. Fig. 8) seem to be less potent than those of "nociceptor neuron" ablation. Thus, considerable uncertainty remains as to whether the mechanisms that enhance T cell exhaustion are fully elucidated.

Our findings (summarized in **Rebuttal Table 1, shown on page 7 of this letter**) strongly support the CGRP-RAMP1 axis as the main driver of intra-tumoral PD1⁺Lag3⁺Tim3⁺ CD8⁺ T cells' increased frequency. Appended below are ten lines of evidences supporting this conclusion:

- i) RAMP1 antagonism with **BIBN (1.3-fold; Fig. 4G)** shows marginally lower efficacy in reducing the levels of intra-tumoral PD1⁺Lag3⁺Tim3⁺ CD8⁺ T cells when compared to **neuronal ablation (1.4-fold; Fig. 3F)**.
 - a. This marginal difference (1.3- vs. 1.4-fold) is likely due to pharmacological issues (BIBN potency/exposure/target engagement) rather than a mechanism that was not elucidated. Thus, unlike the complete genetic ablation of nociceptors (TRPV1^{cre::DTA^{fl/wt}}), the RAMP1 antagonist BIBN was given to the mice systemically (i.p.) once every 2 days as per *Chiu and colleagues (Cell. 2018)* ⁴.
 - b. It is also imperative to note that the efficacy of neuron ablation was measured 16 days post-tumor inoculation, while that of BIBN was measured after 13 days. The larger tumor size in littermate control mice (in the neuron-ablated experiment; **Fig. 3F**) is likely to show a stronger effect size than in mice with smaller tumors (as found in the BIBN experiment; **Fig. 4G**).
- ii) RAMP1 antagonism with BIBN did not affect intra-tumoral PD1⁺Lag3⁺Tim3⁺ CD8⁺ T cells when given to nociceptor-ablated mice (**Supp. Fig. 8L**).
 - a. Therefore, the action of BIBN in reducing PD1⁺Lag3⁺Tim3⁺ CD8⁺ T cells is dependent on active intra-tumoral nociceptor neurons and their release of CGRP within the tumor microenvironment.
- iii) The reduction in intra-tumoral PD1⁺Lag3⁺Tim3⁺ CD8⁺ T cells observed in nociceptor-ablated mice was completely abolished by the daily injection of recombinant CGRP around the tumor (**0.84-fold; Fig. 4E**).
 - a. This data further supports the notion that nociceptor neuron ablation reduction in intra-tumoral PD1⁺Lag3⁺Tim3⁺ CD8⁺ T cells depends on active intra-tumoral nociceptor neurons and their release of CGRP within the tumor microenvironment.
- iv) In RAG1^{ko} mice transplanted with RAMP1^{ko} (CGRP receptor knockout mice) and RAMP1^{wt} CD8 T cells, intra-tumoral PD1⁺Lag3⁺Tim3⁺ CD8⁺ T-cells were drastically lower in RAMP1^{ko} CD8⁺ T-cells than in RAMP1^{wt} CD8⁺ T-cells (**2.43-fold; Fig. 4J-K**).
 - a. This effect was confirmed using both flow cytometry (**Fig. 4J**) and unbiased RNA sequencing (**Fig. 4K**).
- v) In RAG1^{ko} mice transplanted with naïve OT-1 CD8⁺ T-cells, chemoablation of nociceptor neurons using RTX reduced prevent intra-tumoral CGRP release and reduce intra-tumoral PD1⁺Lag3⁺Tim3⁺ CD8⁺ T-cells (**1.98-fold; Fig. 3L**).
- vi) Nociceptor neuron silencing with QX-314 prevents intra-tumoral CGRP release and reduces intra-tumoral PD1⁺Lag3⁺Tim3⁺ CD8⁺ T-cells (**1.18-fold; Supp. Fig. 6R**).
- vii) Optogenetic activation of tumor-innervating nociceptor neurons (Nav1.8^{Cre::ChR2^{fl/wt}}), when started once B16F10 tumors were visible (~20mm³) or well established (~200mm³), resulted in enhanced tumor growth (**Supp. Fig. 4L**, as measured until day 14) and intra-tumoral CGRP release (**Supp. Fig. 4M**).

viii) NaV1.8^{cre}::DTA^{fl/wt} neuron-ablated mice show reduced intra-tumoral CGRP levels and PD1⁺Lag3⁺Tim3⁺ CD8⁺ T-cells frequency (Supp. Fig. 8A). In both mice, intra-tumoral CGRP levels positively correlate ($p \leq 0.01$) with PD1⁺Lag3⁺Tim3⁺ CD8⁺ T-cells frequency (Fig. 4C).

ix)

x) *In vitro*, we found that cytotoxic CD8⁺ T-cells exposed to nociceptor neuron-conditioned media (triggered by KCl) increased the co-expression of PD1⁺Lag3⁺Tim3⁺ in cytotoxic CD8⁺ T-cells (14.25-fold; Supp. Fig. 7C).

a. This effect was completely abolished when the cytotoxic CD8⁺ T cells were exposed to neuron-conditioned media supplemented with the RAMP1 antagonist CGRP₈₋₃₇ (0.97-fold; Supp. Fig. 7C)

xi) *In vitro*, we found that RAMP1^{wt} cytotoxic CD8⁺ T-cells treated with CGRP increased their co-expression of PD1⁺Lag3⁺Tim3⁺ (1.64-fold; Fig. 4A) and blocked the OT-1 cytotoxic CD8⁺ T-cells elimination of B16F10-OVA cells (Fig. 4B).

a. This effect was completely absent in RAMP1^{KO} cytotoxic CD8⁺ T-cells treated with CGRP (0.75-fold; Fig. 4A) or when the OT-1 cytotoxic CD8⁺ T-cells were exposed to neuron-conditioned media supplemented with the RAMP1 antagonist CGRP₈₋₃₇ (0.97-fold; Supp. Fig. 7H-I)

In addition to these ten independent lines of evidence, we found that along with *Calca*, tumor-innervating sensory neurons upregulate the neuropeptide galanin (Fig. 1H-J) and that CD8⁺ T cells express galanin receptors (ImmGen⁵). As we found to be the case for CGRP, galanin also increased cultured cytotoxic PD1⁺Lag3⁺Tim3⁺ CD8⁺ T cell frequency (1.17-fold; Rebuttal Fig. 5A-C). In patient biopsies, the upregulation of galanin—as determined by bulk RNA sequencing—was also correlated with reduced survival (Supp. Fig. 9J; Rebuttal Fig. 5D). Notably, we found no effects of Substance P on any of the tested parameters (Rebuttal Fig. 5E-G).

Work that we have performed for a separate study (*in preparation*) also showed that nociceptor neurons overexpress [redacted] and drive intra-tumoral CD8⁺ T cell exhaustion via cell-cell contact (not shown). Finally, Ru-Rong Ji's group has shown that nociceptor neurons express PD1 (*Ji. Nature Neuro. 2017; PMID: 28530662*)⁶ and STING1 (*Ji. Nature. 2021 PMID: 33442058*)^{7,8}, which—while not directly assessed yet in our studies—may also drive intra-tumoral CD8⁺ T cell exhaustion.

Overall, the release of galanin by intra-tumoral nociceptor neurons is likely to marginally increase the frequency of intra-tumoral PD1⁺Lag3⁺Tim3⁺ CD8⁺ T cells. Via cell-cell contact, PDL1⁺, PD1⁺⁶ and/or STING1⁺^{7,8} nociceptors could potentially increase the frequency of exhausted CD8-T cells.

While we acknowledge these possibilities, our data (summarized in Rebuttal Table 1) strongly support CGRP as the main driver of intra-tumoral PD1⁺Lag3⁺Tim3⁺ CD8⁺ T cells.

Rebuttal Figure 5. Splenocytes-isolated CD8⁺ T-cells from naive C57BL6 mice were cultured under Tc1-stimulating conditions (*ex vivo* CD3/CD28 exposure) for 48h. On alternate days for 4 days, the cytotoxic CD8⁺ T-cells were exposed to Galanin (A-C; 0-1 μ M), Substance P (E-G; 0-1 μ M), or its vehicle (A-C, E-G). As measured after 4 days stimulation, data show that galanin increase cytotoxic CD8⁺ T-cells expression of PD-1+ (A), Lag3+ (B), and PD1+Lag3+ (C) when exposed to galanin. Inversely, substance P has no impact on the tested parameters (E-G). *In-silico* analysis of TCGA data linked the survival rate among 459 melanoma patients with their relative expression levels of various genes of interest (determined by bulk RNA sequencing of tumor biopsy). Kaplan-Meier curves show the patients' survival after segregation in two groups defined by their low or high expression of a gene of interest. Increased galanin expression (labelled as high; red curve; D) in patient's biopsy correlates with worsened survival ($p \leq 0.05$).

[Redacted text (above)]

- 3.** Although the authors have reduced the number of supplementary figures, I still feel that the authors present an enormous amount of data that are often vaguely explained, at least in the immunological part that I feel confident to assess.

We have further streamlined our datasets.

- 4.** A key element of their experimental strategy is the use of DTA mice. Surprisingly, there is no mention in the main text or the figure legends that this is a LoxStop-silenced diphtheria toxin receptor transgenic mouse, nor do the authors mention that the mice were administered diphtheria toxin.

We have not used DTR mice. As described in the Methods section, we only used lox-stop-lox DTA mice. Please refer to the section "Animals" within the Methods section for further details.

- 5.** In Supp. Fig. 4A, it remains completely unclear how the analysis was performed: Were the cells stimulated before their cytokine production was measured?

As described in the Methods section, intracellular cytokine staining was only performed on stimulated (PMA/Ionomycin) CD8⁺ T-cells. Please refer to the section "Intracellular cytokine staining" within the Methods section for further details.

- 6.** In Figure 4H, the legend states that naïve cells were "grown and propagated"? What is meant by this?

We amended the text to read as follows: "Naive splenocyte CD8⁺ T-cells were FACS-purified from RAMP1^{wt} (CD45.1⁺) or RAMP1^{-/-} (CD45.2⁺) mice, expanded and stimulated (CD3/CD28 beads + IL2) *in vitro*".

- 7.** Throughout the manuscript, figures and legends are too small. To read them requires a 300% magnification. For example, in Supp. Figs. 4 and 7, why do the authors show an enormous amount of primary FACS data instead of following standards in the field and showing representative FACS plots and using summary bars or scatter plots to compare responses in different mice?

To show the robustness of our findings, we originally included multiple replicates. We have now changed the figure to only show the gating strategy.

- 8.** I mentioned earlier that it remains completely unclear why the authors label the x-axis in FACS dot plots with multiple markers, e.g., in Fig. 7F (CD8, PD1, Lag3, Tim3)? Moreover, gating also contradicts the standards in this field.

As per the comment, we have reformatted the figure.

- 9.** What is the reason for the particular FSC-SSC gates in Fig. 7I? Why do the 7-AAD gates just pass through the population? Similar problems arise with Fig. 8.

As per the comment, we have now re-gated this data. We added the FMO control for your reference.

10. The authors write in the text that "neither B16F10 cells nor mouse tumor-infiltrating immunocytes express neuronal markers." I thought this reference was to a figure in which tumor-infiltrating cells were analyzed. Instead, expression profiles are shown for cells taken from the IMMGEN database.

In fact, our RNA sequencing data directly support these conclusions. Specifically:

- i) In **Fig. 2A–B** (*Naïve DRG neurons (TRPV1^{cre}::QuASR2-eGFP^{fl/wt}), B16F10-mCherry-OVA, and OVA-specific cytotoxic CD8⁺ T cells were cultured alone or in combination*), we found that in comparison to nociceptor neurons, neither B16F10 nor CD8⁺ T cells express neuronal markers (i.e., Calca, Snap25, Trpv1, and Nav1.8).
- ii) In **Fig. 4K** (*RAG1^{KO} mice transplanted with RAMP1^{wt} and RAMP1^{KO} CD8⁺ T-cells*), we found that tumor-infiltrating CD8⁺ T-cells do not express neuronal markers (i.e., Calca, Snap25, and Trpv1, Nav1.8).
- iii) In **Rebuttal Fig. 6** (*Single-cell RNA sequencing of tumor-infiltrating leukocytes from tumor innervated and denervated mice*), we found that tumor-infiltrating leukocytes do not express neuronal markers (i.e., Calca, Snap25, Trpv1, and Nav1.8).

These data are also completely supported by published datasets. Specifically:

- iv) In **Supp. Fig. 1A**. *In silico analysis* of single-cell RNA sequencing of human melanoma-infiltrating cells⁹ revealed no Trpv1, Nav1.8, Snap25, or Calca expression in malignant melanoma cells (defined as CD90-CD45⁻) from 10 different patients' biopsies (**Supp. Fig. 1A**).
- v) In **Supp. Fig. 1B**. *In silico analysis* of single-cell RNA sequencing⁹ of human cancer-associated fibroblasts, macrophages, and endothelial, natural killer, T, and B cells do not express Calca, Snap25, Trpv1 or Nav1.8 channels (**Supp. Fig. 1B**).
- vi) In **Supp. Fig. 1C**. *In silico analysis* of human immune cells¹⁰ revealed their basal expression of Cd45 and their absence of expression of Calca, Snap25, Trpv1 and Nav1.8 (**Supp. Fig. 1C**).
- vii) In **Supp. Fig. 3A**. *In silico analysis* of three different B16F10 cell cultures¹¹ revealed their basal expression of Braf and Pten. In contrast, B16F10 cells do not express Calca, Snap25, Trpv1, or Nav1.8 channels (**Supp. Fig. 3A**).
- viii) In **Supp. Fig. 3B**. ImmGen⁵ RNA sequencing of leukocyte subpopulations reveals their basal expression of Cd45 and Ramp1. In contrast, immune cells do not express Snap25, Trpv1, or Nav1.8 (**Supp. Fig. 3B**).
- ix) In **Supp. Fig. 3C**. A meta-analysis of seven published nociceptor neuron expression profiling datasets¹² revealed their basal expression of sensory neuron markers (Trpv1, Trpa1) and neuropeptides (Sp, Vip, Nmu, and Calca; **Supp. Fig. 3C**).

Rebuttal Table 1. Summary of data on the role of tumor-innervating nociceptor neuron-released CGRP in driving RAMP1⁺ CD8⁺ T cell exhaustion, as measured by increased frequency of PD1⁺Lag3⁺Tim3⁺. Tumor size, number of cancer cells injected and the day of measurement are reported for your reference

B16F10	day	Figure	IN VIVO	PD1+Lag3+Tim3+ CD8		Tumor Size	
				Fold ?	(%)	Fold	(mm3)
500k	10	4L	Rag1ko + RAMP1 WT CD8 T-cells Rag1ko + RAMP1 KO CD8 T-cells	2.43	7.71 3.17	1.78	797 448
500k	19	3L	Veh + OT1 CD8 RTX + OT1 CD8	1.98	6.83 3.45	1.92	614 319
500k	16	3F	Nociceptor Intact Nociceptor Ablated	1.40	43.95 31.38	2.61	873 334
500k	13	4G	Veh Ramp1 antagonist (BIBN)	1.30	65.56 50.43	4.29	425 99
500k	17	SF 6R	Veh QX-314	1.18	53.49 45.26	4.05	894 221
500k	14	SF 8L	Nociceptor Intact Nociceptor ablated Nociceptor Intact + BIBN Nociceptor Ablated + BIBN	1.03	47.24 34.22 36.21 35.32	0.96	797 472 491 512
500k	11	4E	Nociceptor Intact + CGRP Nociceptor Ablated + CGRP	0.84	35.35 42.26	1.26	482 382

Drug []	Dilution	Figure	IN VITRO	PD1+Lag3+Tim3+ CD8	
				Fold ?	(%)
1:2	SF 7A	CD8 + veh CD8 + neuron Cond Media CD8 + neuron CM + BIBN		14.25	2.20 31.31
				0.97	2.26
1:10	SF 7C	CD8 CD8 and neuron co-culture		3.32	5.49 18.22
100 nM	4A	Ramp1 wt CD8 + veh Ramp1 wt CD8 + CGRP Ramp1 ko CD8 + CGRP		1.64	5.53 9.09
				0.75	4.15
1000 nM	Reb 3A	CD8 + veh CD8 + Galanin		1.17	37.01 43.35
1000 nM	Reb 3D	CD8 + veh CD8 + Substance P		1.02	37.01 37.71

Rebuttal References.

- 1 Acharya, N. *et al.* Endogenous Glucocorticoid Signaling Regulates CD8(+) T Cell Differentiation and Development of Dysfunction in the Tumor Microenvironment. *Immunity* **53**, 658-671 e656, doi:10.1016/j.immuni.2020.08.005 (2020).
- 2 Yang, Y. *et al.* Downregulated TRPV1 Expression Contributes to Melanoma Growth via the Calcineurin-ATF3-p53 Pathway. *J Invest Dermatol* **138**, 2205-2215, doi:10.1016/j.jid.2018.03.1510 (2018).
- 3 Guo, W. *et al.* Down-regulated miR-23a Contributes to the Metastasis of Cutaneous Melanoma by Promoting Autophagy. *Theranostics* **7**, 2231-2249, doi:10.7150/thno.18835 (2017).
- 4 Pinho-Ribeiro, F. A. *et al.* Blocking Neuronal Signaling to Immune Cells Treats Streptococcal Invasive Infection. *Cell* **173**, 1083-1097 e1022, doi:10.1016/j.cell.2018.04.006 (2018).
- 5 Zemmour, D., Goldrath, A., Kronenberg, M., Kang, J. & Benoist, C. The ImmGen consortium OpenSource T cell project. *Nat Immunol* **23**, 643-644, doi:10.1038/s41590-022-01197-z (2022).
- 6 Chen, G. *et al.* PD-L1 inhibits acute and chronic pain by suppressing nociceptive neuron activity via PD-1. *Nat Neurosci* **20**, 917-926, doi:10.1038/nn.4571 (2017).
- 7 Donnelly, C. R. *et al.* STING controls nociception via type I interferon signalling in sensory neurons. *Nature* **591**, 275-280, doi:10.1038/s41586-020-03151-1 (2021).
- 8 Wang, K. *et al.* STING suppresses bone cancer pain via immune and neuronal modulation. *Nat Commun* **12**, 4558, doi:10.1038/s41467-021-24867-2 (2021).
- 9 Jerby-Arnon, L. *et al.* A Cancer Cell Program Promotes T Cell Exclusion and Resistance to Checkpoint Blockade. *Cell* **175**, 984-997 e924, doi:10.1016/j.cell.2018.09.006 (2018).
- 10 Monaco, G. *et al.* RNA-Seq Signatures Normalized by mRNA Abundance Allow Absolute Deconvolution of Human Immune Cell Types. *Cell Rep* **26**, 1627-1640 e1627, doi:10.1016/j.celrep.2019.01.041 (2019).
- 11 Castle, J. C. *et al.* Exploiting the mutanome for tumor vaccination. *Cancer Res* **72**, 1081-1091, doi:10.1158/0008-5472.CAN-11-3722 (2012).
- 12 Crosson, T *et al.* Profiling of how nociceptor neurons detect danger - new and old foes. *J Intern Med* **286**, 268-289, doi:10.1111/joim.12957 (2019).

Reviewer Reports on the Second Revision:

Referees' comments:

Referee #1 (Remarks to the Author):

The authors included impressive amount of new data showing the time course of thermal pain (threshold), tumor growth, and CD8+ cells. While the evidence presented in this study is convincing, the limitations of this study should be discussed.

1) Page 2: “we probed for the presence of nociceptor neurons by assessing TRPV1+ (Transient Receptor Potential cation channel subfamily Vanilloid 1) cells in melanoma patients’ biopsies”. Supplementary Figure 2: “Increased levels of TRPV1+ cells (G) were found in the tumor”. This is confusing. Are these TRPV1+ cells or just TRPV1+ nerve fibers?

2) Supplementary Figure 3: Trpv1, Nav1.8, Snap25, or Ramp1 are not expressed by B16F10 cancer cells or mouse immune cells.

Please be cautious and change “not expressed” to “not detected” using RNaseq method”. Some sensitive method such as RNAscope may detect very low level of mRNA expression. IHC may also detect protein expression. Calcium imaging may also see capsaicin response. It appears the authors ignore the vast literature showing TRPV1 expression in non-neuronal cells including immune cells and tumor cells (PMID: 29580868). TRPV1 expression has been reported in peripheral blood lymphocytes in humans (PMID: 21215279; PMID: 16777226; PMID: 18983665; PMID: 21436684). Capsaicin was shown to induce inhibition of mitogen and interleukin-2-stimulated T cell proliferation (PMID: 8784269). TRPV1 signaling inhibits differentiation and activation of human dendritic cells (PMID: 19397909). Immunological role of neuronal receptor vanilloid receptor 1 expressed on dendritic cells (PNAS, PMID: 15793000). TRPV1 is expressed by glial cells including microglia, astrocytes (reviewed in PMID: 28342781). Capsaicin and TRPV1 regulate gene expression in macrophages (PMID: 14530214). TRPV1 shows an activating effect over the immune system and regulates T cell receptor-induced signaling (Bertin et al., 2014, Nat. Immunol., PMID: 25282159). Overall, the function of the gene is more important than the expression level of the gene.

3) It is unclear how thermal pain threshold was calculated. “Radiant heat was applied to the hindpaw and the time for withdrawal was measured”. I guess your original data is withdrawal latency (second). Please describe your baseline latency. Also, how big is your light spot? If the light spot is big enough, heat stimulation may also involve the adjacent intact tissue, which will show hypersensitivity.

4) “Additionally, melanoma patients often experience chronic itching — a sensory feature that involves an overlapping somatosensory circuitry with the one responsible for transmitting pain information”.

It is interesting to mention itch in your model. However, pain can suppress itch, and loss of pain or analgesia can induce itch.

Here is a typical diagnosis of melanoma in patients: “The skin lesion may feel different and may itch,

ooze, or bleed, but a melanoma skin lesion usually does not cause pain".
<https://www.cancer.net/cancer-types/melanoma/symptoms-and-signs>).

Referee #3 (Remarks to the Author):

The authors have done a great job in revising the MS and I think it is a significant and important report for the field.

However, there is one issue that remains unresolved and that I had raised in the previous rounds of revisions. The authors make a very clear statement by saying in the abstract and in the caption of Figure 4 that "Nociception controls T cell exhaustion." I think this is a clear exaggeration. I agree that there is an effect on the phenotype of T cells, but the magnitude of the changes does not justify the conclusion that nociception "controls" depletion. In my opinion, this implies a much more profound effect than what the authors show. Based on the data presented, the authors should tone down this statement and conclude rather that nociception influences and modulates exhaustion rather than controlling it.

Minor addition point.

Figure 4J: "RAG1^{-/-} → RAMP1^{wt} CD8 + RAMP1^{-/-} CD8". The caption should be changed to 'RAMP1^{wt} CD8 + RAMP1^{-/-} CD8 → RAG1^{-/-}', to reflect that the cells were transferred into RAG mice. This also applies Figure 4H, I.

Author Rebuttals to Second Revision:

REFEREE #1 (*Pain*). The authors included impressive amount of new data showing the time course of thermal pain (threshold), tumor growth, and CD8+ cells. While the evidence presented in this study is convincing, the limitations of this study should be discussed.

We would like to thank the reviewer for their positive review.

1) Page 2: “we probed for the presence of nociceptor neurons by assessing TRPV1+ (Transient Receptor Potential cation channel subfamily Vanilloid 1) cells in melanoma patients’ biopsies”. Supplementary Figure 2: “Increased levels of TRPV1+ cells (G) were found in the tumor”. This is confusing. Are these TRPV1+ cells or just TRPV1+ nerve fibers?

Our data suggest that these are TRPV1+ nerve fibers. We amended the text accordingly.

In the malignant cells (defined as CD90-CD45-) from ten different melanoma patients’ biopsies, the transcripts encoding for TRPV1 or Tubb3 are not detected using sequencing (**Extended data Figure 1A**). As such, the signal we measured by histology can likely be ascribed to tumor neo-innervation ((**Extended data Figure 2**)). In support of this hypothesis, we used immunofluorescence and confirmed that a TRPV1 signal ((**Extended data Figure 2**)) is co-expressed with the neuronal marker Tubb3 (**Rebuttal Figure 1**). In addition, four metastatic melanoma patients’ primary tumor biopsies were stained with S100 (malignant melanoma marker; PMID: 9187834), Tubb3 (neuronal marker; PMID: 1527798) and TRPV1 (nociceptor neuron marker; PMID: 9349813). Although the presence of the TRPV1+ nerve twigs is sparse, we do detect the presence of TRPV1/Tubb3+ fibers surrounding the malignant cells (S100+; **Rebuttal Figure 1**).

2) Supplementary Figure 3: Trpv1, Nav1.8, Snap25, or Ramp1 are not expressed by B16F10 cancer cells or mouse immune cells. Please be cautious and change “not expressed” to “not detected” using RNAsec method”. Some sensitive method such as RNAscope may detect very low level of mRNA expression. IHC may also detect protein expression. Calcium imaging may also see capsaicin response. It appears the authors ignore the vast literature showing TRPV1 expression in non-neuronal cells including immune cells and tumor cells (PMID: 29580868). TRPV1 expression has been reported in peripheral blood lymphocytes in humans (PMID: 21215279; PMID: 16777226; PMID: 18983665; PMID: 21436684). Capsaicin was shown to induce inhibition of mitogen and interleukin-2-stimulated T cell proliferation (PMID: 8784269). TRPV1 signalling inhibits differentiation and activation of human dendritic cells (PMID: 19397909). Immunological role of neuronal receptor vanilloid receptor 1 expressed on dendritic cells (PNAS, PMID: 15793000). TRPV1 is expressed by glial cells including microglia, astrocytes (reviewed in PMID: 28342781). Capsaicin and TRPV1 regulate gene expression in macrophages (PMID: 14530214). TRPV1 shows an activating effect over the immune system and regulates T cell receptor-induced signalling (Bertin et al., 2014, Nat. Immunol., PMID: 25282159). Overall, the function of the gene is more important than the expression level of the gene.

We acknowledge this possibility and have reformulated our statement related to gene expression – which is now referred to as “not detected using RNA sequencing approaches”.

It is worth noting that in addition to the sequencing data (Extended data Figure 1, 3), we also address the function/presence of TRPV1 on non-neuronal cells by showing that:

- Capsaicin does not impact B16F10 cell survival (Rebuttal Figure 2)
- Cultured CD8 T-cells failed to respond to capsaicin (Rebuttal Figure 2)
- CD8⁺ T-cells or B16F10 cells failed to respond to QX-314, whose activity depends on the presence of activated TRP channels (Extended data Figure 8).
- We used single-cell RNA sequencing and CITEseq to profile B16F10 cell-draining lymph nodes. In the absence of nociceptor neurons, we found drastic differences in the number and transcriptome of tumor draining lymph node lymphocytes and myeloid cells. In healthy mice, the ablation of nociceptors had no impact on the number and transcriptome of lymph node lymphocytes and myeloid cells. (Rebuttal Figure 3).

[Redacted figure]

Rebuttal Figure 3

3) It is unclear how thermal pain threshold was calculated. “Radiant heat was applied to the hindpaw and the time for withdrawal was measured”. I guess your original data is withdrawal latency (second). Please describe your baseline latency. Also, how big is your light spot? If the light spot is big enough, heat stimulation may also involve the adjacent intact tissue, which will show hypersensitivity.

The thermal sensitivity was measured using the “gold standard” Ugo Basile Hargreaves apparatus ([www.ugobasile.com; #37570](http://www.ugobasile.com/#37570)), following the classic method developed by Ken Hargreaves (PMID: 3340425; 5600 citations).

In brief, the mice were placed on a glass plate of a Hargreaves’s apparatus and stimulated using radiant heat (infrared beam). The infrared beam intensity was set at 44 and calibrated to result in a ~12 seconds withdrawal time in acclimatized wild-type mice. An automatic cut-off was set to 25 seconds to avoid tissue damage. The radiant heat source was applied to the dorsal surface of the hind paw and latency measured as the time for the mouse to lift/lick/withdraw the paw.

As per the manufacturer, the system is designed with a “*proprietary specific filter, cutting off the visible part of the light spectrum, which would disturb the animal on test and provide an unwanted clue.*” limiting our ability to measure the output beam size on the paw. Nevertheless, the IR generator has an aperture of ~1mm and the distance between the top of the

light generator and the plexiglass is small (~2.5mm). By comparison, the lesion site can measure up to 10-12mm, and as the IR beam is likely highly focused – it is unlikely to significantly impact adjacent intact tissue.

4) “Additionally, melanoma patients often experience chronic itching — a sensory feature that involves an overlapping somatosensory circuitry with the one responsible for transmitting pain information.” It is interesting to mention itch in your model. However, pain can suppress itch, and loss of pain or analgesia can induce itch. Here is a typical diagnosis of melanoma in patients: “*The skin lesion may feel different and may itch, ooze, or bleed, but a melanoma skin lesion usually does not cause pain.*”

A retrospective analysis of 306 medical records of skin melanoma patients revealed that 38.2% experienced pain (DOI: 10.5935/1806-0013.20160010). Similar pain prevalence was found in small sample size study performed in anorectal malignant melanoma patients (~40%; DOI: 10.1007/s12032-014-0445-2) and vulvar melanoma patients (DOI: 10.4236/ojog.2012.22023).

We found that ~30% of flank-inoculated wildtype mice developed scratching behaviour/itching fourteen days after B16F10 injection; a feature that was absent in mice whose nociceptor neurons were genetically ablated or pharmacologically silenced (**Rebuttal Figure 4**). Mice inoculated with non-tumorigenic keratinocytes (MPEK) show no itching behaviour.

Itch is certainly a prevalent symptom in melanoma and will be mediated by specialized pruriceptor sensory neurons that are distinct from nociceptors and the relative contribution of pruriceptors to tumor-innervating neurons dampening of cancer immunosurveillance warrants further investigation, as does the potential impact of pain on itch and itch on pain in these patients.

Rebuttal Figure 4

REFEREE #3 (*T cells, exhaustion*). The authors have done a great job in revising the MS and I think it is a significant and important report for the field.

We would like to thank the reviewer for their positive review.

3.1. However, there is one issue that remains unresolved and that I had raised in the previous rounds of revisions. The authors make a very clear statement by saying in the abstract and in the caption of Figure 4 that "Nociception controls T cell exhaustion." I think this is a clear exaggeration. I agree that there is an effect on the phenotype of T cells, but the magnitude of the changes does not justify the conclusion that nociception "controls" depletion. In my opinion, this implies a much more profound effect than what the authors show. Based on the data presented, the authors should tone down this statement and conclude rather that nociception influences and modulates exhaustion rather than controlling it.

We agree with the reviewer's comment and have toned down our conclusions. We now refer to nociceptor neurons "attenuation", "modulation" or "affect" on cancer immunosurveillance.

3.2. Figure 4J: "RAG1^{-/-}→ RAMP1wt CD8 + RAMP1^{-/-} CD8". The caption should be changed to 'RAMP1wt CD8 + RAMP1^{-/-} CD8→ RAG1^{-/-}, to reflect that the cells were transferred into RAG mice. This also applies Figure 4H, I.

We have amended the caption and text as per the suggestion.